# High-throughput screening of genetic and cellular drivers of syncytium formation induced by the spike protein of SARS-CoV-2

Charles W. F. Chan [1,2,8], Bei Wang[1,2,8], Lang Nan [3,4,8], Xiner Huang [5,8], Tianjiao Mao[3,4], Hoi Yee Chu[1,2], Cuiting Luo[5], Hin Chu [5,6,7] ✉, Gigi C. G. Choi [1,2,3] ✉, Ho Cheung Shum[3,4] ✉ & Alan S. L. Wong [1,2] ✉

Mapping mutations and discovering cellular determinants that cause the spike protein of severe acute respiratory syndrome coronavirus 2 (SARS-CoV-2) to induce infected cells to form syncytia would facilitate the development of strategies for blocking the formation of such cell–cell fusion. Here we describe high-throughput screening methods based on droplet microfluidics and the size-exclusion selection of syncytia, coupled with large-scale mutagenesis and genome-wide knockout screening via clustered regularly interspaced short palindromic repeats (CRISPR), for the large-scale identification of determinants of cell–cell fusion. We used the methods to perform deep mutational scans in spike-presenting cells to pinpoint mutable syncytium-enhancing substitutions in two regions of the spike protein (the fusion peptide proximal region and the furin-cleavage site). We also used a genome-wide CRISPR screen in cells expressing the receptor angiotensin-converting enzyme 2 to identify inhibitors of clathrin-mediated endocytosis that impede syncytium formation, which we validated in hamsters infected with SARS-CoV-2. Finding genetic and cellular determinants of the formation of syncytia may reveal insights into the physiological and pathological consequences of cell–cell fusion.

Severe acute respiratory syndrome coronavirus 2 (SARS-CoV-2), which caused the worldwide COVID-19 pandemic, has been continuously evolving since its emergence, and numerous variants with different degrees of infectivity and lethality have been arising. Predicting how the variants' pathogenicity changes with their acquired mutations and understanding their interactions with host cell factors are critically important for formulating strategies to confront their threats to global public health and prepare for future outbreaks.

SARS-CoV-2 infects cells by binding its surface spike protein with the host cell receptor angiotensin-converting enzyme 2 (ACE2)[1]. The spike protein is a viral fusogen that allows virus–cell fusion and cell–cell fusion following cleavage and priming by surface or endosomal

[1]Laboratory of Combinatorial Genetics and Synthetic Biology, School of Biomedical Sciences, The University of Hong Kong, Pokfulam, Hong Kong SAR, China. [2]Centre for Oncology and Immunology, Hong Kong Science Park, Shatin, Hong Kong SAR, China. [3]Department of Mechanical Engineering, The University of Hong Kong, Pokfulam, Hong Kong SAR, China. [4]Advanced Biomedical Instrumentation Centre, Hong Kong Science Park, Shatin, Hong Kong SAR, China. [5]State Key Laboratory of Emerging Infectious Diseases, Department of Microbiology, The University of Hong Kong, Pokfulam, Hong Kong SAR, China. [6]Centre for Virology, Vaccinology and Therapeutics, Hong Kong Science Park, Shatin, Hong Kong SAR, China. [7]Department of Infectious Disease and Microbiology, The University of Hong Kong-Shenzhen Hospital, Shenzhen, People's Republic of China. [8]These authors contributed equally: Charles W. F. Chan, Bei Wang, Lang Nan, Xiner Huang. ✉e-mail: hinchu@hku.hk; gigichoi@hku.hk; ashum@hku.hk; aslw@hku.hk

**Table 1 | Evaluation of methods for the high-throughput screening of cell–cell fusion**

| Methods | Experimental setup needed | Quantification methods | Throughput for genetic screens | Comparison among variants/perturbations |
|---|---|---|---|---|
| High-content imaging (used in refs. 19,20) | High-content screening microscope | **Less precise**; using the area of syncytium/fraction of fused cells under the microscope | **Limited**; >3,000 drugs were screened in ref. 19 and ~6,000 drugs and >30 spike protein variants were screened in ref. 20. Screening of genetic variants and perturbations requires the individual library constructs to be generated and delivered into microwell arrays for imaging. With an imaging throughput of ~30 s per well, screening a genome-wide CRISPR library of 37,722 sgRNAs is estimated to take ~13 d. | **Potentially higher variation**; There may be a lag time in cell fixation/image acquisition over a large number of samples to evaluate the fast cellular process. The non-pooled experimental setup may be more subject to technical variations such as cell density. |
| Droplet microfluidics-based method (established in this study) | Droplet microfluidics system and FACS sorter | **More quantitative**; using NGS | **Moderate**; 760 spike protein variants were screened in this study. Genetic variant libraries were pooled assembled and delivered into cells. Based on the current system, obtaining droplets with the paired cells for screening a DMS library of ~380 protein variants took ~40 min, followed by ~2 h of FACS sorting. At this rate, screening a genome-wide CRISPR library of 37,722 sgRNAs is estimated to take ~42 h for droplet encapsulation and ~126 h for FACS sorting (total: ~7 d). | **Less variation**; Pooled assay allows head-to-head comparison of the variants/perturbations under the same experimental setting. Less interference by the neighbouring cells/syncytia due to compartmentalization. Fusion-incompetent cells are greatly depleted, thus offering a wider assay range in defining the enriched fusion-competent variants. |
| Size-exclusion selection-based method (established in this study) | Cell strainer and FACS sorter | **More quantitative**; using NGS | **High**; 760 spike protein variants were screened in this study. Genetic variant and perturbation libraries are pooled assembled and delivered into cells. Only a filtration step using a cell strainer is needed to collect the fused cells, which takes seconds. With a reverse selection approach to collect unfused cells, a genome-wide CRISPR library with 37,722 sgRNAs was screened in this study and it took ~8 h to sort the unfused cells (total: ~8 h). | **Less/moderate degree of variation**; Pooled assay allows head-to-head comparison of the variants/perturbations under the same experimental setting. There are chances that some fusion-incompetent cells are trapped by neighbouring syncytia as bystanders and retained on the cell strainer, thus resulting in more variation and a narrower assay range. |

proteases of the target cells[2,3]. Extensive lung tissue damage, with the presence of large multinucleated syncytial pneumocytes formed due to cell–cell fusion, is characterized in post-mortem samples from individuals who died of COVID-19 and syncytia are considered as a frequent feature of severe COVID-19 (ref. 4). Syncytia are formed by two or more cells fusing. SARS-CoV-2 induces syncytium formation when the spike protein on the surface of an infected cell interacts with receptors on neighbouring cells. Syncytia potentially contribute to pathology by facilitating viral dissemination, cytopathicity, lymphocyte elimination and inflammatory response[5,6]. Via syncytia, SARS-CoV-2 can also enter cells through cell fusion between infected and uninfected cells, thus contributing to the spread of the virus through cell-to-cell transmission[7].

The presence of multinucleated syncytia in the lung tissues of COVID-19 patients appears to be primarily determined by the fusogenicity of the spike protein of SARS-CoV-2. It was confirmed that the SARS-CoV-2 spike protein alone is sufficient to drive syncytium formation, as in vitro co-culture assay showed that the cells expressing spike only, in the absence of other SARS-CoV-2 viral proteins, could fuse with the neighbouring ACE2-expressing cells and form syncytia[3]. A cryo-electron microscopy-determined structural model of the trimeric spike has revealed that spike is expressed on the surface of the virions as a trimer, leading to the crown-like appearance[8]. Many studies, including protein structure and biochemical studies, have been performed to understand how the spike protein works[9,10]. The spike protein, a heavily glycosylated type I transmembrane protein with 1,273 amino acids, comprises two functional subunits, S1 and S2. S1 contains a receptor-binding domain (RBD) responsible for viral attachment to the host receptor, and the S2 subunit contains a protease cleavage site (S2'), a hydrophobic fusion peptide and two heptad repeat regions responsible for membrane fusion for entry into host cells. At the S1/S2 boundary, there is a polybasic cleavage site that is not found in SARS-CoV-1 or other SARS-like coronaviruses[11]. S1/S2 is cleaved during virus assembly in virus-producing cells by furin. When infecting target cells, the spike protein interacts with ACE2. The pre-cleaved spike is further cleaved by transmembrane serine protease 2 (TMPRSS2) to allow plasma membrane entry. The spike that is not cleaved by TMPRSS2 is endocytosed and subsequently cleaved by endosomal proteases such as cathepsin L for endosomal entry. Following up on these analyses, the next important step is to systematically map the sequence-to-phenotype relationship of the SARS-CoV-2 spike protein and determine how SARS-CoV-2 spike mutations affect its ability to induce fusion and disease severity.

During viral evolution, Alpha, Beta and Delta strains of SARS-CoV-2 spike, considered variants of concern (VOCs), emerged, forming more and larger syncytia in the infected cells than their parental D614G strain (designated as wild-type (WT) in this study)[12]. In molecular-level studies, P681H and P681R mutations found in the S1/S2 cleavage site of the Alpha and Delta variants, respectively, were shown to increase syncytium formation[12,13]. The two mutations may expose the furin cleavage site of spike such that its proteolytic cleavage occurs more readily, stimulating subsequent fusion[13,14]. The Omicron variant of the spike protein also promotes the formation of obvious syncytia, although the extent is less than that induced by the Delta variant[15,16]. The fusion abilities vary among the ever-growing numbers of Omicron subvariants, which have acquired additional mutations[17]. Rapid characterization of the fusion ability of emerging spike variants will aid the identification of mutations of concern.

Large-scale screening to characterize cell–cell interaction/fusion instead of single-cell behaviour is technically more challenging, as grouping of two different cell types for characterization of their interaction requires compartmentalization methods such as droplet microfluidics[18]. In all previous studies, individual spike variants were constructed one-by-one, and their syncytium-forming potential was individually monitored by microscopy after co-culturing two types of cells: spike-presenting sender cells and fusion-permissible receiver cells[3], and quantified on the basis of the area of syncytium formed under the microscope, which limits precision. The scalability of those experiments constrains its utility for characterizing the many subvariants and emerging variants. High-content imaging could be automated

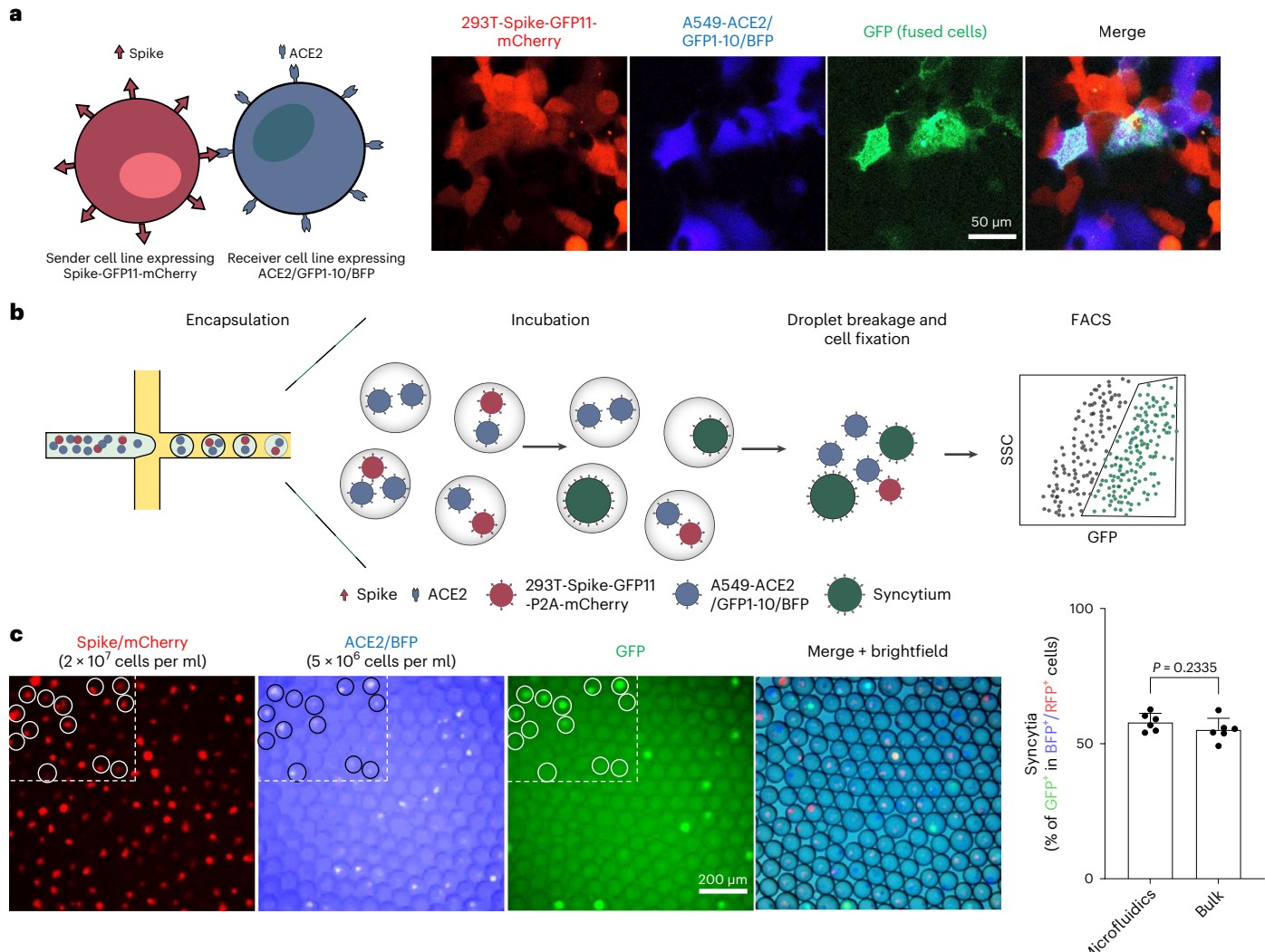

**Fig. 1 | A droplet microfluidics-based system for high-throughput screening of syncytium formation. a**, Co-culture of spike-expressing sender cells (mCherry⁺) and ACE2-expressing receiver cells (BFP⁺) resulted in syncytium formation (GFP⁺) due to cell–cell fusion. Scale bar, 50 μm. **b**, Workflow for cell encapsulation and incubation in droplets and the detection/collection of GFP⁺ syncytium (boxed). Side scatter (SSC) was used to identify the cell population.

**c**, Representative images of sender cell–receiver cell fusion in droplets. A fusion rate of 57.9% ± 3.3% (data shown are mean ± s.d., *n* = 6 biological replicates, with *P* value from unpaired two-tailed Student's *t*-test) was measured by FACS after 24 h of incubation in droplets, which is comparable to the bulk setting (55.2% ± 4.3%) in regular co-culture experiments without using droplet microfluidics. Red fluorescent protein (RFP) refers to the mCherry signal detected under FACS.

for large-scale profiling experiments[19,20], but there is lag time in imaging the first well until the last one among the tens of thousands. The extended time delay is not ideal for capturing rapidly progressing cellular processes, such as syncytium formation that can be seen within an hour after the sender and receiver cells encounter[20], among a large library of variants for their head-to-head comparison (Table 1). Moreover, screening of unique genetic variants/perturbations requires each of the library constructs to be individually built and delivered into single wells of the arrays, which is technically demanding and expensive. Cell–cell fusion rate across experiments can also be greatly affected by cell density and sender-to-receiver cell ratio; thus, non-pooled assays are more subject to technical variations than pooled experiments when doing comparisons across many variants. A pool-based method coupling cell–cell fusion with a screening readout of retroviral vector particle packaging and release that transfer genes encoding the fusion-competent membrane protein was previously reported[21]. This method could be adopted for studying the SARS-CoV-2 spike protein. Adjusting cell density and cell type ratios in such pooled experiment would minimize syncytia with polyclonal sender cells, which increase

noise due to some relatively less fusion-competent cells being fused with neighbouring syncytia containing the more fusion-competent variants and enriched together as large syncytia. However, before this work, to the best of our knowledge, no large-scale quantitative analysis that directly reveals how mutations of the spike protein affect the cell–cell fusion process has been reported.

In addition, how the spike protein hijacks the host cell machinery to fuse cells together is not well understood. Understanding how spike-induced cell–cell fusion occurs may allow us to develop effective strategies to mitigate syncytium formation induced by SARS-CoV-2. Several genome-wide CRISPR screens were carried out in the search for host factors required for SARS-CoV-2 infection[22–28], in which live SARS-CoV-2 was used to infect the cells; a suite of its viral proteins were expressed and each of them could hijack various machineries of the host cells to aid viral replication, assembly, release and cell death. Currently, no systematic studies pinpoint the cellular determinants of syncytium formation to fill in this knowledge gap.

Here we establish droplet microfluidics and size-exclusion selection strategies to study cell–cell fusion, and couple them with

large-scale mutagenesis and CRISPR screening. In this study, we experimentally scanned mutations over two regions of SARS-CoV-2 spike: the fusion peptide proximal region and the furin cleavage-site region, for their impact on spike's syncytium-forming potential, as well as screened through the entire human genome for host factors that are crucial for spike-induced syncytium formation. We developed these methods in human cells, which offer the advantage of conferring the post-translational modifications, including glycosylation, to spike that it acquires under physiological conditions to induce syncytia. Our work provides opportunities to rapidly understand the pathological consequence (that is, syncytium formation) of SARS-CoV-2 genetic variation by profiling a large panel of variants, enabled by our methods, and identifies candidate genetic factors in human cells that could serve as potential therapeutic targets for combating syncytium formation. Without these methods, it would be technically challenging to study these targets en masse.

## Results

### A droplet microfluidics-based system for high-throughput screening of syncytium formation

We aimed at establishing high-throughput systems to profile the syncytium-forming potential systematically and quantitatively across sender and receiver cell variants. To visualize cell–cell fusion, we adapted the green fluorescent protein (GFP)-split complementation system wherein the sequence of GFP was split between the tenth and the eleventh β-strand to produce GFP1–10 and GFP11, which are non-fluorescent by themselves; however, the reconstituted GFP becomes fluorescent upon complementation[29]. By co-culturing the fusogenic sender cells transfected with spike and the small GFP11 fragment, and the receiver cells transfected with ACE2 and the GFP1–10 fragment, the spike-mediated cell–cell fusion can be visualized by the GFP signal (Extended Data Fig. 1a). To minimize expression variations of the spike protein, GFP1–10 and GFP11 among cells, we created a HEK293T sender cell line stably expressing the spike protein with GFP11-P2A-mCherry, as well as a receiver A549 cell line expressing ACE2, GFP1–10 and blue fluorescent protein (BFP), through lentiviral transductions. A549-ACE2 cells are derived from human lung epithelial cells and are widely used as a model to study SARS-CoV-2 biology. The fast-growing nature of A549 is suitable for high-throughput screening experiments. HEK293T cells have no ACE2 expression and were used as the spike-expressing sender cells to avoid cell fusion in the stable cell line, as a low level of ACE2 was detected in A549 cells[30]. After co-culturing the sender and receiver cells for 24 h, we observed obvious syncytium formation (Fig. 1a).

To enable parallel profiling of the syncytium-forming potential of cell variants, we set up a microfluidic system to compartmentalize individual sender and receiver cells in droplets. Although the droplet generation frequency with a standard single-channel device can be as high as 1.5 kHz, the generation time could last as long as 8 h for generating 42 millions of droplets needed for a screening experiment of ~1,000 variants. Such a long duration of microfluidic processing is not desired as it reduces cell viability. Therefore, we designed an 8-channel device to increase the throughput (Extended Data Fig. 1b). In comparison, the 8-channel device can maintain an ultrahigh overall frequency of 12 kHz to complete the droplet generation process in ~1 h. To minimize the uneven distribution of the contents of the single-cell and oil inlets across the eight drop makers, we used a symmetrical layout for the 8 channels such that the flow rates of the aqueous and oil phases can be evenly distributed in each channel. We optimized the flow rates of two phases to generate droplets for encapsulating human cells. In our experiments, we observed droplets with diameters ranging from 70–76 μm (Extended Data Fig. 1b). The coefficient of variation (c.v.) was 1.5%, which is on par with other microfluidics settings that generate droplets with a c.v. < 3%. We next determined the droplet (co)occupancy rates when supplied with different concentrations of HEK293T sender cells, A549-ACE2 receiver cells and their mixtures (Extended Data Fig. 1c,d). For the mixtures, we set the concentration of one cell type to undergo variant screening (that is, sender or receiver) as the limiting factor and ensured that most droplets contained only a single cell of that type. After cell encapsulation and incubation, the droplets were demulsified and collected for cell fixation. GFP-positive syncytia were analysed and sorted out by fluorescence-activated cell sorting (FACS) (Fig. 1b). With this system and experimental setup, we tested and fixed the incubation time at 24 h in subsequent experiments, when a cell–cell fusion rate of 57.9 ± 3.3% was achieved in the droplets encapsulated with both mCherry-positive sender and BFP-positive receiver cells to enable our assay to detect both fusion-enhancing and inhibiting variants (Fig. 1c). The cell–cell fusion rate in droplets is comparable to what we observed in the bulk setting (that is, without using droplet microfluidics) (Fig. 1c).

### Deep mutational scan (DMS) of the syncytium-forming potential of SARS-CoV-2 spike variants

With the droplet microfluidic screening system, we profiled the syncytium-forming potential of spike variants en masse to define residues important for SARS-CoV-2-induced syncytium formation and understand how mutations observed in current and future SARS-CoV-2 isolates may impact their syncytium-forming abilities (Fig. 2a). We built a pooled saturation mutagenesis library of spike variants (that is, 20 amino acid residues × 19 positions from residues 836 to 854) over the fusion peptide proximal region (FPPR) (Supplementary Fig. 1a). We used the degenerate 'NNS' codon to encode all 20 amino acids and also included stop codons as negative references. We chose the FPPR as one of our regions of interest because it is considered relevant to the conformational changes in spike[31]. Also, various mutations at this region have been reported in the COG-UK-ME, a database that has documented SARS-CoV-2 mutations that have been detected in the COG-UK genome sequence dataset[32] (Supplementary Fig. 1b).

**Fig. 2 | Deep mutational scanning of SARS-CoV-2 spike's syncytium-forming potential using the droplet microfluidics-based system. a**, Workflow for deep mutational scanning in sender cells using a droplet microfluidics-based system. GFP⁺ syncytia containing the fusion-competent spike variants are sorted out for NovaSeq-based sequencing. **b**, Heat maps depicting how all single mutations affect the syncytium-forming potential of spike's FPPR and furin cleavage site regions. Squares are coloured by mutational effect (that is, FC) according to colour bars on the right, with red and purple indicating syncytium-enhancing and inhibiting substitutions, respectively. FC represents each variant's relative abundance in the sorted GFP-positive cell pool vs the cell pool before mixing and is normalized to WT. Spike variants with increased syncytium-formation potential were enriched (with fold change >1), while those with decreased syncytium-formation potential were depleted (with fold change <1). Grey cross, mutations with no measurement; black dot, SARS-CoV-2 WT amino acid; *, stop codon. **c**, Validation of syncytium-enhancing and inhibiting mutations at the FPPR in WT D614G and Omicron spike. GFP-split complementation system assay was applied and representative images are shown. **d**, Quantification of the syncytium area (top) and average size of syncytium (bottom) for spike variants in **c**. Data shown are mean ± s.d.; n = 6 biological replicates. P values coloured in blue indicate statistical comparison between individual spike D614G and Omicron mutants to the D614G WT. P values coloured in red indicate statistical comparison between individual Omicron mutants to the omicron WT. Statistical significance was determined using one-way ANOVA. ***P < 0.001, ****P < 0.0001. **e**, Mutability score for individual position of spike protein studied in **b** and RBD. The number indicates the position of the amino acid on the spike protein. The letter indicates the wild-type amino acid on the position. **f**, Mutational effects on syncytium formation revealed by the pooled screen using the droplet microfluidics-based system and individual validation assay for the spike variants. Black dots, validated spike FPPR variants.

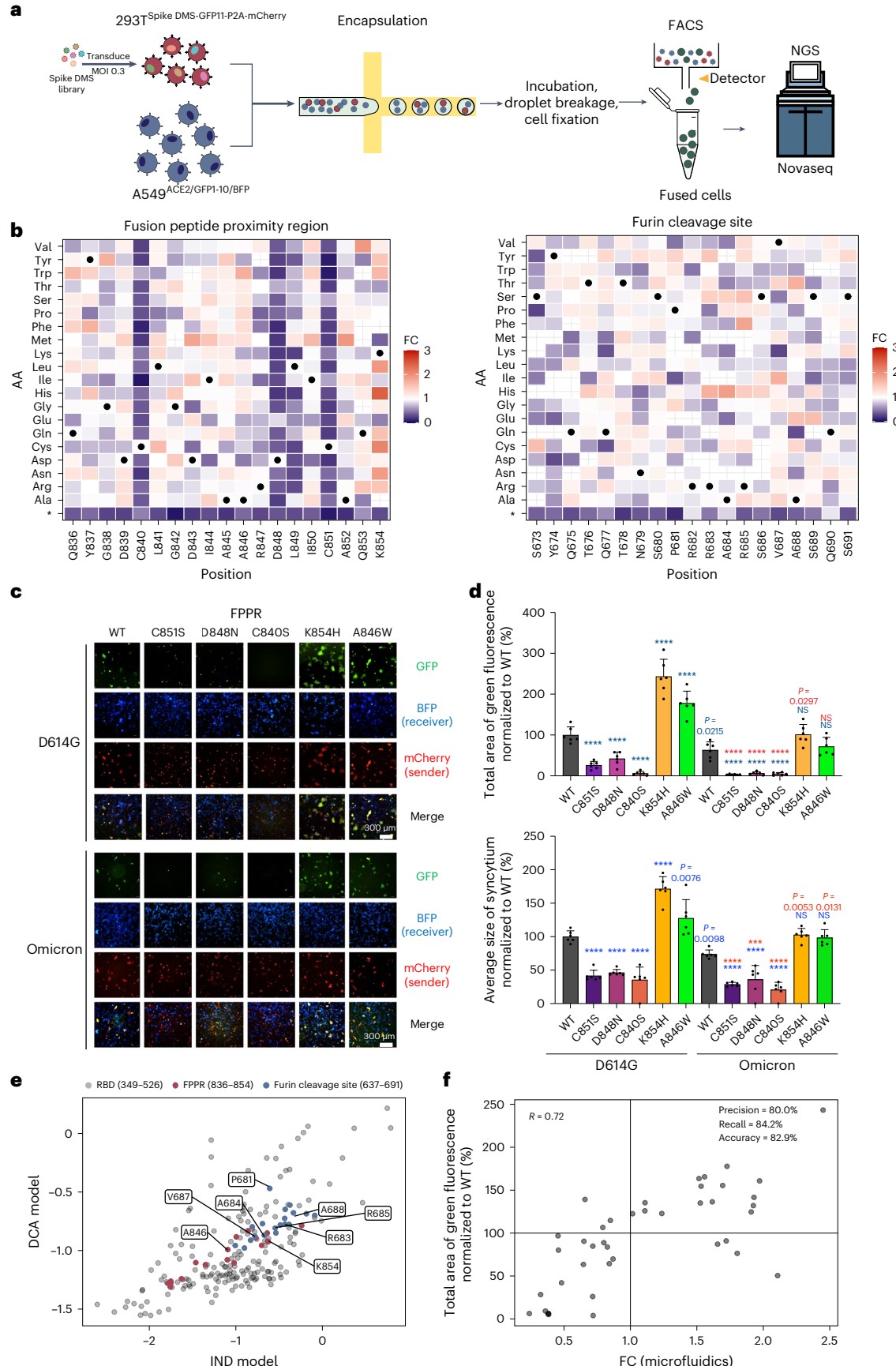

The FPPR is located close to the S1/S2 boundary and the S2' cleavage site. One proposed model[31] is that the flexible FPPR region is occasionally flipped out of position due to intrinsic protein dynamics, which allows the RBD to sample the up conformation and capture ACE2 on the target host cell membrane. This rearrangement would in turn expose the S2' cleavage site and potentially also the adjacent S1/S2 site to initiate the cascade of proteolytic cleavage of spike and its structural transition from the prefusion to the post-fusion state, which induces cell membrane fusion. We used lentiviruses to deliver the pooled library of the spike variants with GFP11-P2A-mCherry into HEK293T sender cells (Supplementary Fig. 1c). A multiplicity of infection (MOI) of ~0.3 was used to ensure that most cells acquired a single copy of the variant. To ensure that the high-coverage library contained a sufficient representation of each variant (>500-fold coverage of each spike variant), we used >1,500-fold more cells for lentiviral infection than the size of the library being tested. To maintain the library distribution throughout the entire screening process, all the mCherry-positive infected sender cells were sorted out, expanded and then co-cultured with receiver A549 cells expressing ACE2, GFP1–10 and BFP for 24 h before droplet breakage and cell fixation. We performed NovaSeq-based sequencing on the genomic DNA from the collected GFP-positive cells to quantify the abundance of each spike variant as an index of syncytium-forming potential (Supplementary Fig. 1d). We collected the genome DNA from the infected sender cell pool immediately before mixing with the receiver cells for comparison and confirmed that there were no overrepresented variants in the library (Supplementary Fig. 1c). Spike variants that have increased syncytium-formation potential were enriched (with fold change >1, comparing each variant's relative abundance in the GFP-positive cell pool vs the cell pool before mixing, normalized to WT), while those that have decreased syncytium-formation potential and those that acquired a premature stop codon after mutation were depleted (with fold change <1).

Our profiling result showed that all spike variants with mutations at C840, D848 and C851 tended to have decreased syncytium formation potential (Fig. 2b and Source Data). Individual validation assays confirmed that mutations at C840, D848 and C851 reduced syncytium formation when compared with WT spike (Fig. 2c,d). Similar effects were also detected when the mutations at C840, D848 and C851 were grafted onto the Omicron variant of spike (Fig. 2c,d and Extended Data Fig. 2a), indicating their conserved roles in determining the spike protein's function. These results are in good agreement with the previously reported structural framework of spike, wherein an internal disulfide bond is formed between C840 and C851, which is reinforced by a linked salt bridge that involves D848. This framework was proposed to stabilize the spike protein structure[31]. Interestingly, we found that the mutations at C840, D848 and C851 do not necessarily affect spike's cell surface localization, the cleavage of its S1 subunit and its binding ability with ACE2 (Extended Data Fig. 2b–d and Source Data), suggesting that this framework determines spike's syncytium-forming potential independent of these factors. Of note, we also observed spike variants with enhanced ability to form syncytia (Fig. 2b and Source Data) and mutability comparable to other residues at the spike's RBD (Fig. 2e, Supplementary Fig. 2 and Source Data). Using the observed fold change

distributions of WT spike variants with synonymous codons and those with stop codons, we defined a threshold of fold change >1.625 at which no variant with stop codons was identified, thus potentially minimizing the identification of false positives (Supplementary Fig. 3). With this, 11 syncytium-enhancing hits were identified (Source Data). We performed individual validation assays and confirmed 9 hits exhibiting greater syncytium-forming potential than WT (that is, resulted in larger average size of syncytium and/or total area of syncytia (Fig. 2c,d and Extended Data Fig. 3a,b)). Among them, the K854H and A846W variants showed the greatest enhancement in forming syncytia. K854 has been shown to form a salt bridge with D614 (ref. 31) while the D614G mutation became the first identified SARS-CoV-2 isolate that quickly became the predominant form worldwide upon its emergence, and this variant exhibits more efficient cell entry[33–35]. D614G was reported to weaken its salt bridge formation with K854, which primes the spike protein to adopt a more opened (that is, RBD-up) conformation for ACE2 binding and subsequent membrane fusion, as well as affect the FPPR density between residues 842 and 846 (refs. 31,36–38). Our molecular modelling showed that K854H further decreases its interaction with G614 (Extended Data Fig. 4a), and the A-to-W substitution at residue 846 increases its interaction with residue 843, which could modulate the FPPR density (Extended Data Fig. 4b). The mechanisms by which these K854H and A846W mutations enhance syncytium formation remains to be delineated, while they appeared to increase spike's ACE2 binding affinity (Extended Data Fig. 2b,c), as reported for D614G[39], but not affect its cell surface expression and S1 subunit cleavage (Extended Data Fig. 2d). Notably, the K854H variant confers Omicron's spike with syncytium-forming potential comparable to that of WT D614G (Fig. 2c,d). In sum, our results reveal that syncytium-enhancing mutations such as K854H exist at FPPR and should be monitored, as they may be observed and emerge in future viral isolates.

To evaluate the data quality of our screen more comprehensively, we analysed how the screen read counts of variants correlate with their syncytium-forming potency. In addition to the above 14 validated variants, we randomly picked 27 variants in the FPPR library and validated their syncytium-forming potentials using individual assays (Extended Data Fig. 3a,b). Overall, we observed a good correlation ($R$ = 0.85) between the quantified average size of the formed syncytium and total syncytium area (Extended Data Fig. 3c). We also observed a high consistency ($R$ = 0.72) between the screen and individual validation results (Fig. 2f), highlighting that our screen provides quantitative measurements of the variants' syncytium-forming potentials reasonably well. Among the total of 41 variants, 19 were found to be syncytium-enhancing in the individual validation assays. Of the 19 variants, 16 were discovered as syncytium-enhancing 'hits' in our screen (Fig. 2f), indicating a high true discovery rate for our system in defining syncytium-enhancing mutations. The screen could also identify syncytium-inhibiting variants, albeit some 'false positives' (that is, 4 out of 22 inhibitory variants validated in the individual assays) could be detected (Fig. 2f). The precision, recall and accuracy of our screen were 80.0%, 84.2% and 82.9%, respectively.

We also applied the droplet microfluidic screening system to examine the region harbouring the furin cleavage site in addition

**Fig. 3 | A size-exclusion selection-based strategy for high-throughput screening of syncytium formation. a**, Workflow for DMS in sender cells using a size-exclusion selection-based system. Large syncytia are collected using the cell strainer, while small GFP⁺ syncytia are collected by FACS. The two populations containing the fusion-competent spike variants are subjected to NovaSeq-based sequencing. **b**, Heat maps depicting how all single mutations affect the syncytium-forming potential of spike's FPPR and furin cleavage site regions. Squares are coloured by mutational effect (that is, FC) according to colour bars on the right, with red and purple indicating syncytium-enhancing and inhibiting substitutions, respectively. FC represents each variant's relative abundance in the syncytia collected using the cell strainer or FACS vs the cell pool before mixing

and is normalized to WT. Spike variants with increased syncytium-formation potential were enriched (with fold change >1), while those with decreased syncytium-formation potential were depleted (with fold change <1). Grey cross, mutations with no measurement; black dot, SARS-CoV-2 amino acid; *, stop codon. **c**, Correlation of the profiling results (that is, FC) obtained using droplet microfluidics-based and size-exclusion selection-based strategies. Validated syncytium-enhancing and inhibiting mutations/residues are highlighted and labelled. **d**, Mutational effects on syncytium formation revealed by the pooled screen using the size-exclusion selection-based system and individual validation assay for the spike variants.

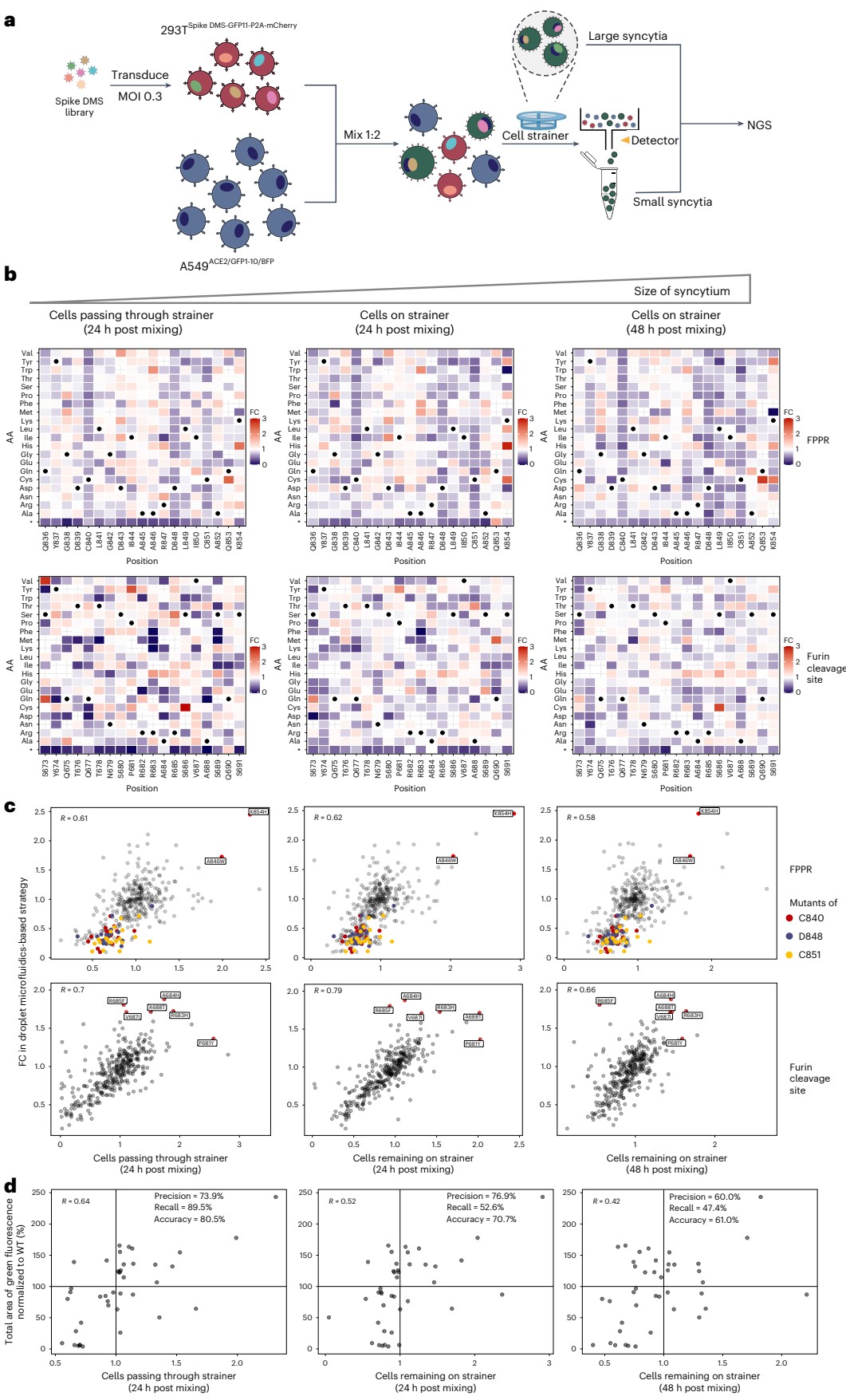

to the FPPR. Comparison between the SARS-CoV-2 and SARS-CoV-1 protein sequences identified the gain of a short stretch of sequence (that is, PRRA) before the S1/S2 region in the SARS-CoV-2 spike that creates a putative furin cleavage site[40]. It was found that deleting the PRRA sequence from the SARS-CoV-2 spike protein abolishes its ability to form syncytia and viral transmission, while inserting the PRRA sequence into the SARS-CoV-1 spike confers the ability to fuse cells[5,41]. This finding indicates that PRRA and potentially its surrounding sequence could be important in the process of membrane fusion leading to syncytium formation by facilitating interactions with and efficient cleavage by proteases. For example, it was reported that furin protease can recognize motif sequences with X-Arg-X-Lys/Arg-Arg-X (that is, XBXBBX, where B is a basic amino acid residue and X is a hydrophobic residue)[42]. Mutations in this region could impact the syncytium-formation potential of the spike protein. Indeed, P681H or P681R mutation is present in the evolved Alpha, Delta and Omicron variants, indicating the high mutability of this residue. P681R in the Delta variant was shown to slightly increase syncytium formation[13], which was also observed in our deep DMS result, while the increase was at a much lesser extent than another substitution, P681Y (Fig. 2b). P681Y mutation was reported in an immunocompromised patient with persistent SARS-CoV-2 Omicron BA.1 subvariant replication[43]. Here we performed individual validation assays and confirmed P681Y's syncytium-enhancing effect (Extended Data Fig. 5a,b), as well as its increased efficiency to be S1-cleaved even in cells with minimal furin and TMPRSS2 expressions (Extended Data Fig. 5c,d). Our DMS results further revealed that most single mutations at PRRA and its neighbouring sequence including the basic residue R685 were not sufficient in abolishing syncytium formation (Fig. 2b). These suggest that despite the high mutability of these sites, many of the single mutants can be efficiently cleaved by the proteases. This could be attributed to the rather flexible motif sequence that the furin protease or other proteases could recognize. Our results isolated several syncytium-enhancing and mutable variants that are present in this region (Fig. 2b,e and Supplementary Fig. 2). At the fold change cut-off of >1.625 as defined in the FPPR screen, 5 syncytium-enhancing hits were identified (Source Data) and 4 of them (that is, R683H, A684H, V687I and A688T) were validated with individual assays showing larger average size of syncytia and/or total area of syncytia (Extended Data Fig. 5a,b). These mutations did not affect the spike's cell surface expression level and S1 subunit cleavage (Extended Data Fig. 5c,d). The potential emergence of P681Y, as well as the other identified syncytium-enhancing mutations at residues 683, 684, 687 and 688 that have similar mutability scores (Fig. 2e and Extended Data Fig. 2) should be monitored.

Overall, our results validated the utility of the droplet microfluidics-based screening approach to profile the syncytium-formation potential of spike variants.

## A size-exclusion selection-based strategy for DMS of spike's syncytium-forming potential

We also explored the feasibility of developing an alternative compartmentation-free selection tactic to enhance the throughput of SARS-CoV-2 spike-mediated syncytia screening. Since large syncytia formed in the co-culture system cannot be subjected to flow-cytometry analysis, we attempted to use a 70 μm cell strainer to collect them. At the same time, we also performed FACS to collect all GFP-positive cells that passed through the cell strainer, that is, the small syncytia resulting from fusion (Fig. 3a). Therefore, we separated fused cells from unfused cells to analyse the relative abundance of each variant within a library of cells by sequencing the DMS region encoded in the integrated lentiviral sequence in the genomic DNA (Fig. 3a). Spike variants that have enhanced syncytium-formation potential were enriched in both collected populations of large and small syncytia, while those that have reduced syncytium-formation potential and those that contain a premature stop codon after mutation were depleted. Using this size-exclusion selection-based strategy, we were able to obtain profiling results that largely resemble the data collected from the droplet microfluidics-based system (Fig. 3b,c and Source Data). Using size-exclusion selection-based strategy could thus offer a simple and scalable way to perform large-scale DMS for syncytia screening.

We evaluated the quality of data collected from the screen via the size-exclusion selection strategy. Among the 19 (out of 41) FPPR library variants with individually validated syncytium-enhancing potentials (Extended Data Fig. 3a–c), 17 were discovered as syncytium-enhancing 'hits' (with fold change >1) in our screen using the GFP-positive small syncytia collected via FACS (Fig. 3d). This gives a true discovery rate similar to that of the droplet microfluidics-based screening approach. The hit number dropped to 10 and 9 when screening larger syncytia collected and remaining on the cell strainer after 24 and 48 h post mixing, respectively (Fig. 3d), resulting in reduced true discovery rates. Indeed, we noted that the cells-remaining-on-strainer-based strategy resulted in less depletion for the fusion-incompetent cells in the pooled assay, which gives greater noise and a narrower assay range in defining the enriched fusion-competent variants (Supplementary Fig. 3). The non-compartmented cell pool in the size-exclusion selection-based system and the longer duration allowed for syncytium formation, potentially favouring the trapping of fusion-incompetent cells by the neighbouring syncytia as bystanders and their retention on the cell strainer. Some relatively less fusion-competent cells may also be fused with neighbouring syncytia containing the more fusion-competent variants and enriched together as large syncytia with polyclonal sender cells. These could account for the relatively less enrichment and likelihood of the true syncytium-enhancing variants (particularly the weaker ones) to be discovered as hits and more non-enhancing variants being isolated as 'false positives', thus increasing the false discovery rate. Among the three experimental parameters for the size-exclusion-based selections, allowing a shorter duration (that is, 24 h) to form smaller-sized syncytia, in particular ones that are small enough to be collected by FACS, is recommended.

All in all, selecting either the droplet microfluidics-based or the size-exclusion selection-based screening strategy could depend on the desired sensitivity and throughput of the genetic screen to be performed (Table 1). Future efforts could explore using droplet

**Fig. 4 | Genome-wide CRISPR screen reveals host factors important for SARS-CoV-2 spike-induced syncytium formation. a**, Workflow for genome-wide CRISPR screening in receiver cells using a reverse selection approach. Large syncytia and small GFP+ syncytia were removed using cell strainer and FACS, respectively. Unfused A549-ACE2-SpCas9-GFP1–10-sgRNA cells were subjected to NovaSeq-based sequencing. **b**, Enrichment of sgRNAs in the unfused receiver cell population revealed by the RRA score and FC. FC represents each variant's relative abundance in the unfused receiver cell pool vs the unmixed cell pool and is normalized to WT. The sgRNA screen hits are highlighted in red and the known tumour suppressor genes are highlighted in blue. The genome-wide CRISPR screen data were collected from two biological replicates. **c**, Validation of screen hits using sgRNA-directed gene knockouts in A549-ACE2-SpCas9-GFP1–10 receiver cells. The HEK293T-GFP-11 sender cells used express WT D614G spike. **d**, Quantification of the syncytium area (left) and average size of syncytium (right) observed in **c**. Data shown are mean ± s.d. (n = 6 biological replicates). **e,f**, Quantitative PCR with reverse transcription (RT–qPCR) (**e**) and western blotting (**f**) results on *CHC* knockdown. **g,h**, Inhibition of syncytium formation by *CHC* knockdown (**g**) and Pitstop2 treatment (**h**). GFP-split complementation system assay was applied as in **c**. Representative images are shown. Data shown are mean ± s.d. (n = 3 (**e**), n = 6 (**g**) and n = 9 (**h**) biological replicates). P values indicated were compared with safe harbour-targeting sgRNA (**d**), control (ctr) shRNA (**e,g**) or DMSO-treated control (**h**). Statistical significance was determined using one-way ANOVA. ****P < 0.0001. Scale bars, 200 μm.

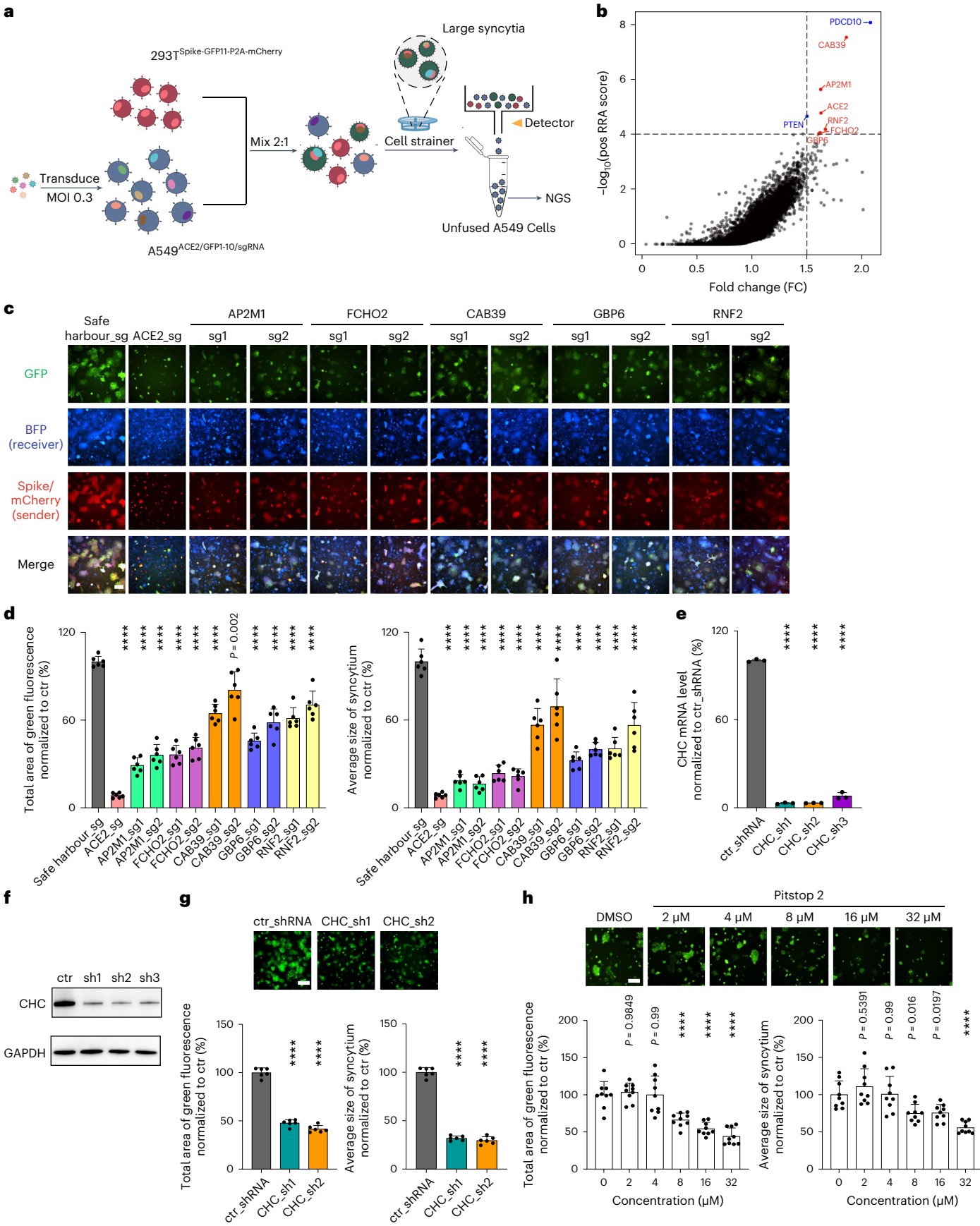

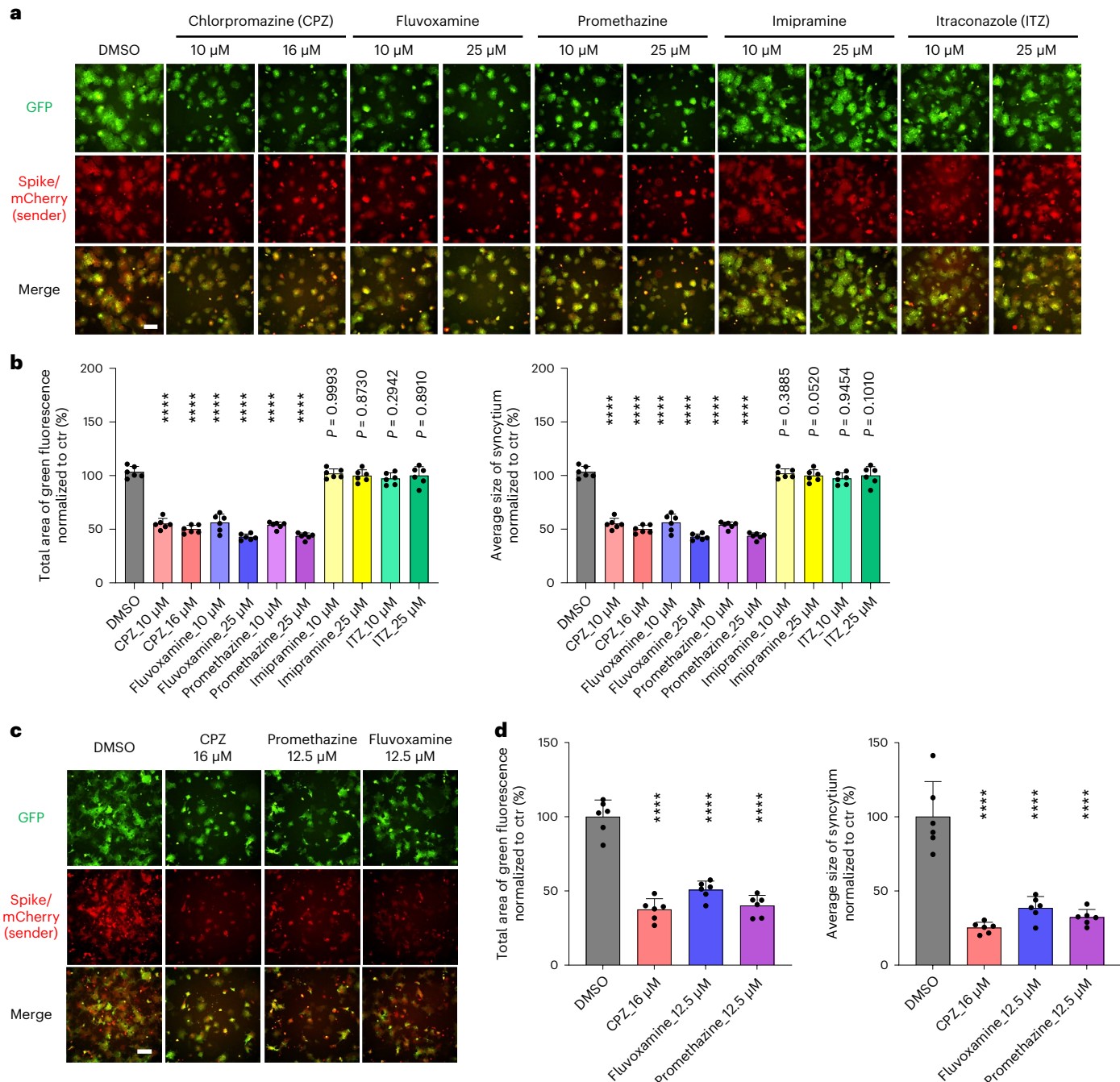

**Fig. 5 | Clathrin-mediated endocytosis inhibitors impede SARS-CoV-2 spike-induced syncytium formation. a–d**, GFP-split complementation system assay was applied using A549-ACE2-GFP1–10 (**a**,**b**) and Vero E6-GFP1–10 (**c**,**d**) as receiver cells. The HEK293T-GFP-11 sender cells used express WT D614G spike.

Representative images are shown. Data shown are mean ± s.d. (*n* = 6 biological replicates). *P* values indicated were compared with DMSO-treated control. Statistical significance was determined using one-way ANOVA. *****P* < 0.0001. Scale bars, 200 µm.

microfluidics to achieve cell size measurement and integrate with fluorescence as dual readouts to further enhance the screening data quality.

### Genome-wide CRISPR screening identifies host factors required for spike-induced syncytium formation

To identify host factors that are required for spike-mediated syncytia, we sought to set up a reciprocal genome-wide CRISPR screen on the basis of the cell–cell fusion system. Because of the large library size needed for genome-wide CRISPR-based knockout screening, we decided to use the size-exclusion selection-based screening strategy. Still, it is technically difficult to directly adopt the selection

approach that collects the fused cell population via the cell strainer or the GFP-positive fused cells to look for depleted single guide RNAs (sgRNAs) representing the genes required for cell–cell fusion in the ultra-large pool of cells. Here we took a reverse selection approach to collect all the sgRNA-infected receiver A549-ACE2-Cas9-GFP1–10 cells that are resistant to fusion when co-culturing with the spike-expressing sender cells, with the aim of isolating host factors crucial for the syncytium formation. In our co-culture system, we observed that most of the multinucleated syncytia die or can be removed by a 40 µm cell strainer after prolonged culture (that is, ~6 d). To confirm this result, we mixed additional HEK293T cells that stably express BFP into the co-culture

system, reasoning that if the fused cells die and/or are being removed by the strainer, the percentage of BFP⁺ cells will increase. We mixed HEK293T-BFP cells, HEK293T-spike-GFP1–10 and HEK293T-ACE2-GFP-11 cells at a ratio of 1:1:1. The percentage of BFP⁺ cells rose from 33.8% to 60.5% on day 6, which agrees with our reasoning (Supplementary Fig. 4).

Next, we applied our reverse selection approach to carry out genome-wide CRISPR knockout screening to identify the important host factors for syncytium formation (Fig. 4a). Approximately 150 million A549-ACE2-SpCas9-GFP1–10 cells were infected with the lentiviral MiniLibCas9 sgRNA library[44] at an MOI of ~0.3 to have each of the 37,722 sgRNAs to be present in >500 unique cells on average (Extended Data Fig. 6a). On day 6 post infection, the sgRNA-infected cells were sorted out for further culture. After allowing the cells to recover for a week after sorting and expanding in numbers, these A549-ACE2-SpCas9-GFP1–10-sgRNA library cells were mixed with 293T-spike-GFP11-P2A-mCherry cells. At the same time, ~15 million A549-ACE2-SpCas9-GFP1–10-sgRNA library cells were collected before mixing. The mixed cells were passaged every 3 d. During cell passaging, a 40 μm cell strainer was used to remove the syncytia clumps to enrich the unfused cells. On day 7 post mixing, unfused A549-ACE2-SpCas9-GFP1–10-sgRNA library cells were sorted out for the late-timepoint sample. The genomic DNA of all the samples were extracted and the sgRNA region was PCR amplified for NovaSeq-based sequencing (Extended Data Fig. 6b). We performed the CRISPR screen in two biological replicates and confirmed the robustness of our library screen in identifying known essential genes (Extended Data Fig. 6c–e). Upon comparing the sgRNA abundance in unmixed A549-ACE2-SpCas9-GFP1–10-sgRNA library cells and unfused A549-ACE2-SpCas9-GFP1–10-sgRNA library cells using MAGeCK[45] and JACKS[46], we identified the known *ACE2* receptor as well as five other hits (*FCHO2*, *AP2M1*, *CAB39*, *RNF2* and *GBP6*) at arbitrary cut-offs of robust ranking aggregation (RRA) score >4 and fold change >1.5 as potential host factors that play important roles in syncytium formation (Fig. 4b and Source Data). Inhibition of known tumour-suppressors *PTEN* and *PDCD10* is expected to increase cell proliferation, thus these genes were not included in our further analysis. We performed individual validation experiments and confirmed that knockout of the five hits in A549-ACE2 receiver cells inhibited syncytium formation when mixed with the WT spike-expressing sender cells (Fig. 4c,d). Since knockout of *FCHO2* and *AP2M1* reduced syncytium formation induced by WT spike at a greater extent than the other three genes (Fig. 4c,d) and they also inhibited Omicron spike-induced syncytium formation (Extended Data Fig. 7a), *FCHO2* and *AP2M1* were thus selected for further characterization. Since A549-ACE2 cells do not express TMPRSS2, our results suggest that the fusion events are not TMPRSS2-mediated. In the absence of TMPRSS2, the spike protein can enter cells through endosomal pathways[47]. We generated *FCHO2* and *AP2M1* knockouts in Vero E6-TMPRSS2 receiver cells and found that both knockouts inhibited syncytium formation when mixed with the sender cells (Extended Data Fig. 7b). This result indicates that these two factors are also important for syncytium formation in cells that express TMPRSS2.

AP2M1 and FCHO2 are core regulators of clathrin-mediated endocytosis (CME)[48,49]. RNA-seq and gene ontology (GO) enrichment analysis on *FCHO2* and *AP2M1* knockout A549-ACE2 cells revealed the positive regulation of cell-substrate/matrix adhesion, among other processes, in both *FCHO2* and *AP2M1* knockout cells (Supplementary Fig. 5). Cell-substrate/matrix adhesion was reported to increase the force required for deforming a membrane during clathrin-coated vesicle formation and inhibit CME[50]. We further showed that genetic knockdown of clathrin heavy chain (CHC) (Fig. 4e–g) and treatment with a clathrin inhibitor Pitstop 2 (Fig. 4h) both suppressed the cell–cell fusion process. These results support the involvement of CME in spike-induced syncytium formation. We moved on to test approved drug candidates that could inhibit spike-induced syncytium formation. A previous study showed that the CME inhibitor promethazine reduces SARS-CoV-2-induced cytotoxicity in Vero E6 cells[25]. In a drug screen with more than 3,000 approved drugs, promethazine, fluvoxamine and itraconazole (ITZ) were scored to have some inhibitory effects on SARS-CoV-2-induced syncytium formation in Vero E6 cells, albeit not further characterized in the study[19]. Here we carried out a more comprehensive evaluation of the effect of five approved endocytosis inhibitors and found that treatment with the three CME inhibitors (that is, chlorpromazine (CPZ), fluvoxamine and promethazine, but not Imipramine and ITZ which both primarily affect micropinocytosis) greatly impeded syncytium formation in both A549-ACE2 and Vero E6 cells (Fig. 5a–d). The drug doses used did not affect cell viability (Supplementary Fig. 6). The decrease in syncytia observed after CME inhibition did not appear to be due to an effect on the ACE2 surface level in the receiver cells or spike's localization to cell surface for ACE2 binding (Supplementary Fig. 7a–e). Future work may examine whether it involves actin cytoskeletal rearrangement as previously described for myoblast cell–cell fusion[51].

We extended our work to validate our findings using live SARS-CoV-2 both in vitro and in vivo. We confirmed that treatment with all three CME inhibitors (CPZ, fluvoxamine and promethazine) inhibited syncytium formation in the SARS-CoV-2 D614G-infected cells (Fig. 6a,b). In addition, the three inhibitors also greatly reduced the expression levels of SARS-CoV-2 RNA-dependent RNA polymerase (RdRp) in both cell lysate and supernatant samples after virus infection (Fig. 6c and Supplementary Fig. 8a), indicating the reduced production of new viruses. Similarly, *FCHO2* and *AP2M1* knockout (Supplementary Fig. 8b), as well as *CHC* knockdown (Supplementary Fig. 8c), reduced the RdRp levels in the SARS-CoV-2-infected cells. Since *FCHO2* and *AP2M1* knockout cells were also less susceptible to the spike-pseudotyped virus (Supplementary Fig. 8d), CME inhibition exerts anti-viral effects probably via reducing viral entry, in addition to reducing syncytium formation. In our in vivo experiments (Fig. 6d), treatment with CPZ and fluvoxamine reduced the virus RdRp gene expression level (Fig. 6e), the infectious SARS-CoV-2 titre (Fig. 6f), the area of positivity of the SARS-CoV-2 nucleocapsid protein (Fig. 6g) and bronchiolar epithelium damage, alveolar congestion, infiltration and haemorrhage (Supplementary Fig. 9) in the lung tissues of the virus-infected hamsters. Also, less syncytium-like multinucleated cells were detected within the SARS-CoV-2 nucleocapsid-positively stained lung tissues in the CPZ- and fluvoxamine-treated hamsters (Fig. 6g). These results support the

**Fig. 6 | Clathrin-mediated endocytosis inhibitors impede SARS-CoV-2 live virus-induced syncytium formation. a**, GFP-split complementation system assay was applied using Vero E6-GFP1–10 and Vero E6-GFP11 cells, and Vero E6-TMPRSS2-GFP1–10 and Vero E6-TMPRSS2-GFP11 cells (**a**), mixed at a 1:1 ratio. The cells were treated with the indicated drugs and infected with SARS-CoV-2 D614G live virus. Representative images are shown. **b**, Quantification of the area of syncytium formation in **a**. Data shown are mean ± s.d. (*n* = 6 biological replicates). *P* values indicated were compared with DMSO-treated control. Statistical significance was determined using one-way ANOVA. **c**, RT–qPCR measurements of SARS-CoV-2 RdRp levels in CME inhibitor-treated Vero E6 and Vero E6-TMPRSS2 cells. The RdRp levels in both cell lysate and supernatant after SARS-CoV-2 D614G virus infection were measured. **d**–**g**, Lung tissues of CME inhibitor/ DMSO-treated, SARS-CoV-2 D614G virus-infected hamsters were collected (**d**). The RdRp level was measured by RT–qPCR (**e**). The infectious virus titres were measured by plaque assay (**f**). The presence of syncytium-like multinucleated cells was detected by immunofluorescence (IF) staining and indicated by arrows in the representative images (**g**). Lung tissues were stained by antibodies against SARS-CoV-2 nucleocapsid protein (SARS-CoV-2 N) (green) and sodium potassium ATPase (red). Nuclei were stained with DAPI (blue). Multinucleated cells were indicated by white arrows in the representative images (**i**, **ii**, **iii**). Scale bars, 200 μm (**a**) and 100 μm (**g**). In **c**, **e** and **f**, data shown are mean ± s.d.; *n* = 4 (**c**), *n* = 6 (**e**,**f**) biological replicates. *P* values indicated were compared with DMSO-treated control. Statistical significance was determined using one-way ANOVA. ****$P < 0.0001$. Schematic diagrams in **a** and **d** were created with BioRender.com.

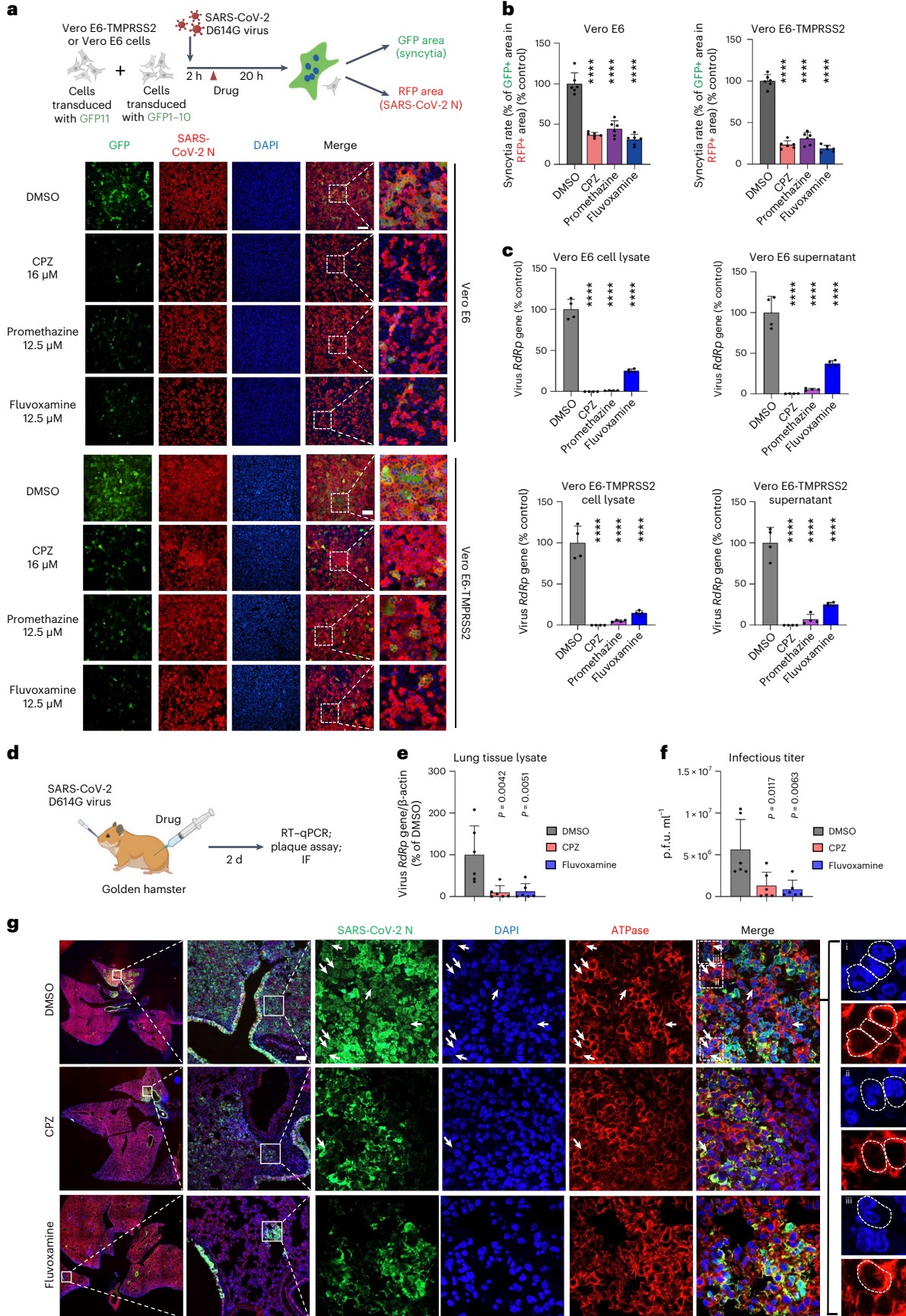

idea that the CME inhibitors reduce the viral load, spread, syncytium formation and pathogenesis in the SARS-CoV-2-infected tissues, and demonstrate the in vivo relevance of our findings.

Collectively, our results underscore the involvement of the CME machinery in driving the cell–cell fusion process. These results also validate our reciprocal high-throughput screening approach for identifying host factors in receiver cells that are pivotal for syncytium formation. Using our method could also uncover gene knockouts that may enhance syncytium formation (Extended Data Fig. 8a–c), although this is not the focus of the current study. CRISPR activation screening in receiver cells, as well as CRISPR knockout and activation screening in sender cells, could be performed to identify additional determinants.

## Discussion

Here we have established droplet microfluidics and size-exclusion selection strategies to enable two-way reciprocal screening of variant libraries in a sender–receiver cell fusion system. Our pool-based methods are scalable for quantitative assessment of cell–cell fusion, bypassing the laborious steps to individually construct and characterize genetic mutants and perturbations. Functional annotation of mutations that confer SARS-CoV-2 spike with greater syncytium-forming potential should aid evaluation of the pathological consequences of the existing and emerging viral variants. So far, previous work primarily focused on studying how mutations at the RBD region and the ectodomain of the spike protein affect ACE2 binding affinity and/or antibody escape using phage or yeast surface display[52–58]. Extending the efforts to functionally annotate the other parts of the spike protein that are not in direct contact with the ACE2 receptor and more importantly under human cell environments, we scanned the FPPR and furin cleavage site region of the spike protein and revealed the presence of syncytium-enhancing mutations at both regions. Our work indicates that single mutations at the non-RBD region including FPPR and furin cleavage-site region of the spike is sufficient in enhancing its syncytium-forming ability. Of important note, the K854H substitution indeed confers Omicron's spike with syncytium-forming potential comparable to that of the D614G strain, while sites including 683, 684, 687, 688 and 854 were predicted to have mutability scores comparable or close to that of 681 at which mutations were found in Omicron and Delta variants, and other mutations at the RBD of spike. Emergence of syncytium-promoting mutations at these sites in any future viral isolates should be scrutinized. Although DMS over the entire spike sequence is challenging given its large size to generate all the possible variants (that is, 20 amino acid residues × 1,273 positions), future work could couple our scalable screening systems established for studying syncytia in human cells with deep mutation learning[59] and high-order combinatorial mutagenesis[60] methods to functionally annotate mutations at the other regions of the spike that have high mutability as well as combinatorial mutational effects to map epistatic relationships.

To identify the cellular determinants of SARS-CoV-2 spike-induced syncytium formation, we performed a whole-genome CRISPR screening and identified two core CME regulators, FCHO2 and AP2M1, as key factors, in addition to the ACE2 receptor. Inhibition of syncytium formation upon treatment with a selective CME inhibitor Pitstop 2 further affirms the involvement of CME in this process. The CRISPR screen performed in this study was designed to specifically look for determining factors for spike-induced syncytium formation given its impact on disease severity, which differs from previous CRISPR screens[22–28] that gave an overview of host factors involved in the virus infection process and life cycle; thus, more and different hits may be identified in those screens. While we found that *ACE2* and a few other CME-related genes (*AP1G1*, *AP1B1*, *AAGAB*) were scored as hits in the previous SARS-CoV-2 virus infection-based CRISPR screens[26–28], the potent syncytium formation-modifying hits (including *FCHO2* and *AP2M1*) identified in our screen were not previously uncovered, emphasizing that different aspects of viral biology are revealed by these screening

methods. With the intention to ease COVID-19 severity via combating syncytium formation, pharmacological treatment using CME inhibitors could be an important option to consider. CPZ and fluvoxamine are widely used drugs for treating psychiatric disorders, and are also known to disrupt CME[61]. Our results showed that CME inhibitor treatment is effective in suppressing syncytium formation induced by spike. Taken together, our genetic data demonstrating the involvement of the CME machinery in driving SARS-CoV-2 spike-induced syncytium formation provide a plausible drug mechanism-of-action to support the repurposing of CPZ[62], fluvoxamine[63] and other potential CME inhibitors for alleviating COVID-19 severity in patients.

More broadly, our paired-cell profiling systems can be applied to study a variety of pathological, physiological and even synthetic conditions that are relevant to biomedical applications. Apart from SARS-CoV-2, a broad spectrum of viruses including human immunodeficiency virus[64], *Herpesviridae*[65], respiratory syncytial virus[66], as well as other *Coronaviridae*, induces syncytium formation. Fusion was also reported between tumour and normal somatic cells to form hybrid cells that are more malignant and exhibit increased metastatic behaviour[67]. Defining the common and unique determinants for each type of virus-induced syncytium formation and revealing cellular regulators for tumour–normal somatic cell fusion could help combat the disease pathogeneses. Fusion of specific cell types forms multinucleated cells including syncytiotrophoblasts, myotubes and osteoclasts to aid their physiological functions in controlling maternal–fetal material exchange at the placenta, coordinating muscle contraction and facilitating bone resorption, respectively[68]. Artificial fusion of B cells and myeloma cells produces hybridoma as the workhorse for antibody production[69]. Fusion of human embryonic stem cells with somatic cells reprograms them to pluripotency as cell sources for regenerative medicine[70]. Fusion of dendritic cells with tumour cells produces hybrids that express the tumour-associated antigens and is being tested as potential cancer immunotherapy reagents[67]. The application of high-throughput profiling systems together with CRISPR screening will aid the understanding of the mechanisms by which these different cells fuse and their engineering to achieve greater fusion efficiencies for real-life applications.

## Methods

### Plasmid construction

All the constructs used in this study were generated with standard cloning strategies, including PCR, overlapping PCR, oligo annealing, digestion and ligation. Primers were purchased from Genewiz. The plasmid sequence was verified by Sanger sequencing. The pCAG-spike(D614G)-GFP11-mCherry plasmid was modified from Addgene plasmid 158761. Briefly, GFP11 and mCherry sequences were amplified from Addgene plasmid 68716 and 79124, respectively, and then cloned into the downstream of spike (D614G), resulting in the pCAG-spike-GFP11-P2A-mCherry vector. The pCAG-spike(Omicron)-GFP11-mCherry plasmid was modified from pCAG-spike(D614G)-GFP11-mCherry plasmid, that is, the spike (Omicron) cassette was amplified from Addgene plasmid 179907 to replace spike(D614G). For the pCMV-BSD-GFP1–10 plasmid, BSD and GFP1–10 sequences were amplified from Addgene plasmid 68761 and 70224, respectively, and then cloned into the pFUGW lentiviral vector backbone, resulting in the Lenti-pGMV-BSD-p2A-GFP1–10 vector. To generate lentiviral vectors expressing an sgRNA that targets a specific gene, oligo pairs for the target sequence were annealed and cloned into the BbsI restriction sites in Addgene plasmid 67989. To generate lentiviral vectors expressing shRNAs, oligo pairs for *CHC* were annealed and cloned into EcoRI/AgeI restriction sites in Addgene plasmid 10879. The plasmids, sgRNA sequences and primers used are listed in Supplementary Tables 1 and 2.

### Virus and biosafety

The SARS-CoV-2 D614G virus strain was isolated from laboratory-confirmed COVID-19 patients in Hong Kong. The virus was cultured using Vero E6-TMPRSS2 cells and titred by plaque assays.

All experiments with SARS-CoV-2 were performed according to the approved standard operating procedures of the Biosafety Level 3 facility at the Department of Microbiology, School of Clinical Medicine, The University of Hong Kong.

## Cell culture and generation of cell line

HEK293T (ATCC) and Vero E6 cells were grown in DMEM medium containing 10% FBS and 1x penicillin-streptomycin. A549-ACE2 cells and Vero E6-TMPRSS2 cells were cultured with DMEM medium (with 10% FBS and 1x penicillin-streptomycin) supplemented with $0.5\,\mu g\,ml^{-1}$ puromycin (ant-pr-1, InvivoGen) or $1\,mg\,ml^{-1}$ G418 (ant-gn-1, InvivoGen), respectively. All cells were incubated at 37 °C with 5% $CO_2$. A549-ACE2-Cas9-GFP1–10 and Vero E6-TMPRSS2-Cas9-GFP1–10 cells were generated by co-transducing pAWp30 (Addgene, 73857) and Lenti-GFP1–10-blast (pBW93) into the A549-ACE2 and Vero E6-TMPRSS2 cells, respectively, followed by selection with $500\,\mu g\,ml^{-1}$ zeocin (R25001, Life Technologies) and $10\,\mu g\,ml^{-1}$ blasticidin (ant-bl-05, InvivoGen) for -10 d for stable Cas9- and GFP1–10-integrated cells. HEK293T-spike-GFP11-P2A-mCherry cells were generated by transducing Lenti-spike-GFP11-P2A-mCherry into HEK293T cells. Single mCherry-positive colonies were expanded. The monoclonal cell line with high fusogenicity was screened by mixing with A549-ACE2 cells. A549-ACE2-GFP1–10-BFP cells were generated by transducing Lenti-BFP and lenti-GFP1–10-blast into A549-ACE2 cells, followed by selection with $10\,\mu g\,ml^{-1}$ blasticidin for -10 d and then screening for cells with homogenous BFP expression by FACS.

## Droplet microfluidic system

The microfluidic device was fabricated using a typical soft lithography replica moulding technique. First, two channel moulds designed for two layers of channels were fabricated on two silicon wafers (N100) with SU-8 photoresist (2025, MicroChem) using maskless lithography (SF-100 Xcel, Intelligent Micro Patterning). The bottom layer for the multiple droplet generator was fabricated with a height of $60\,\mu m$, while the height of the top layer for droplet convergence was $70\,\mu m$. Then the PDMS pre-polymer base (Sylgard 184, Dow Corning) was crosslinked with the curing agent using the weight ratio of 10:1. After sufficient mixing by a conditioning mixer (AR-100, THINKY), the mixture was poured onto the channel moulds and cured at 65 °C for 4 h. Subsequently, the two layers of PDMS channels were peeled off from the mould. After inlet and outlet punching, the bottom layer was bonded to a glass substrate (ISOLAB) through oxygen plasma treatment (PDC-002). Then, the top layer was bonded to the bottom layer after alignment under the microscope, followed by heating at 90 °C for 12 h. The whole channel was treated with a hydrophobic agent (Aquapel, PPG) to guarantee stable droplet generation. For sample preparation, cells were trypsinized and resuspended in DMEM medium supplemented with 10% FBS, 1x penicillin-streptomycin as the inner phase, and 18% OptiPrep (92339-11-2, Merck) was added to prevent cell sedimentation. To encapsulate cells in droplets, a fluorinated oil (HFE 7500, 3M) supplemented with 0.5% (w/w) of surfactant (RAN Biotechnologies) was used as the continuous phase. The flow rates of the cell solution and the oil were precisely controlled at $6,000\,\mu l\,h^{-1}$ and $9,000\,\mu l\,h^{-1}$, respectively, by syringe pumps (neMESYS 290N, CETONI). The droplets with a size of -75 μm can thus be generated at a frequency of -12 kHz.

## DMS library construction

To construct the Lenti-pCAG-spike-GFP11-mCherry storage vector, we removed the fragment that encoded residues S673 to K854 on the spike and introduced two Esp3I restriction sites by overlapping PCR, resulting in the Lenti-pCAG-spike$_{(1–672)}$-Esp3I-Esp3I-spike$_{(855–1273)}$-GFP11-mCherry vector. We designed mutagenic primers containing binding sequences, degenerate NNS codons that tile across FPPR (836-854) or Furin (673-691) sites, and Esp3I digestion site flanking the end. Synonymous mutations were introduced to the protein-coding

sequence containing Esp3I restriction sites in the construct. Nineteen NNS primers for each site were pooled at an equal molar ratio, resulting in an FPPR-NNS primer pool and a Furin-NNS primer pool. The mutagenesis FPPR or Furin insert was amplified from Addgene plasmid 158761 by using KARA HiFi HotStart ReadyMix with the FPPR-NNS primer pool/S_DMS-fs1 or S_DMSrs/Furin-NNS primer pool (Supplementary Table 2), respectively, and then cloned into the Esp3I–Esp3I site of the storage vector, resulting in the FPPR and Furin DMS libraries. Overall, the FPPR and the Furin DMS libraries each had 380 variants at the protein level (that is, 608 variants at the nucleotide level).

## Spike DMS library screening

Around $1.6 \times 10^6$ HEK293T WT cells were transduced with the lentivirus-packaged FPPR or Furin DMS library at an MOI of 0.3 to achieve >500-fold representation for each variant. On day 6 post infection, mCherry-positive cells were sorted out and expanded for further screening. Around $6 \times 10^5$ 293T spike DMS cells were collected before mixing. For the droplet microfluidic-based strategy, the inner phase cell solution was pre-mixed A549-ACE2-GFP1–10-BFP cells and HEK293T-spike DMS library cells at concentrations of $6.6 \times 10^6\,ml^{-1}$ and $1.6 \times 10^6\,ml^{-1}$, respectively. To achieve >500-fold coverage for each spike variant, -8 million HEK293T-spike DMS library cells were used for each replicate to ensure that enough droplets contained the paired cells. The generated droplets were collected in a T25 flask and incubated for -24 h at 37 °C. Furthermore, after droplet breakage, cells were collected and fixed, and GFP$^+$ and GFP$^-$ cells within the BFP$^+$/RFP$^+$ cell population were then sorted out. For the size-exclusion selection-based strategy, A549-ACE2-GFP1–10-BFP cells and HEK293T-spike DMS library cells were mixed and co-cultured in 12-well plates at concentrations of $1 \times 10^6$ and $5 \times 10^5$ cells per well, respectively. Around 24 or 48 h post cell mixing, the cell mixture was trypsinized and passed through a 70 μm strainer, and large syncytia that remained on the strainer were collected. Then the cells passing through the strainer were subjected to cell sorting, that is, small syncytia (GFP$^+$ cells within the BFP$^+$/RFP$^+$ cell population) were sorted out. To achieve -500-fold coverage for each spike variant, -6 million HEK293T-spike DMS library cells were used for each replicate.

## Genome-wide CRISPR screening

The MinLibCas9 library (Addgene, 164896) was used for the CRISPR-mediated gene knockout screen. A549-ACE2-Cas9-GFP1–10 cells ($1.5 \times 10^8$) were transduced with lentivirus-packaged Min-LibCas9 sgRNA library at an MOI of 0.3 to achieve -500-fold representation for each sgRNA. On day 6 post infection, BFP$^+$ cells were sorted out for further culture. On day 14 post infection, -35 million A549-ACE2-Cas9-GFP1–10-sgRNA cells were mixed with HEK293T-spike-GFP11-mCherry cells at a 1:2 ratio. At the same time, -15 million A549-ACE2-Cas9-GFP1–10-sgRNA cells were collected before mixing. Then the cell mixture was passaged every 2 d. During each passaging, cells were passed through a 40 μm cell strainer to remove the syncytium clumps to enrich the unfused cells. On day 7 post cell mixing, unfused A549-ACE2-Cas9-GFP1–10-sgRNA cells were collected as the late-timepoint sample.

## Sample preparation for NovaSeq-based sequencing

For the fresh cell sample, the genomic DNA was isolated by using DNeasy blood and tissue kits (69504, Qiagen) according to manufacturer instructions. For the fixed cell sample, cells were resuspended in 180 μl buffer ATL and 20 μl proteinase K was added. Then, cells were incubated at 56 °C for 1 h and at 90 °C for another 1 h, followed by the addition of RNase A, buffer AL and 96%–100% ethanol. The mixture was passed through a DNeasy Mini spin column (Qiagen), washed and eluted according to manufacturer protocol. A two-step PCR protocol was used to amplify the sgRNA region or DMS region for Illumina sequencing via a previously published protocol[71]. Briefly, in the first

PCR step, the integrated region containing the sgRNA sequences or the DMS sequences of the FPPR or the Furin site was amplified by using KARA HiFi HotStart ReadyMix. Primers 5′- ACACTCTTTCCCTA CACGACGCTCTTCCGATCTCTTGTGGAAAGGACGAAACA-3′ and 5′- GTG ACTGGAGTTCAGACGTGTGCTCTTCCGATCTCTAAAGCGCA TGCTCCAGAC-3′ were used for amplifying the sgRNA region. Primers 5′- CACGACGCTCTTCCGATCTCGGACCCCAGTAAACCCTC-3′ or 5′- CACGACGCTCTTCCGATCTCATGTGAACAATTCATACGAATGTG-3′ and 5′-CAGACGTGTGCTCTTCCGATCTCCGAATGTCCATCCAGACGTT-3′ or 5′-CAGACGTGTGCTCTTCCGATCTATTTGTTGGGATGGCAATGGAG-3′ were used for amplifying the DMS region of the FPPR or the furin cleavage site, respectively. To ensure sufficient coverage, all extracted genomic DNA was used in the first step of PCR, where 800 ng gDNA was added per 50 µl PCR reaction. PCR products were purified by using Agencourt AMoure XP beads (A63881, Beckman Coulter Genomics). Then, the second PCR of 13 cycles was performed to add Illumina adapters and sequencing index to the amplicons using KARA HiFi HotStart ReadyMix. The final PCR products were purified with Agencourt AMoure XP beads. The concentrations of different libraries were quantified by real-time PCR using TB Green Premix Ex *Taq* (Tli RNaseH Plus) (RR420A, Kapa Biosystems) with primers 5′- AATGATACGGCGACCACCGA-3′ and 5′- CAAGCAGAAGACGGCATACGA-3′, then the libraries were pooled for Novaseq.

### Screen data analysis

For the genome-wide CRISPR screen, two previously published methods, MAGeCK[45] and JACKS[46], were used to rank genes on the basis of RRA scores and gene essentiality score, respectively. sgRNA abundances were assessed in 'before mixing' samples vs unfused samples. The top screen hits were determined on the basis of MAGeCK RRA and JACKS scores. Kyoto Encyclopaedia of Genes and Genomes (KEGG) and Gene Ontology (GO) enrichment analyses were performed on genes that exhibited depletion between the plasmid library and the 'before mixing' samples. For the DMS screen, the fold change of each variant was calculated by comparing its relative abundance before and after selection according to the DiMsum method[72].

### DMS screen hit validation

HEK293T WT cells were seeded in 24-well plates at a confluence of ~70% and transfected with 500 ng plasmid expressing only one DMS variant using FuGene HD transfection reagent (E2312, Promega) according to manufacturer protocol. After 24 h, HEK293T-spike variant cells were mixed with A549-ACE2-GFP1–10-BFP cells at a ratio of 1:10 in 24-well plates. After 24 h, microscopy images of the co-cultured cells were taken using the GE IN Cell Analyzer 6500HS high-throughput imaging system. Cell fusion was quantified by measuring the area of syncytia (GFP+ area) using the GE IN Carta image analysis software (v.2.x) and normalized to the WT variant.

### Genome-wide knockout screen hit validation

A549-ACE2-Cas9-GFP1–10 cells or Vero E6-TMPRSS2-Cas9-GFP1–10 cells were seeded in 12-well plates at a confluence of ~30% and transduced with lentivirus expressing sgRNA with 10 µg ml⁻¹ polybrene (TR-1003-G, Sigma). For A549-ACE2 cells, at day 8 post infection, ~2.5 × 10⁵ A549-ACE2-Cas9-GFP1–10-sgRNA cells were mixed with 3 × 10⁴ HEK293T-spike(D614G)-GFP11-mCherry cells or HEK293T-spike(Omicron)-GFP11-mCherry cells in a 24-well plate. After 24 h, microscopy images of the co-culture cells were taken. Cell fusion was quantified by measuring the total area of all the syncytia (GFP+ area) and the average size of syncytia, normalized to the control sample. For Vero E6-TMPRSS2 cells, at day 8 post infection, ~4 × 10⁵ HEK293T-spike(D614G)-GFP11-mCherry cells or HEK293T-spike(Omicron)-GFP11-mCherry cells were mixed with 1 × 10⁴ or 3 × 10⁴ Vero E6-TMPRSS2-Cas9-GFP1–10-sgRNA cells in a 24-well plate. After 24 h, the cell mixture was trypsinized and passed through a 70 µm cell strainer for FACS assay.

Cell fusion was quantified using GFP+/BFP+ and normalized to the control sample.

### Lentivirus production

For sgRNA library or spike DMS library packaging, HEK293T cells were seeded at ~80% confluency in a 15-cm dish. Library plasmid (9 µg), 9 µg of pCMV-VSV-G vector and 18 µg of pCMV-dR8.2-dvpr vector were transfected using 72 µl polyethylenimine. The medium was changed ~12 h post transfection. The lentivirus supernatants were collected three times at 48, 72 and 96 h post transfection, then combined and filtered with a 0.45-µm polyethersulfone membrane. The virus was concentrated using an Amicon Ultra-15 centrifugal unit, aliquoted and stored at −80 °C. For individual sgRNA or shRNA packaging, HEK293T cells were seeded at ~80% confluency in 6-well plates. sgRNA plasmid (500 ng), 500 ng of pCMV-VSV-G vector and 1 µg of pCMV-dR8.2-dvpr vector were transfected using 4 µl polyethylenimine. The medium was changed ~16 h post transfection. Lentivirus supernatants were collected at 48 and 72 h post transfection, then combined and filtered with a 0.45-µm polyethersulfone membrane.

### Flow cytometry and cell sorting

BD FACSAria Fusion and BD Influx cell sorters were used for cell sorting. Agilent NovoCyte Advanteon BVYG and ACEA NovoCyte Quanteon analysers were used for analysis. Flowjo (v.10.8.1) was used to analyse data generated from flow cytometry experiments. For cell sorting of samples infected with the sgRNA or spike DMS libraries, cells were trypsinized and resuspended in sorting buffer (PBS with 2% FBS and 2x penicillin-streptomycin) and then collected in collection buffer (DMEM medium with 20% FBS and 2x penicillin-streptomycin). For fixed-cells sorting in spike DMS library, droplets were first broken using 1H,1H,2H,2H-perfluoro-1-octanol (PFO) (370533, Sigma). Droplets (2 ml) were aliquoted into 15 ml falcon tubes, with 1 ml of DMEM medium added on top of the oil phase. PFO (600 µl) was added, briefly mixed and centrifuged at 300 g for 30 s. The oil phase at the bottom was removed. After washing twice with PBS, cells were fixed with 4% PFA for 20 min at room temperature. Then cells were washed twice and resuspended in PBS. To prevent cell sedimentation, 10% OptiPrep was added during cell sorting.

### Spike and ACE2 surface staining

Cells were detached using 0.5 mM EDTA, washed once with 5% FBS in PBS, then blocked with 10% FBS for 1 h at 4 °C. Next, cells were incubated with primary antibody in blocking solution for 1 h at 4 °C. Goat anti-ACE2 (1:100) (AF933, R&D) was used to label surface ACE2, and rabbit anti-spike S2 (944−1214 aa) (1:500) (28867-1-AP, Proteintech) was used to label surface spike. After washing twice with PBS, cells were incubated with donkey anti-goat IgG(H + L) cross-adsorbed secondary antibodies conjugated with AF568 (1:1,000) (A-11057, Thermo Fisher) or goat anti-rabbit IgG(H + L) cross-adsorbed secondary antibodies conjugated with AF488 (1:1,000) (A-11008, Thermo Fisher) for 1 h at 4 °C in the dark. After washing twice with PBS, cells were resuspended in PBS for FACS or cell sorting. To evaluate S1 subunit cleavage, mouse anti-spike S1 subunit AF488-conjugated antibody (1:200) (FAB105403G, R&D) was used to label surface S1 subunit for 2 h at 4 °C in the dark. The cells were resuspended in PBS for FACS after being washed with PBS twice. The level of S1 subunit cleavage was calculated using (1 − Proportion of cells with S1-positive staining) × 100%.

### ACE2 binding assay

Cells transduced with the FPPR DMS library were detached using 0.5 mM EDTA, washed once with 5% FBS in PBS, then incubated with 100 nM biotinylated human ACE2 protein (10108-H08H-B, SinoBiological) for 2 h at 4 °C. After washing twice with PBS, cells were incubated with streptavidin conjugated with AF405 (S32351, Thermo Fisher) for

1 h at 4 °C in the dark. After washing twice with PBS, cells were resuspended in PBS for FACS or cell sorting.

## Cell viability assay

A549-ACE2-GFP1–10-BFP and Vero E6 cells were seeded at a concentration of 5,500 cells per well in 100 μl of medium in 96-well plates and incubated overnight. For CPZ (S5749, Selleckchem) and ITZ (S2476, Selleckchem) drug treatments, cells were incubated with various amounts (final concentration: 0 μM, 2 μM, 4 μM, 8 μM, 16 μM and 32 μM) at 37 °C overnight. For promethazine HCl (S4293, Selleckchem) and fluvoxamine (S1336, Selleckchem) drug treatments, cells were incubated with virus amounts (final concentration: 0 μM, 3.125 μM, 6.25 μM, 12.5 μM, 25 μM, 50 μM and 100 μM) at 37 °C overnight. For Pitstop2 (HY-115604, Biosystem), cells were incubated with various amounts (final concentration: 0 μM, 2 μM, 4 μM, 8 μM, 16 μM and 32 μM) at 37 °C for 20 min. Then, cell viability was measured using the 2,3-bis-(2-methoxy-4-nitro-5-sulfophenyl)-2H-tetrazolium-5-carboxanilide (XTT) assay. Briefly, drug-containing medium was replaced with 100 μl of detection solution (100 μl of 1x XTT solution with 0.1% volume ratio of 3 mg ml$^{-1}$ phenazine methosulfate) and incubated at 37 °C for 3 h. Then the absorbance was measured at 470 nm using the Varioskan LUX Multimode microplate reader.

## Drug response study on syncytium formation

For the cell–cell fusion assay, the sender cells used were HEK293T cells transfected with plasmid that expresses SARS-CoV-2 spike, GFP11 and mCherry, while the receiver cells used were A549-ACE2-GFP1–10 cells or Vero E6-GFP1–10 cells. HEK293T cells were seeded at a confluence of ~70% and transfected with D614G spike-GFP11-mCherry vector using FuGene HD transfection reagent according to manufacturer protocol. After 24 h, ~3 × 10$^4$ HEK293T-spike-GFP11-mCherry cells were mixed with 2.5 × 10$^5$ A549-ACE2-Cas9-GFP1–10 cells or 1.8 × 10$^5$ Vero E6-GFP1–10 cells in 24-well plates. At 3 h after seeding, drugs (CPZ, fluvoxamine, ITZ and promethazine) or dimethyl sulfoxide (DMSO) control were added at the indicated concentrations. After 24 h, images of the cell mixture were taken. Syncytia were quantified by measuring the total area of all the syncytia (GFP$^+$ area) and the average size of syncytia, and then normalizing to the control sample. For Pitstop 2, 3 × 10$^4$ A549-ACE2-GFP1–10 cells were incubated with various amounts of Pitstop 2 (0 μM, 2 μM, 4 μM, 8 μM, 16 μM and 32 μM) in FBS-free medium at 37 °C for 20 min. After incubation, drug-containing medium was replaced with fresh medium, and A549-ACE2-GFP1–10 cells were mixed with 4 × 10$^5$ 293T-spike-GFP11-mCherry cells in 24-well plates. After 24 h, images of the cell mixture were taken. Syncytia were quantified by measuring the total area of all the syncytia (GFP+ area) and the average size of syncytia, and then normalizing to the control sample. For cell fusion inhibition assay with authentic SARS-CoV-2, Vero E6-GFP11 plus Vero E6-GFP1–10 cells and Vero E6-TMPRSS2-GFP11 plus Vero E6-TMPRSS2-GFP1–10 cells (6 × 10$^4$ cells per well mixed at a 1:1 ratio) in chamber slide (PEZGS0816; MILLIPORE) were challenged with SARS-CoV-2 D614G at an MOI of 0.5 and 0.025, respectively. After 2 h of incubation at 37 °C, cells were washed once with PBS and cultured in 1% FBS DMEM with drugs (CPZ 16 μM, promethazine 12.5 μM, fluvoxamine 12.5 μM) for 24 h. Then, the cells were washed once with PBS and fixed with 4% paraformaldehyde for 30 min at room temperature. After fixation, cells were permeabilized in 0.1% Triton X-100 and stained with in-house anti-SARS-CoV-2 nucleocapsid (N) antibody (1:3,000) and DAPI. Syncytia rate was quantified using GFP$^+$ area (that is, cell fusion area)/RFP$^+$ area (that is, virus-infected area).

## Pseudovirus production and infectivity test

HEK293T cells were seeded at ~80% confluency in a 15-cm dish. Lentiviral backbone plasmid (9 μg) encoding EF1α-EGFP (Addgene, 138152), 15 μg Omicron spike expression plasmid (Addgene, 179907) and 12 μg of pCMV-dR8.2-dvpr vector were transfected using 72 μl polyethylenimine. The medium was changed at ~12 h post transfection. Lentivirus-containing supernatants were collected at 48, 72 and 96 h post transfection, then combined and passed through a 0.45-μm polyethersulfone membrane. The lentiviral supernatants were concentrated from ~45 ml to 1 ml using lentivirus precipitation solution (VC100, ALSTEM) according to manufacturer instructions. At day 10 post sgRNA infection (that is, sgRNA-infected cells are BFP$^+$), ~1 × 10$^5$ A549-ACE2-Cas9-GFP1–10-sgRNA cells or Vero E6-TMPRSS2-Cas9-GFP1–10-sgRNA cells were seeded in 48-well plates and transduced with 80 μl of the concentrated pseudovirus. At day 5 post infection, infection rate was quantified by FACS. The infectivity was quantified using GFP$^+$/BFP$^+$ and normalized to cells transduced with safe harbour-targeting sgRNA.

## Western blot

For western blot analysis, cells were collected and lysed with RIPA lysis buffer containing 1x protease inhibitor cocktail. Equal amounts of extracted protein were separated by 10% SDS–PAGE gel and then transferred onto a polyvinylidene fluoride membrane. CHC and GAPDH were probed using rabbit anti-clathrin heavy chain antibody (ab21679, abcam) (1:900) and rabbit anti-GAPDH antibody (2118S, Cell Signaling) (1:5,000), respectively. Anti-rabbit IgG, HRP-linked secondary antibodies (1:10,000) (7074, Cell Signaling) were used, and enhanced chemiluminescence reagents were used for imaging (1705062, Bio Rad).

## RNA extraction and RT–qPCR

Total messenger RNA from cell lysate, supernatant and hamster lung tissue samples were extracted using the RNeasy mini kit (74106, QIAGEN) according to manufacturer protocol. QuantiNova Probe RT–PCR kit (208354, QIAGEN) was used to quantify the expression of RdRp. The QuantiNova SYBR Green RT–PCR kit (208154, QIAGEN) was used to quantify the expression of β-actin and GAPDH, which were used as internal controls for normalization. All primer sequences used are listed in Supplementary Table 2.

## RNA-seq

RNA-seq was performed at the Centre for PanorOmic Science in the LKS Faculty of Medicine, HKU. Briefly, A549-ACE2-Cas9 cells were infected with AP2M1_sgRNA, FCHO2_sgRNA or safe harbor_sgRNA. On day 9 post infection, total mRNA was extracted using TaKaRa MiniBEST Universal RNA extraction kit (9767, TaKaRa) following manufacturer protocol. The complementary DNA library was prepared using KAPA mRNA HyperPrep kit and sequenced on an Illumina NovaSeq 6000 system. Three replicates were sampled for each group. Transcripts abundance was quantified using Kallisto[73]. Then, gene abundance was quantified using tximport[74]. DESeq2 was used to identify differentially expressed genes (DEGs)[75]. Genes with fold change >1.2 or <0.8 and $P_{adj}$ < 0.05 were defined as DEGs (Supplementary Data). GO enrichment analysis was performed on DEGs identified from either *AP2M1* KO or *FCHO2* KO samples using the R (v.2021.09.2 + 382) package clusterProfiler (v.4.4.4).

## Immunofluorescence and histology staining

For cultured cells, the SARS-CoV-2-infected cells were fixed in 10% formalin for 30 min. After fixation, cells were washed twice with PBS and permeabilized with 0.1% Triton X-100 at 4 °C for 10 min. Then, cells were washed twice with PBS, followed by blocking with 2% BSA at room temperature for 1 h. Next, the cells were incubated with in-house rabbit anti-SARS-CoV-2 N antibody (1:3,000) overnight at 4 °C. After washing with PBS three times, cells were incubated with goat anti-rabbit secondary antibody (1:1,000; A-11008, Thermo Fisher) for 1 h at room temperature. DAPI was used for nuclear staining. Images were acquired using a Nikon Ti2-E widefield microscope. For hamster lung tissues, the SARS-CoV-2-infected hamster lungs were collected and fixed in 10% formalin for 24 h. Immunofluorescence staining was

performed following a previously published method[76]. The in-house guinea pig anti-SARS-CoV-2 N antibody (1:1,000) was used to identify SARS-CoV-2 and the rabbit anti-sodium potassium ATPase antibody (1:500; ab76020, Abcam) was used to detect cell membrane. DAPI was used to stain nuclei. The following secondary antibodies were used: goat anti-guinea pig-AF488 (1:1,000; A-11073, Thermo Fisher) and goat anti-rabbit-AF568 (1:1,000; A-11011, Thermo Fisher). Images were acquired using an LSM900 inverted confocal microscope. For hematoxylin and eosin staining, hamster lung section samples were dewaxed and stained with Gill's hematoxylin and eosin-Y following a previously published method[76]. Images were acquired using an Olympus BX53 light microscope.

### Golden hamster model
The animal study was approved by the Committee on the Use of Live Animals in Teaching and Research of The University of Hong Kong, and the experiments conducted complied with all relevant ethical regulations. Golden Syrian hamsters (aged 4–6 weeks, male) were obtained from the Chinese University of Hong Kong Laboratory Animal Service Centre through the HKU Centre for Comparative Medicine Research. The drugs were dissolved in DMSO and diluted in PBS. Hamsters were pretreated with one dose of chlorpromazine (10 mg kg$^{-1}$), fluvoxamine (20 mg kg$^{-1}$) or DMSO intraperitoneally 24 h before virus inoculation. Then, the hamsters were intranasally challenged with $5 \times 10^3$ plaque-forming units (p.f.u.) SARS-CoV-2 D614G prediluted in 50 µl PBS under ketamine (100 mg kg$^{-1}$) and xylazine (10 mg kg$^{-1}$) anaesthesia. The infected hamsters were treated with chlorpromazine, fluvoxamine or DMSO at 3 h and 24 h post infection. The body weight of the hamsters was monitored daily and no significant change was observed. Hamsters were killed on day 2 after infection. Lung tissues from infected hamsters were collected for subsequent immunofluorescence staining, RT–qPCR analysis or plaque assay as previously described[76].

### Plaque assay
The plaque assays were conducted as previously described[76]. In brief, infected hamster lung tissues were homogenized in DMEM using Tissue Lyzer II and supernatants were collected and serially diluted 10-fold. Vero E6-TMPRSS2 cells were seeded in 12-well plates at day −1. After overnight culture, cells were inoculated with the diluted supernatants for 2 h at 37 °C, washed with PBS three times and covered with 2% agarose/PBS mixed with DMEM/2%FBS at a 1:1 ratio. The cells were incubated at 37 °C for 48 h and fixed with 4% paraformaldehyde. The fixed samples were stained with 0.5% crystal violet in 25% ethanol/distilled water to visualize plaque formation.

### Mutability prediction
The mutability scores of the direct coupling analysis (DCA) and independent (IND) models of the FPPR and Furin cleavage sites were calculated using the code constructed in ref. 77 (https://github.com/juan-rodriguez-rivas/covmut). Covmut_proteome.py was run using Uniref100 as the 'distant' database and a representative version of the GISAID spike variant amino acid sequence database updated to 20 September 2022 as the 'close' database. To construct the 'close' database, CD-HIT[78] was used to cluster the sequences of GISAID spike variants at 90% identity and generated 34,182 representative sequences. The resultant mutability scores were compared with the fold change (FC) values obtained from our spike DMS library profiling. The mutability scores of the DCA and IND models of RBD were collected from https://github.com/GiancarloCroce/DCA_SARS-CoV-2 (ref. 77)

### Molecular modelling
The SARS-CoV-2 spike single amino acid substitutions on the FPPR were introduced using the mutagenesis wizard programme of PyMol. PyMol was used for atom–atom distance measurement and visualization of the protein model.

### Statistical analysis
Statistical analyses were performed using GraphPad Prism 9 software. All data are shown as mean ± s.d. Statistical significance of differences between more than two groups was calculated using one-way analysis of variance (ANOVA). The number of biological replicates are listed in the figure legends.

### Reporting summary
Further information on research design is available in the Nature Portfolio Reporting Summary linked to this article.

## Data availability
The main data supporting the results in this study are available within the paper and its Supplementary Information. The molecular structures of the spike proteins of the SARS-CoV-2 variants are available from the Protein Data Bank, with accession codes 6XR8, 7KRQ and 7TO4. The raw and analysed datasets generated during the study are available for research purposes from the corresponding authors on reasonable request. Source data for the figures are provided with this paper.

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

## Acknowledgements

We thank members of the Wong, Shum and Chu labs for helpful discussions; and the Centre for PanorOmic Sciences at LKS Faculty of Medicine, the University of Hong Kong, for providing support on the next-generation sequencing experiments, flow-cytometry analysis and cell sorting. Some of the figures were drawn using Biorender.com. This work was supported by the National Natural Science Foundation of China Excellent Young Scientists Fund (32022089) (to A.S.L.W.), the Centre for Oncology and Immunology Limited (to A.S.L.W.) and the Advanced Biomedical Instrumentation Centre (to H.C.S.) under the Health@InnoHK Initiative funded by the Innovation and Technology Commission, the Government of Hong Kong SAR, China, and Collaborative Research Fund C7103-22G, the Research Grants Council of Hong Kong SAR, China (to H.C.).

## Author contributions

G.C.G.C., C.W.F.C., B.W., L.N., X.H., H.C., H.C.S. and A.S.L.W. conceived the work. C.W.F.C., B.W., L.N., X.H., T.M., C.L., H.C., G.C.G.C., H.C.S. and A.S.L.W. designed and performed the experiments, and interpreted and analysed the data. C.W.F.C., B.W. and H.Y.C. performed the computational analyses on next-generation sequencing data. G.C.G.C., H.C., H.C.S. and A.S.L.W. supervised the study. C.W.F.C., B.W., G.C.G.C. and A.S.L.W. wrote the paper, with inputs from all authors.

## Competing interests

A.S.L.W., G.C.G.C., H.C.S., C.W.F.C., B.W. and L.N. have filed an US provisional patent application (63/481,830) based on this work. The other authors declare no competing interests.

## Additional information

**Extended data** is available for this paper at https://doi.org/10.1038/s41551-023-01140-z.

**Correspondence and requests for materials** should be addressed to Hin Chu, Gigi C. G. Choi, Ho Cheung Shum or Alan S. L. Wong.

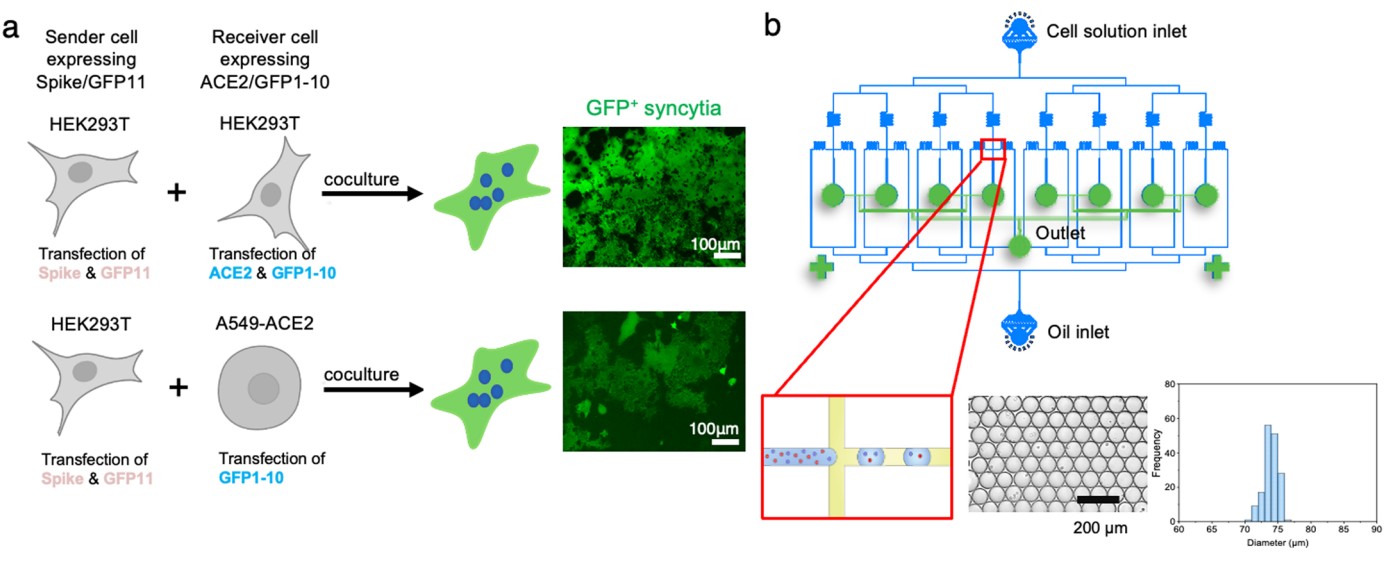

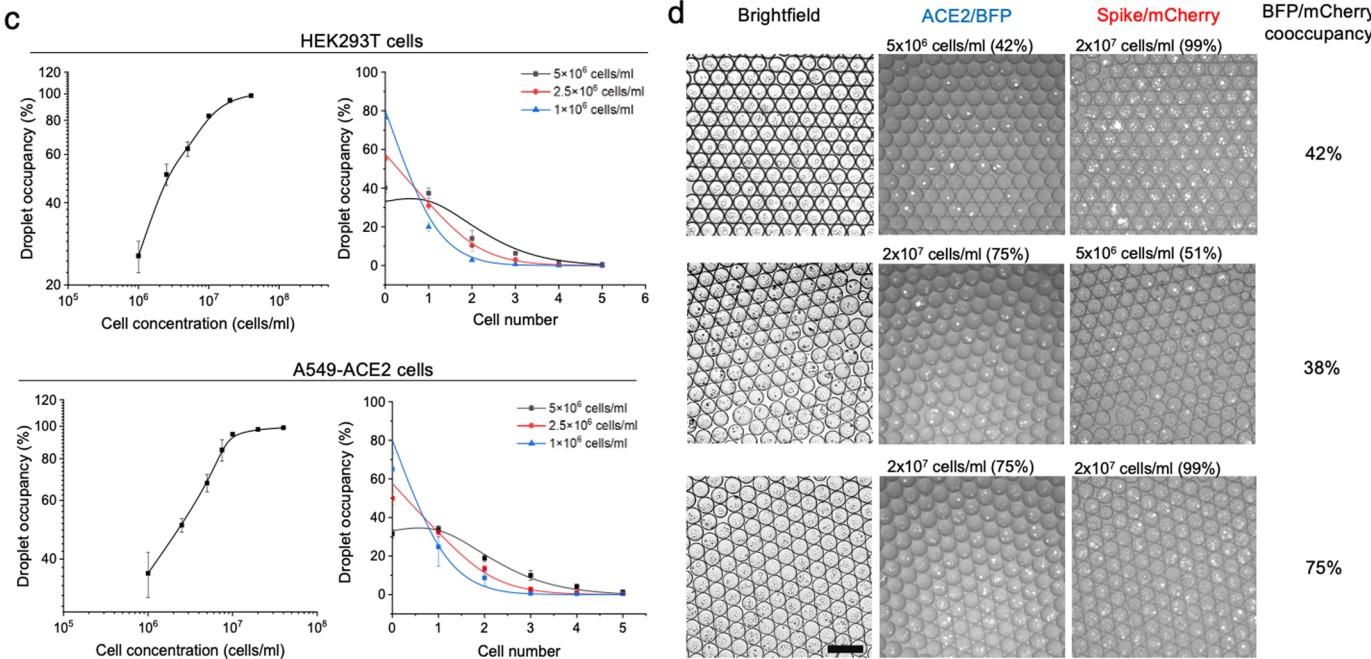

**Extended Data Fig. 1 | Establishment of systems and conditions for sender-cell and receiver-cell encapsulation in droplets. a)** Visualization of cell–cell fusion using GFP-split complementation system assay. GFP11 in Spike-expressing sender cells and GFP1-10 in ACE2-expressing receiver cells are nonfluorescent by themselves, while coculture and fusion of the sender and receiver cells resulted in reconstituted GFP that becomes fluorescent upon complementation. Created with BioRender.com **b)** Design of microfluidics device for generation of monodispersed droplets with human cells. Conditions used: continuous phase (Qc): HFE with 0.5% (w/w) of surfactant; dispersed phase (Qd): HEK293T cells ($2.5 \times 10^6$ cells/ml) in culture medium (18% OptiPrep); flow rate: Qc: Qd = 9,000: 6,000 µL/hour. **c)** Droplet occupancy increased with the sender/receiver cell concentration used and aligned with Poisson distribution. **d)** Droplet co-occupancy in different mixing concentrations of the sender and receiver cells.

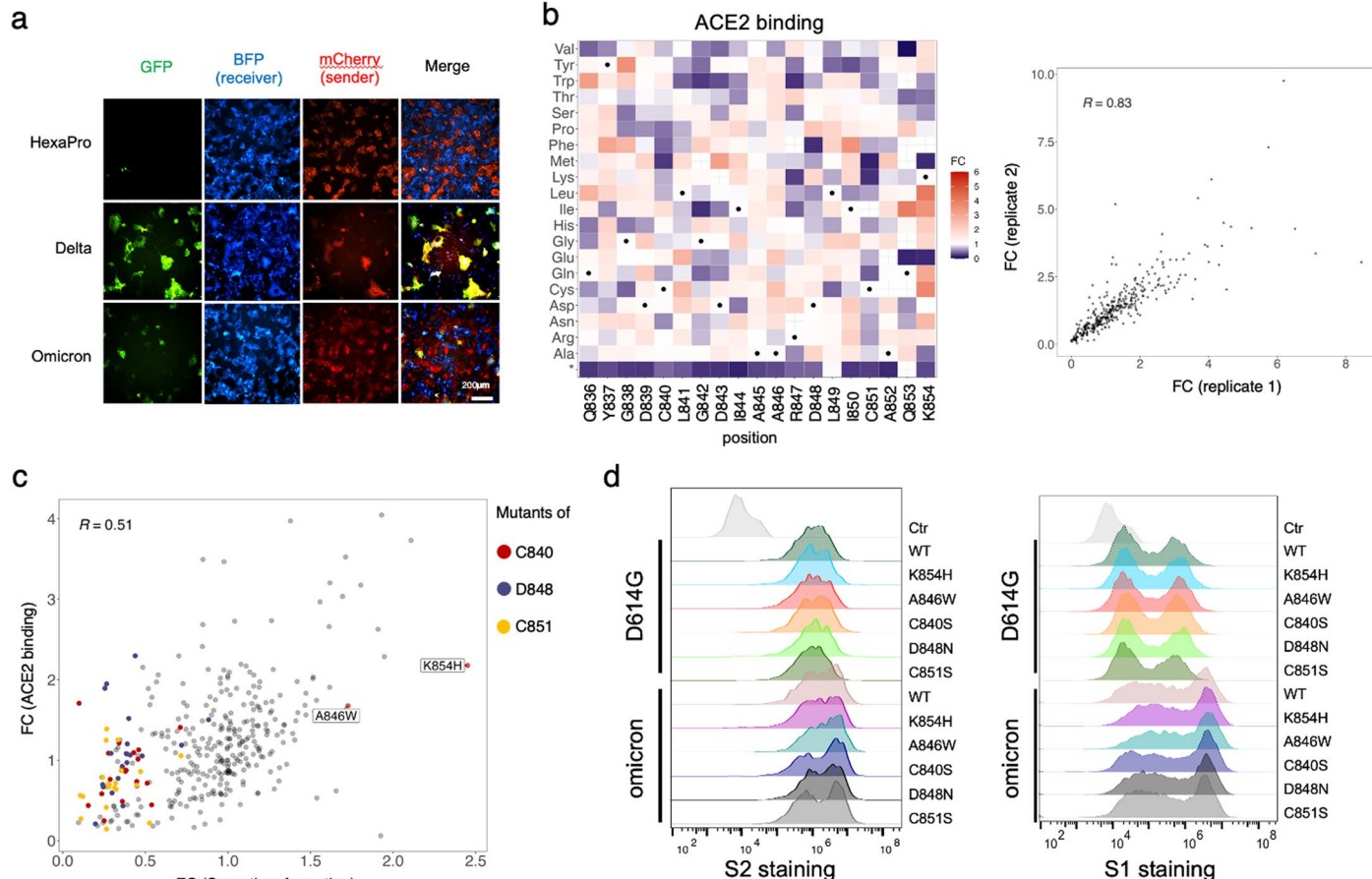

**Extended Data Fig. 2 | Deep mutational scanning of the ACE2-binding ability of the SARS-CoV-2 Spike. a)** Representative images showing the syncytium-forming potential of Omicron Spike variants. The fusion-defective HexaPro mutant and the Delta variant were included for comparison. **b)** Heatmap depicting how all single mutations affect the ACE2 binding of Spike's FPPR. Squares are colored by mutational effect according to scale bars on the right, with red and purple indicating ACE2 binding-enhancing and inhibiting substitutions, respectively. The mutations with no measurement are in gray cross. The SARS-CoV-2 amino acid is indicated with a black dot. Stop codon is indicated with *. The FC value represents the ACE2 binding ability for each of the single mutations in the Spike FPPR library. It is calculated as the fold change

comparing each variant's relative abundance in FACS-sorted (that is, AF405-positive) ACE2-bound cell pool versus the unsorted cell pool and is normalized to wild-type. High reproducibility of the profiling result was detected between two biological replicates. **c)** Correlation of the syncytium-forming potential and the ability of ACE2 binding of all single mutations of Spike's FPPR. Mutants validated are highlighted and labeled. R is the Pearson correlation coefficient. **d)** Cell surface staining of Spike FPPR variants in HEK293T sender cells. Anti-Spike antibodies against its S2 and S1 subunit were used to evaluate the total surface expression of Spike protein and the level of S1-cleaved Spike on cell surface, respectively. Ctr represents the control cells without Spike expression.

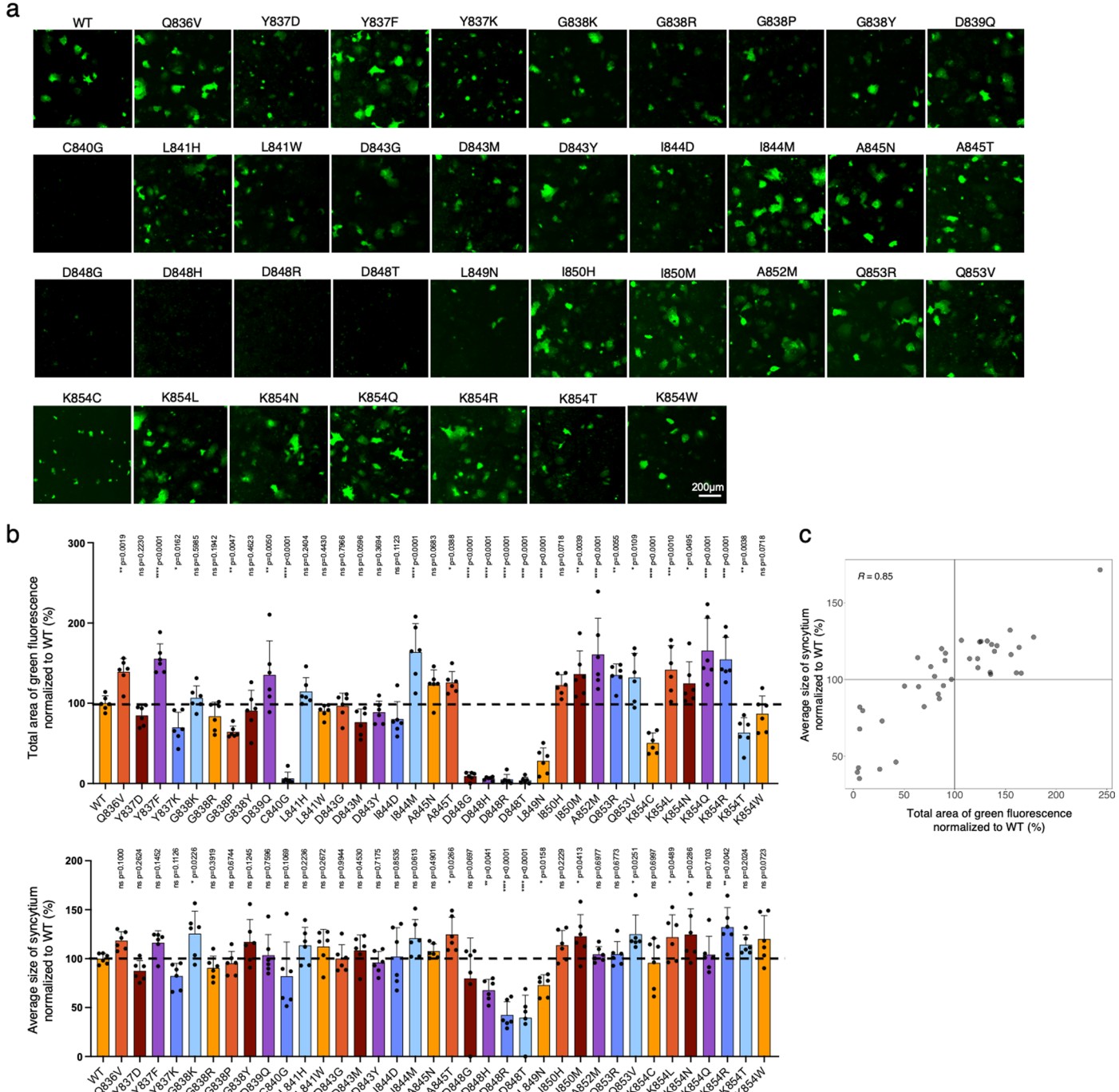

**Extended Data Fig. 3 | Individual validation of FPPR variants of Spike.**
**a)** GFP-split complementation system was applied. A549-ACE2-GFP1-10 receiver cells and HEK293T-GFP-11 sender cells that express WT D614G Spike were used. Representative images are shown. **b)** Quantification of the syncytium area and average size of syncytium for Spike variants in (a). Data shown are mean ± SD

(n = 6). P-values indicated were compared with the D614G WT. Statistical significance was determined using one-way ANOVA. ns: no significance. *P < 0.05, **P < 0.01, ***P < 0.001, ****P < 0.0001. n indicates the number of biological replicates. **c)** Correlation of the syncytium area and average size of syncytium quantified for the Spike variants.

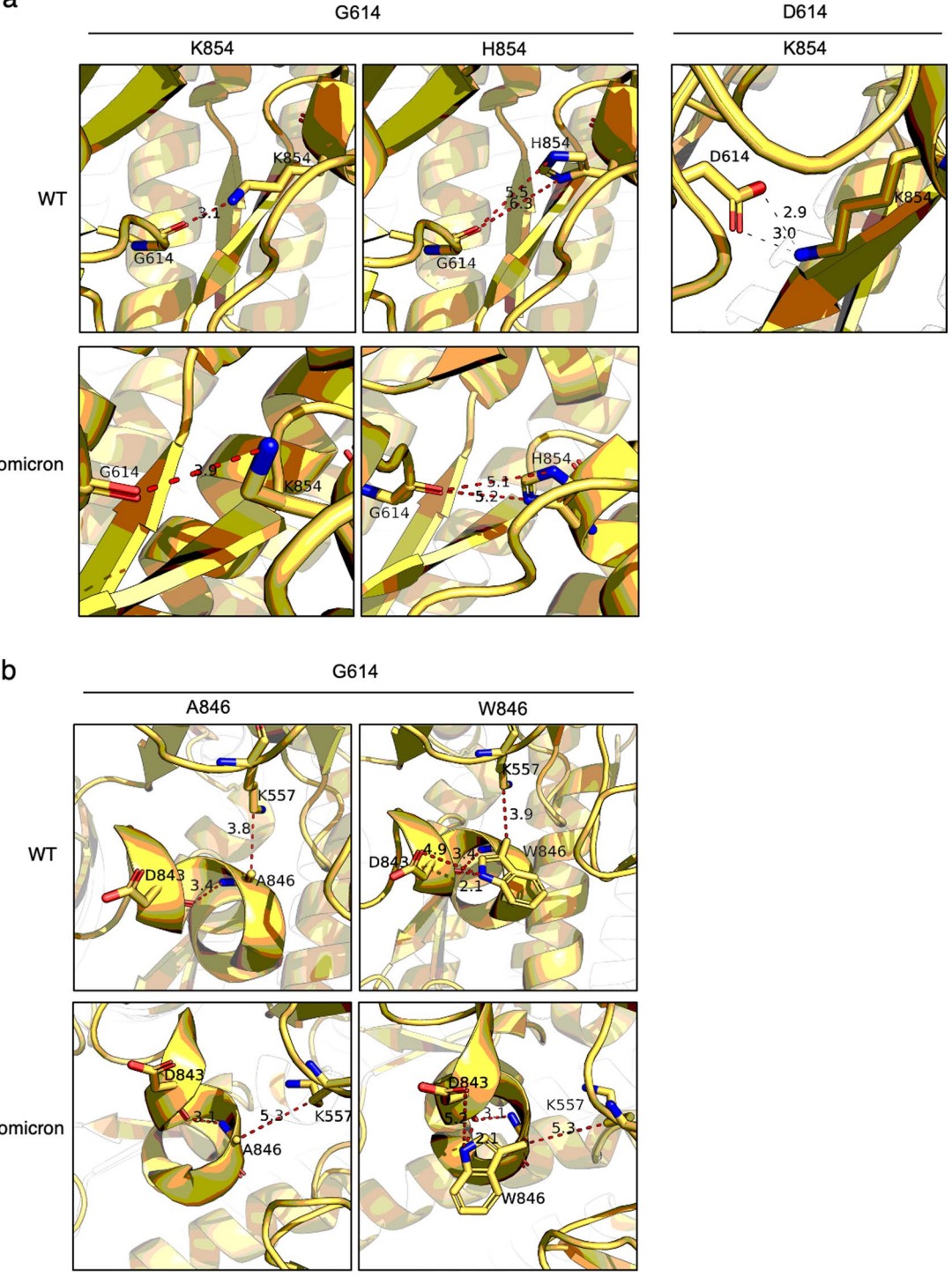

**Extended Data Fig. 4 | Structure models of K854H and A846W mutants of the SARS-CoV-2 Spike. a-b)** Parental D614 Spike (PDB: 6XR8), D614G Spike (PDB: 7KRQ), Omicron Spike (PDB: 7TO4) structures were used for the molecular modelling of K854H **(a)** and A846W **(b)** mutants.

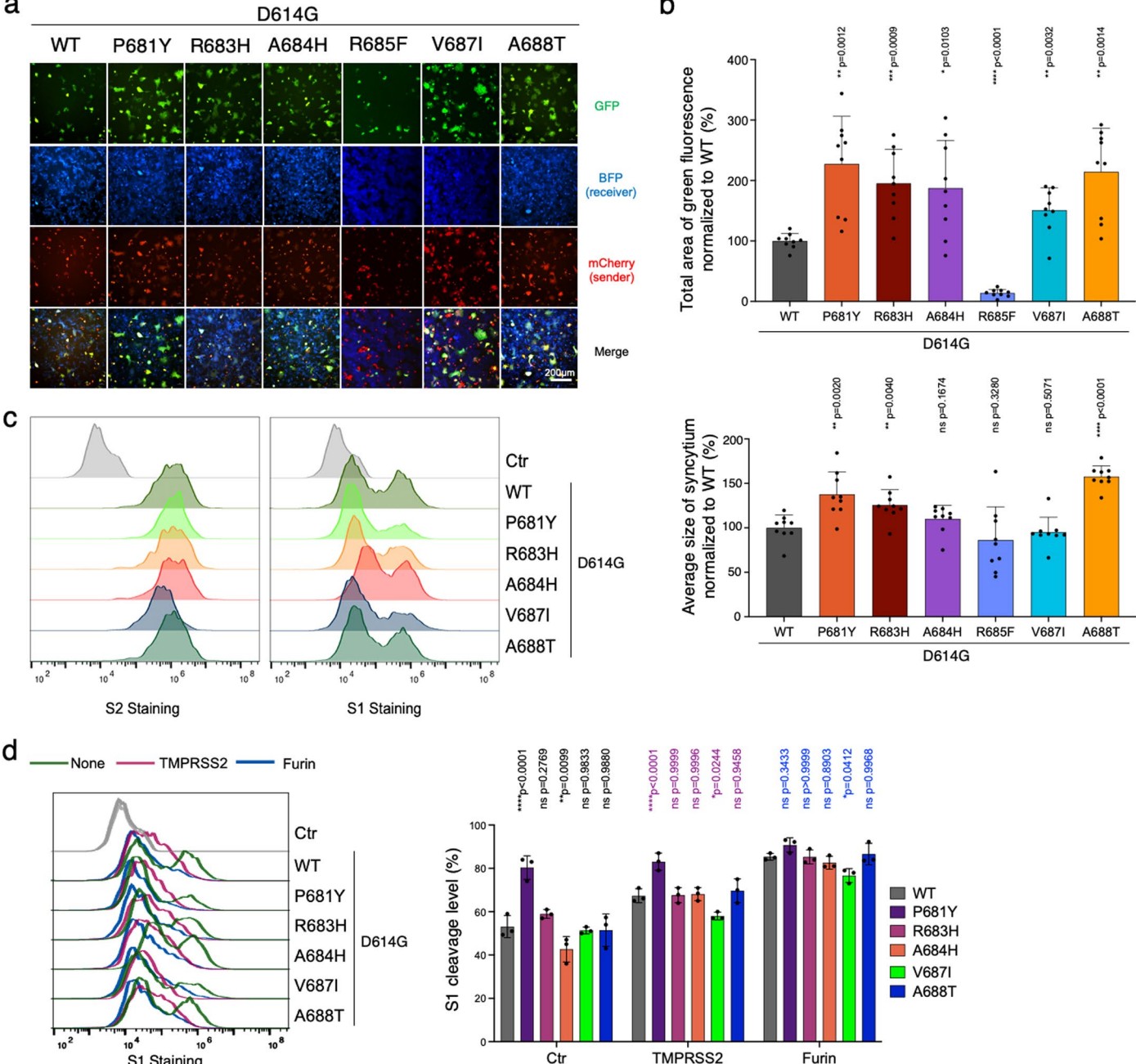

**Extended Data Fig. 5 | Validation of syncytium-enhancing mutations at the furin cleavage site region in D614G Spike. a)** GFP-split complementation system was applied for assessing the syncytium-enhancing mutations at the furin cleavage site region in D614G Spike. A549-ACE2-GFP1-10 receiver cells and HEK293T-GFP-11 sender cells that express WT D614G Spike were used. Representative images are shown. **b)** Quantification of the syncytium area and average size of syncytium for Spike variants in (a). Data shown are mean ± SD (n = 9). P-values indicated were compared with the D614G WT. **c-d)** Cell surface staining of Spike furin cleavage site variants in HEK293T sender cells. Anti-Spike antibodies against its S2 and S1 subunit were used to evaluate the total surface

expression of Spike protein and the level of S1-cleaved Spike on cell surface, respectively. Ctr represents the control cells without Spike expression. In **(d)**, the TMPRSS2 and Furin groups are HEK293T sender cells overexpressed with TMPRSS2 and Furin, respectively. HEK293T sender cells used in **(c)** and the 'none' group in **(d)** express only minimal levels of TMPRSS2 and Furin. The percentage of Spike cleavage is determined by the proportion of S1-stained cells. Data shown are mean ± SD (n = 3). P-values indicated were compared with the D614G WT in each group (that is, Ctr/TMPRSS2/Furin). Statistical significance was determined using one-way ANOVA. ns: no significance. *P < 0.05, **P < 0.01, ***P < 0.001, ****P < 0.0001. n indicates the number of biological replicates.

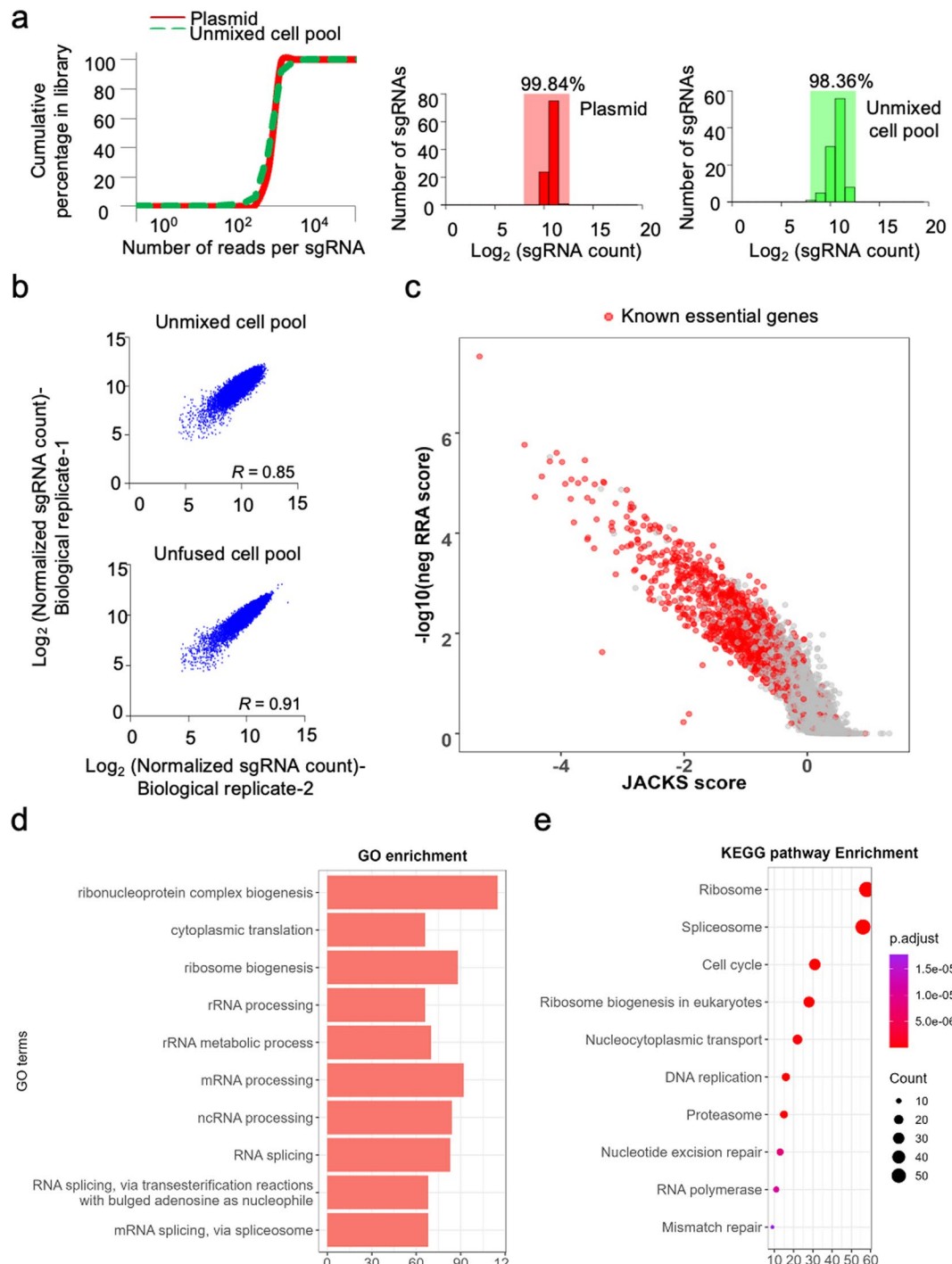

**Extended Data Fig. 6 | Quality assessment of the genome-wide CRISPR screen. a)** Distribution of sgRNA reads in the plasmid pool extracted from E. coli and unmixed A549-ACE2-Cas9-GFP1-10-sgRNA cell pool at 14-day post-infection. 99.8% and 98.4% of all expected sgRNAs were obtained in the plasmid and cell pools, respectively. **b)** High reproducibility of sgRNA representations was detected between two biological replicates before (that is, unmixed) and after (that is, unfused) selection. R is the Pearson correlation coefficient.

**c)** Correlation of -log10(negative RRA score) generated by MAGeCK and JACKS score by comparing sgRNA abundance in library-infected A549-ACE2-Cas9-GFP1-10-sgRNA cells (14-day post-infection) and the plasmid pool. Known essential genes are labeled in red. **d-e)** GO enrichment analysis **(d)** and KEGG classification enrichment analysis **(e)** of the depleted genes in A549-ACE2-Cas9-GFP1-10-sgRNA cells at 14-day post-infection.

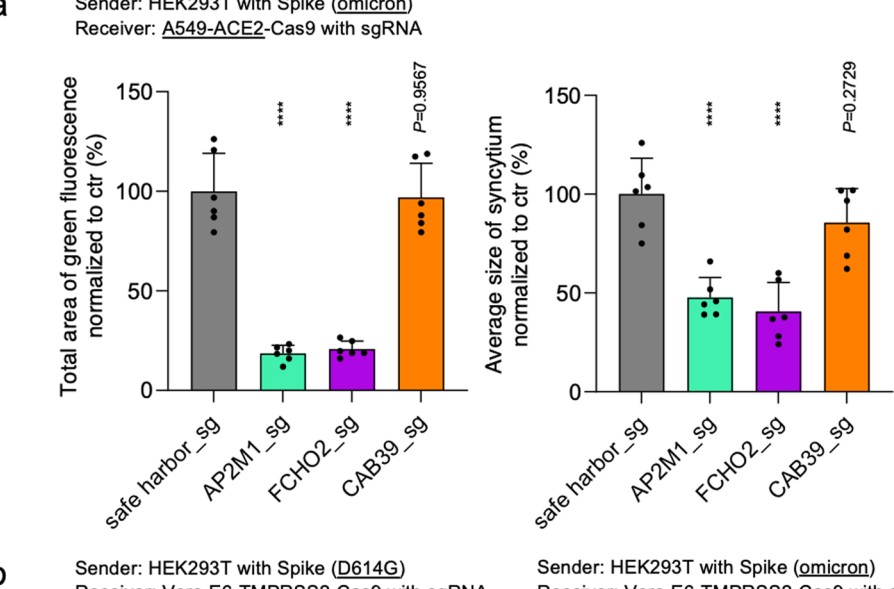

a
Sender: HEK293T with Spike (omicron)
Receiver: A549-ACE2-Cas9 with sgRNA

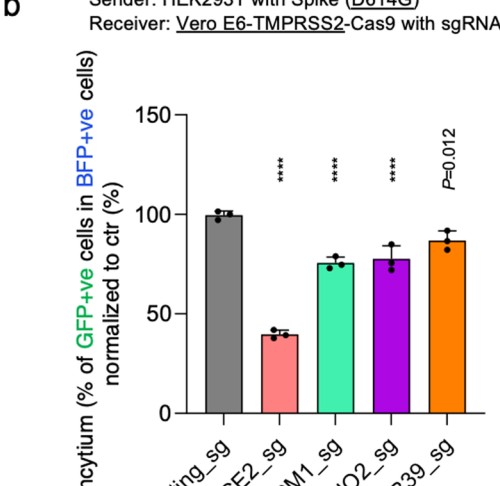

b
Sender: HEK293T with Spike (D614G)
Receiver: Vero E6-TMPRSS2-Cas9 with sgRNA

Sender: HEK293T with Spike (omicron)
Receiver: Vero E6-TMPRSS2-Cas9 with sgRNA

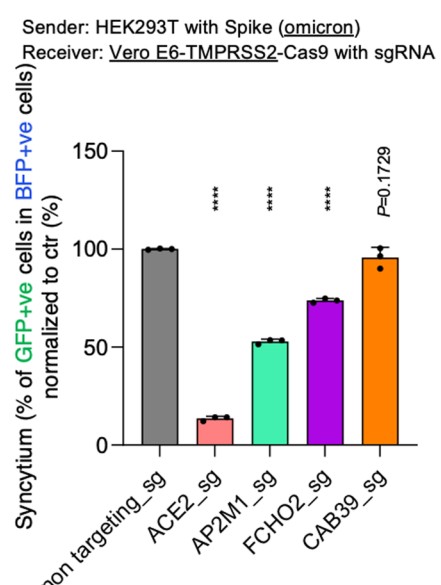

**Extended Data Fig. 7 | Knockout of FCHO2 and AP2M1 inhibits Omicron Spike-induced syncytium formation. a-b)** Validation of screen hits using sgRNA-directed gene knockouts in A549-ACE2-Cas9 and Vero E6-TMPRSS2-Cas9 receiver cells. The sender cells used to express either WT D614G Spike or Omicron Spike. Data shown are mean ± SD (n = 6 for **(a)** and n = 3 for **(b)**). P-values indicated were compared with safe harbor-targeting **(a)** or non-targeting **(b)** sgRNA control. Statistical significance was determined using one-way ANOVA. n indicates the number of biological replicates. ****P < 0.0001.

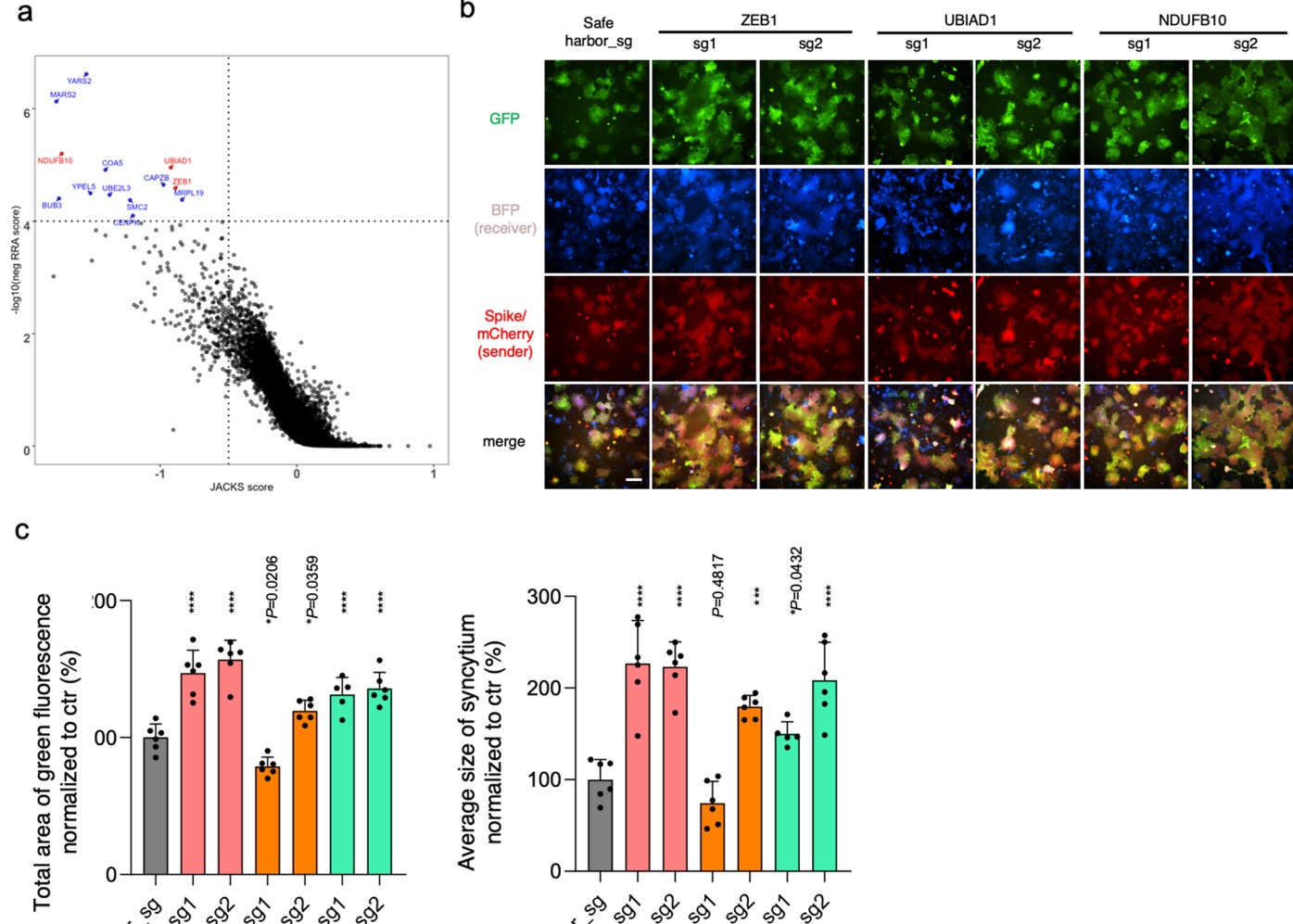

**Extended Data Fig. 8 | A genome-wide CRISPR screen reveals host-factor knockouts that enhance SARS-CoV-2 Spike-induced syncytium formation.** **a)** Depletion of sgRNAs in the unfused receiver cell population revealed by the RRA and JACKS scores. The sgRNA screen hits are highlighted in red, and the known essential genes are highlighted in blue. The genome-wide CRISPR screen data were collected from two biological replicates. **b)** Validation of screen hits using sgRNA-directed gene knockouts in A549-ACE2-Cas9-GFP1-10 receiver cells. The HEK293T-GFP-11 sender cells used express WT D614G Spike. Scale bar, 200 µm. **c)** Quantification of the syncytium area and average size of syncytium observed in **(b)**. Data shown are mean ± SD (n = 6). P-values indicated were compared with safe harbor-targeting control. Statistical significance was determined using one-way ANOVA. n indicates the number of biological replicates. ***P < 0.001, ****P < 0.0001.

# Reporting Summary

## Statistics

For all statistical analyses, confirm that the following items are present in the figure legend, table legend, main text, or Methods section.

| n/a | Confirmed | |
|---|---|---|
| ☐ | ☒ | The exact sample size (*n*) for each experimental group/condition, given as a discrete number and unit of measurement |
| ☐ | ☒ | A statement on whether measurements were taken from distinct samples or whether the same sample was measured repeatedly |
| ☐ | ☒ | The statistical test(s) used AND whether they are one- or two-sided *Only common tests should be described solely by name; describe more complex techniques in the Methods section.* |
| ☒ | ☐ | A description of all covariates tested |
| ☒ | ☐ | A description of any assumptions or corrections, such as tests of normality and adjustment for multiple comparisons |
| ☐ | ☒ | A full description of the statistical parameters including central tendency (e.g. means) or other basic estimates (e.g. regression coefficient) AND variation (e.g. standard deviation) or associated estimates of uncertainty (e.g. confidence intervals) |
| ☐ | ☒ | For null hypothesis testing, the test statistic (e.g. *F*, *t*, *r*) with confidence intervals, effect sizes, degrees of freedom and *P* value noted *Give P values as exact values whenever suitable.* |
| ☒ | ☐ | For Bayesian analysis, information on the choice of priors and Markov chain Monte Carlo settings |
| ☒ | ☐ | For hierarchical and complex designs, identification of the appropriate level for tests and full reporting of outcomes |
| ☐ | ☒ | Estimates of effect sizes (e.g. Cohen's *d*, Pearson's *r*), indicating how they were calculated |

*Our web collection on statistics for biologists contains articles on many of the points above.*

## Software and code

Policy information about availability of computer code

| Data collection | High-throughput sequencing data were collected using Illumina NovaSeq 6000 with NovaSeq Control software. Flow-cytometry data were collected using the BD FACSDiva(v9.0) software. High-throughput imaging data were collected using GE IN Cell Analyzer 6500HS. |
|---|---|
| Data analysis | DiMSum was used for analysing the Spike DMS NGS data (https://github.com/lehner-lab/DiMSum). MAGeCK (https://sourceforge.net/p/mageck/wiki/Home/) and JACKS (https://github.com/felicityallen/JACKS) were used to analyse the genome-wide CRISPR screen data. Kallisto (https://github.com/pachterlab/kallisto), tximport (https://github.com/mikelove/tximport), and DESeq2(http://bioconductor.org/packages/devel/bioc/vignettes/DESeq2/inst/doc/DESeq2.html#input-data) were used to analyse the RNA-seq data. Flowjo (version 10.8.1) was used to analyse data generated from the flow-cytometry experiments. Prism (v9) was used to plot the bar plots and to perform one-way ANOVA tests. R (v2021.09.2+382) with packages ggplot2 (v3.3.6) and tidyverse (v1.3.2) was used for plotting the scatter plots and heatmaps. R (v2021.09.2+382) with packages clusterProfiler(v.4.4.4) was used for GO analysis. GE IN Carta image analysing software(v2.x) and ImageJ (v1.53u) was used for the quantification of total area of green fluorescence. PyMOL (v2.5.2) was used for the molecular modelling and visualization of protein models. |

For manuscripts utilizing custom algorithms or software that are central to the research but not yet described in published literature, software must be made available to editors and reviewers. We strongly encourage code deposition in a community repository (e.g. GitHub). See the Nature Portfolio guidelines for submitting code & software for further information.

## Data

Policy information about availability of data

All manuscripts must include a data availability statement. This statement should provide the following information, where applicable:
- Accession codes, unique identifiers, or web links for publicly available datasets
- A description of any restrictions on data availability
- For clinical datasets or third party data, please ensure that the statement adheres to our policy

> The main data supporting the results in this study are available within the paper and its Supplementary Information. The molecular structures of the Spike proteins of the SARS-CoV-2 variants are available from the Protein Data Bank, with accession codes 6XR8, 7KRQ and 7TO4. Source data for the figures are provided with this paper. The raw and analysed datasets generated during the study are available for research purposes from the corresponding authors on reasonable request.

## Human research participants

Policy information about studies involving human research participants and Sex and Gender in Research.

| | |
|---|---|
| Reporting on sex and gender | The study did not involve human research participants. |
| Population characteristics | — |
| Recruitment | — |
| Ethics oversight | — |

Note that full information on the approval of the study protocol must also be provided in the manuscript.

# Field-specific reporting

Please select the one below that is the best fit for your research. If you are not sure, read the appropriate sections before making your selection.

☒ Life sciences          ☐ Behavioural & social sciences          ☐ Ecological, evolutionary & environmental sciences

For a reference copy of the document with all sections, see nature.com/documents/nr-reporting-summary-flat.pdf

# Life sciences study design

All studies must disclose on these points even when the disclosure is negative.

| | |
|---|---|
| Sample size | Sample sizes were chosen to be able to show reproducibility and statistical significance. No methods were used to predetermine sample size. >500-fold more cells for lentiviral infection than the size of the library being tested in genetic screens were used to ensure high fold-representation. From the NGS data, >98% coverage of the variants could be achieved with this sample size. |
| Data exclusions | It was previously reported that filtering of the genetic screen data to remove library members with low representation in the reference set resulted in a reduced false-negative rate (Sim et al., Genome Biol. 2011; 12(10): R104). Yet, the exclusion criteria has not been standardized and thus were not pre-established. For Illumina sequencing data from the Spike DMS screens and the genome-wide CRISPR screen in this study, only single variants or sgRNAs that gave more than 50 absolute reads in the unsorted population were analysed to improve data reliability. |
| Replication | All data were reliably reproduced. The methods and materials used in our experiments are described to facilitate replication. Transfection of of the onstructs into human cells was performed independently to produce biological replicates. Infected cell pools were sorted into bins independently to produce biological replicates for genomic-DNA extraction for NGS sequencing. All biological replicates were analysed independently, and replicate numbers are provided in the text and figure legends. |
| Randomization | No randomization was used for samples, as samples with particular genetic constituents were needed for the experiments. During the construction of mutant libraries, cell culture, transfection, infection, cell sorting, sample preparation for NGS sequencing and data analysis, samples were not grouped in a way relating to the identity of the sample. The timing of when samples were ready determined the grouping of the samples in sequencing runs. |
| Blinding | Blinding was not relevant to the study, as samples with particular genetic constituents were needed for the experiments. Sample labelling was used to prevent mixing up experimental samples. |

# Reporting for specific materials, systems and methods

We require information from authors about some types of materials, experimental systems and methods used in many studies. Here, indicate whether each material, system or method listed is relevant to your study. If you are not sure if a list item applies to your research, read the appropriate section before selecting a response.

## Materials & experimental systems

| n/a | Involved in the study |
|-----|----------------------|
| ☐ | ☒ Antibodies |
| ☐ | ☒ Eukaryotic cell lines |
| ☒ | ☐ Palaeontology and archaeology |
| ☐ | ☒ Animals and other organisms |
| ☒ | ☐ Clinical data |
| ☒ | ☐ Dual use research of concern |

## Methods

| n/a | Involved in the study |
|-----|----------------------|
| ☒ | ☐ ChIP-seq |
| ☐ | ☒ Flow cytometry |
| ☒ | ☐ MRI-based neuroimaging |

## Antibodies

| Antibodies used | Rabbit Anti-SARS-CoV-2 Spike (944-1214aa) polyclonal antibodies: https://www.ptglab.com/products/spike-protein-944-1214aa-Antibody-28867-1-AP.htm<br>Supplier name: Proteintech<br>Catalog number: 28867-1-AP<br>Dilution: 1:500 |
|---|---|
| | SARS-CoV-2 Spike S1 Subunit Alexa Fluor® 488-conjugated Antibody<br>https://www.rndsystems.com/products/sars-cov-2-spike-s1-subunit-alexa-fluor-488-conjugated-antibody-1035206_fab105403g<br>Supplier name: R&D<br>Clone #1035206<br>Catalog number: FAB105403G<br>Dilution: 1:200 |
| | Goat Anti-ACE2 polyclonal antibodies: https://www.rndsystems.com/products/human-mouse-rat-hamster-ace-2-antibody_af933<br>Supplier name: R&D Systems<br>Catalog number: AF933<br>Dilution: 1:100 |
| | Donkey anti-Goat IgG(H+L) Cross-Adsorbed Secondary Antibody-AF568: https://www.thermofisher.com/antibody/product/Donkey-anti-Goat-IgG-H-L-Cross-Adsorbed-Secondary-Antibody-Polyclonal/A-11057<br>Supplier name: Thermo Fisher<br>Catalog number: A-11057<br>Dilution: 1:1000 |
| | Goat anti-Rabbit IgG(H+L) Cross-Adsorbed Secondary Antibody-AF488: https://www.thermofisher.com/antibody/product/Goat-anti-Rabbit-IgG-H-L-Cross-Adsorbed-Secondary-Antibody-Polyclonal/A-11008<br>Supplier name: Thermo Fisher<br>Catalog number: A-11008<br>Dilution: 1:1000 |
| | in-house rabbit anti-SARS-CoV-2 N antibody |
| | in-house guinea pig anti-SARS-Cov-2 N antibody |
| | Recombinant Anti-Sodium Potassium ATPase antibody https://www.abcam.com/products/primary-antibodies/sodium-potassium-atpase-antibody-ep1845y-plasma-membrane-loading-control-ab76020.html<br>Supplier name: abcam<br>Catalog number: ab76020<br>Dilution: 1:500 |
| | Goat anti-Guinea Pig IgG (H+L) Highly Cross-Adsorbed Secondary Antibody, Alexa Fluor™ 488<br>https://www.thermofisher.com/antibody/product/Goat-anti-Guinea-Pig-IgG-H-L-Highly-Cross-Adsorbed-Secondary-Antibody-Polyclonal/A-11073<br>Supplier name: Thermo Fisher<br>Catalog number: A-11073<br>Dilution: 1:1000 |
| | Goat anti-Rabbit IgG (H+L) Cross-Adsorbed Secondary Antibody, Alexa Fluor™ 568<br>https://www.thermofisher.com/antibody/product/Goat-anti-Rabbit-IgG-H-L-Cross-Adsorbed-Secondary-Antibody-Polyclonal/A-11011<br>Supplier name: Thermo Fisher<br>Catalog number: A-11011<br>Dilution: 1:1000 |
| | Anti-Clathrin heavy chain antibody (ab21679)<br>https://www.abcam.com/products/primary-antibodies/clathrin-heavy-chain-antibody-ab21679.html |

Supplier name: abcam
Catalog number: ab21679
Dilution: 1:900

GAPDH (14C10) Rabbit mAb #2118
https://www.cellsignal.com/products/primary-antibodies/gapdh-14c10-rabbit-mab/2118
Supplier name: Cell Signaling
Catalog number: 2118
Dilution: 1:5000

Anti-rabbit IgG, HRP-linked Antibody #7074
https://www.cellsignal.com/products/secondary-antibodies/anti-rabbit-igg-hrp-linked-antibody/7074
Supplier name: Cell Signaling
Catalog number: 7074
Dilution: 1:10000

**Validation**

Polyclonal anti-SARS-CoV-2 Spike antibodies recognize an epitope located on the S2 (944-1214aa) of the Spike protien. The antibodies are validated by the commercial vendor. (https://www.ptglab.com/products/pictures/pdf/28867-1-AP.pdf)

Monoclonal anti-SARS-Cov-2 spike S1 antibody is validated by the commercial vendor. (https://resources.rndsystems.com/pdfs/datasheets/fab105403g.pdf?
v=20230516&_ga=2.4377843.1359546541.1684248237-1703500799.1684248237&_gac=1.82504036.1684248259.CjwKCAjw04yjBh
ApEiwAJcvNoSpk9L07LH1ShqLhoNpqEDdgtriMW5aU7QzupvzVfEWLos7SFuWu-hoCADkQAvD_BwE)

Polyclonal anti-ACE2 antibodies recognize human/mouse/rat/hamster ACE2. The antibodies are validated for flow cytometry, WB, and IHC by the commercial vendor. (https://resources.rndsystems.com/pdfs/datasheets/af933.pdf?
v=20221204&_ga=2.166088636.324426401.1670231890-376477508.1670231890&_gac=1.242411958.1670231890.EAIaIQobChMI_
KaG0ZLi-wIVRNeWCh0nVgB2EAAYASAAEgInj_D_BwE)

The in-house rabbit anti-SARS-CoV-2 N antibody and guinea pig anti-SARS-Cov-2 N antibody were validated immunofluorescence staining in the previous publication (PMID: 36662861)

Recombinant Anti-Sodium Potassium ATPase antibody recognizes an intracellular epitope of Sodium/potassium-transporting ATPase alpha-1 subunit. This antibody is recommended for ICC, Flow Cytometry, WB, and IHC. This antibody is validated by the commercial vendor. (file:///C:/Users/bwang/Downloads/datasheet_76020.pdf)

Anti-Clathrin heavy chain antibody is recommended for WB, IHC, and ICC. This antibody is validated by the commercial vendor. (file:///C:/Users/bwang/Downloads/datasheet_21679.pdf)

GAPDH (14C10) Rabbit mAb is used to detect endogenous levels of total GAPDH protein in Human, Mouse, Rat, Monkey, Bovine, Pig. It is recommended for WB, IHC, IF, and Flow Cytometry. This antibody is validated by the commercial vendor. (https://www.cellsignal.com/products/primary-antibodies/gapdh-14c10-rabbit-mab/2118)

Anti-rabbit IgG, HRP-linked Antibody is used for chemiluminescent detection. This antibody is validated by the commercial vendor. (https://www.cellsignal.com/products/secondary-antibodies/anti-rabbit-igg-hrp-linked-antibody/7074)

Goat anti-Guinea Pig IgG(H+L) Secondary Antibody-AF488 is recommended for ICC/IF and IHC. This antibody is validated by the commercial vendor. (https://www.thermofisher.com/order/genome-database/dataSheetPdf?
producttype=antibody&productsubtype=antibody_secondary&productId=A-11073&version=300)

Goat anti-Rabbit IgG(H+L) Secondary Antibody-AF568 is recommended for Flow Cytometry, ICC/IF and IHC. This antibody is validated by the commercial vendor. (https://www.thermofisher.com/order/genome-database/dataSheetPdf?
producttype=antibody&productsubtype=antibody_secondary&productId=A-11011&version=300)

anti-Goat IgG(H+L) Secondary Antibody-AF568 is recommended for flow cytometry, IHC, and ICC/IF. This antibody is validated by the commercial vendor. (https://www.thermofisher.com/order/genome-database/dataSheetPdf?
producttype=antibody&productsubtype=antibody_secondary&productId=A-11057&version=271)

anti-rabbit IgG(H+L) Secondary Antibody-AF488 is recommended for flow cytometry, IHC, and ICC/IF. This antibody is validated by the commercial vendor. (https://www.thermofisher.com/order/genome-database/dataSheetPdf?
producttype=antibody&productsubtype=antibody_secondary&productId=A-11008&version=271)

# Eukaryotic cell lines

Policy information about cell lines and Sex and Gender in Research

| | |
|---|---|
| Cell line source(s) | HEK293T and Vero E6 cells were obtained from American Type Culture Collection (ATCC). VeroE6-TMPRSS2 was obtained from the Japanese Collection of Research Bioresources Cell Bank. A549-ACE2 cells were obtained from InvivoGen https://www.invivogen.com/a549-ace2-tmprss2-cells. |
| Authentication | HEK293T cells were authenticated by STR profiling by the commercial vendor. A549-ACE2, Vero E6 and Vero E6-TMPRSS2 cell lines were not authenticated after receiving them. |
| Mycoplasma contamination | Mycoplasma contamination was tested and confirmed to be negative. All cell-culture medium was supplemented with antibiotic-antimycotic solution to prevent bacterial and fungal contamination. |

| Commonly misidentified lines (See ICLAC register) | No commonly misidentified cell lines were used (HEK293T, A549 and Vero E6 cells are not included in the ICLAC register). |
| --- | --- |

# Animals and other research organisms

Policy information about studies involving animals; ARRIVE guidelines recommended for reporting animal research, and Sex and Gender in Research

| Laboratory animals | Syrian hamsters (4 to 6 weeks old, male) were obtained from the Chinese University of Hong Kong Laboratory Animal Service Centre through the HKU Centre for Comparative Medicine Research. |
| --- | --- |
| Wild animals | The study did not involve wild animals. |
| Reporting on sex | Male Golden Syrian hamsters. |
| Field-collected samples | The study did not involve samples collected from the field. |
| Ethics oversight | The animal studies were approved by the Committee on the Use of Live Animals in Teaching and Research of The University of Hong Kong. |

Note that full information on the approval of the study protocol must also be provided in the manuscript.

# Flow Cytometry

## Plots

Confirm that:

☒ The axis labels state the marker and fluorochrome used (e.g. CD4-FITC).

☒ The axis scales are clearly visible. Include numbers along axes only for bottom left plot of group (a 'group' is an analysis of identical markers).

☒ All plots are contour plots with outliers or pseudocolor plots.

☒ A numerical value for number of cells or percentage (with statistics) is provided.

## Methodology

| Sample preparation | Cell cultures were treated with trypsin and diluted in complete media or PBS for flow-cytometry experiments. |
| --- | --- |
| Instrument | BD LSRFortessaTM, ACEA NovoCyte Quanteon, Agilent NovoCyte Advanteon BVYG were used for data collection. Cell sorting was performed on a BD Influx cell sorter. |
| Software | All cytometry data were analysed by FlowJo (v10.8.1). |
| Cell population abundance | Drop delay was determined using BD Accudrop beads. Cells were filtered through nylon mesh filters before sorting through a 100-μm nozzle using 1.0 Drop Pure sorting mode. Details are described in Methods. |
| Gating strategy | Viable and intact cells were gated from FSC/SSC for analysis. Details are described in Methods. |

☒ Tick this box to confirm that a figure exemplifying the gating strategy is provided in the Supplementary Information.

