## [Peer Review File · Nature Biomedical Engineering]

High-throughput screening of genetic and cellular drivers of syncytium formation induced by the spike protein of SARS-CoV-2

Corresponding author: Alan Wong

Editorial note

This document includes relevant written communications between the manuscript's corresponding author and the editor and reviewers of the manuscript during peer review. It includes decision letters relaying any editorial points and peer-review reports, and the authors' replies to these (under 'Rebuttal' headings). The editorial decisions are signed by the manuscript's handling editor, yet the editorial team and ultimately the journal's Chief Editor share responsibility for all decisions.

Any relevant documents attached to the decision letters are referred to as **Appendix #**, and can be found appended to this document. Any information deemed confidential has been redacted or removed. Earlier versions of the manuscript are not published, yet the originally submitted version may be available as a preprint. Because of editorial edits and changes during peer review, the published title of the paper and the title mentioned in below correspondence may differ.

Correspondence

Mon 06 Feb 2023

Decision on Article nBME-22-2865

Dear Dr Wong,

Thank you again for submitting to *Nature Biomedical Engineering* your manuscript, "Massively parallel paired-cell profiling reveals determinants for SARS-CoV-2 Spike-induced syncytium formation". The manuscript has been seen by 3 experts, whose reports you will find at the end of this message.

You will see that the reviewers appreciate the work. However, they express concerns about the degree of support for the claims, and provide useful suggestions for improvement. We hope that with significant further work you can address the criticisms and convince the reviewers of the merits of the study. In particular, we would expect that a revised version of the manuscript provides:

- * Comprehensive functional validation of the identified Spike variants.
- * An extended drug screen, ideally with in vivo validation, as suggested by Reviewer #3.
- * Additional insights into the molecular mechanisms by which the top mutations identified lead to enhanced syncytium formation.
- * Discussion of the performance advantages of your droplet microfluidics and size-exclusion setup, with respect to the state of the art.
- * Thorough methodological details, as per the many relevant comments of all reviewers.

When you are ready to resubmit your manuscript, please upload the revised files, a point-by-point rebuttal to the comments from all reviewers, the reporting summary, and a cover letter that explains the mainimprovements included in the revision and responds to any points highlighted in this decision.

Please follow the following recommendations:

- * Clearly highlight any amendments to the text and figures to help the reviewers and editors find and understand the changes (yet keep in mind that excessive marking can hinder readability).
- * If you and your co-authors disagree with a criticism, provide the arguments to the reviewer (optionally, indicate the relevant points in the cover letter).
- * If a criticism or suggestion is not addressed, please indicate so in the rebuttal to the reviewer comments and explain the reason(s).
- * Consider including responses to any criticisms raised by more than one reviewer at the beginning of the rebuttal, in a section addressed to all reviewers.
- * The rebuttal should include the reviewer comments in point-by-point format (please note that we provide all reviewers will the reports as they appear at the end of this message).
- * Provide the rebuttal to the reviewer comments and the cover letter as separate files.

We hope that you will be able to resubmit the manuscript within 20 weeks from the receipt of this message. If this is the case, you will be protected against potential scooping. Otherwise, we will be happy to consider a revised manuscript as long as the significance of the work is not compromised by work published elsewhere or accepted for publication at *Nature Biomedical Engineering*.

We hope that you will find the referee reports helpful when revising the work. Please do not hesitate to contact me should you have any questions.

Best wishes,

Liqian

Dr Liqian Wang
Associate Editor, Nature Biomedical Engineering

Reviewer #1 (Report for the authors (Required)):

Chan et al present a droplet microfluidic platform for high throughput screening of a) mutations in the Spike protein of SARS-Covid-2 and b) cellular factors in the receiver cell, enhancing syncytia formation as a proxy for pathological impact. In general, this work could be of interest for a wider community, but for truly assessing the impact, a couple of points have to be addressed comprehensively:

Another method for high-throughput screening of mutations enhancing syncytia formation has been published previously (Nucleic Acids Res. 34(5):e41. doi: 10.1093/nar/gkl053. 2006) and should be discussed in the context of the current work. It does overcome many disadvantages of conventional low-throughput imaging-based methods that are mentioned in the introduction. What is the specific advantage of using droplets in the current work?

The selection of large syncytia using a cell strainer will result in syncytia with polyclonal sender cells. Doesn't this significantly increase the noise and false discovery rate?

The authors set up a system with a cell-cell fusion rate of about 58% for droplets with at least one sender and one receiver cell. How can this be successfully used for the identification of inhibitory variants? Based on such a low fusion rate, a lot of false positives would inevitably be selected.

Extended Data Figure 2 nicely shows that the number of plasmids/cells of a particular variant correlates with the obtained number of corresponding reads only for a very narrow range. How could it be ensured that this range was maintained throughout the entire screening procedure (transduction, cell cultivation and proliferation, etc.)?

In line with this, I was missing sequencing data showing an equal distribution of the different variants across the library immediately before screening. How can it be ruled out that certain variants were already overrepresented in the unselected library?

Functional validation of newly identified variants promoting or inhibiting syncytia formation has only been performed for a very few variants (Extended Figures 5 and 10). What is needed is a more comprehensive analysis on how the read counts of selected variants correlate with their syncytia-forming potency. Does any of the two setups (using droplets or a cell strainer) indeed allow to quantitatively select the very best variants for the given selection criteria or is it rather a selection of any variant above a certain threshold in terms of functionality?

Minor things:

The scale bar for the heat maps in Fig. 2b is too small

The Chip design shown in Extended Data Figure 1b inevitable results in droplets of different size and occupancy (the contents of the single cell- and oil-inlets will be distributed unevenly across the eight drop makers). Why not using a single droplet maker, capable of generating thousands of droplets per second?

Reviewer #2 (Report for the authors (Required)):

Charles W. F. Chan et al. developed two screening approaches for studying determinants of cell-cell fusion. First, they describe a microfluidic strategy to encapsulate cell pairs into droplets. The pairs which fuse are detected by GFP complementation using flow cytometry and sorted for further characterization. They performed a deep mutational scan of two SARS-CoV-2 spike regions identifying several new mutations impacting syncytia formation. Second, they describe a strategy based on size exclusion, as a higher-throughput alternative to the compartmentation method. They performed a genome wide screen and identify core regulators of clathrin-mediated endocytosis as potential host factors that play a role in S-induced syncytium formation, which they then validate experimentally.

The approaches developed in this paper are innovative and fill a methodological gap as high-throughput screening methods for cell-cell fusion determinants are missing. The techniques may have many applications for the study of cell-cell fusion in physiological and pathological settings.

Main comments

1. Fig. 2: Why did the authors only validate 2 mutations which increase syncytia and 3 which decrease? There are many other mutations in the screen such as: A852M, D843M, G838K, D843G/Y, I844D and others. If there was a cut-off, even if arbitrary, it should be indicated

2. Fig 2.e. and Extended Fig. 5. – Surface level quantification of S expression by flow cytometry on the sender cells should be checked to eliminate transfection bias.

3. Extended Fig. 2. – Axis legend only states “replicates 1-2”. What do the values on the axis represent?

4. Figure 2 b. – It is surprising that R685A does not have a greater impact as it should abrogate furin cleavage and be similar to a Δ PRRA mutant and therefore inhibit syncytia. Could you comment on the high mutability of the R685 site in this screen?

5. Fig. 3. c – Bottom axis legend is missing.

6. Extended fig. 3 – It is unclear how the ACE2 binding ability was calculated for each of the single mutations. Was this performed on the pool of transduced cells, and if so, how were the mutations correlated to ACE2 binding? Was it done on individually transduced cells? This should be further detailed in the figure legend and M&M.

Minor comments:

1. Fold changes (FC) and how data are normalized should be better described in the text or figure legends.
2. FPPR (fusion-peptide proximal region) is not defined
3. Some sentences are very long, like line 269: "Using size-exclusion selection-based strategy thus offers a simple and scalable way to perform large-scale DMS for syncytia screening, whilst we noted that the droplet microfluidics-based screening strategy resulted in a greater depletion for the fusion-incompetent cells in the pooled assay which gives wider assay range in defining the enriched fusion-competent variants (Extended Data Fig. 7), which could be due to some fusion-incompetent cells being trapped by neighbouring syncytia as bystanders and thus retained on the cell strainer."

Reviewer #3 (Report for the authors (Required)):

In this work, the authors used droplet microfluidics to scan SARS-Cov-2 mutations associated with syncytium formation. A GFP-split complementation system was used to detect sender-receiver cells fusion (i.e., GFP-positive syncytia) using FACS. The authors used a pooled saturation mutagenesis library and identified syncytia-forming mutations, some of which were previously identified. The authors then carried out a genome-wide KO screen to identify the genetic host factors contributing to syncytia formation. FACS was used for screening after removing large syncytia with a cell strainer. The top hits were functionally validated with CRISPR KOs and two drugs were screened with known inhibitory effect of clathrin-mediated endocytosis (CME). Overall, SARS-Cov-2 remains a major concern and identifying factors contributing to syncytia formation in the lung is interesting. However, the engineering-related aspects are not the main focus but perhaps the biology is novel enough to stand on its own. Below are additional comments.

- 1) The droplet microfluidic system used in this study is very standard.
- 2) No information was provided on the size of syncytia formed which is very important. Given that the authors have already identified a panel of mutations, it might be interesting to correlate each mutation with the average size of formed syncytium.
- 3) It is true that using a cell strainer is a size-exclusion approach, however it would have been more interesting if the authors could have used this type of high-throughput microfluidics approach for that purpose.
- 4) Several CRISPR screens were previously performed to identify the host factors that promote SARS-Cov-2 infection. The authors might need to check to see whether similar hits were recovered.
- 5) The drug screen should be extended because the drugs tested were previously evaluated for their ability to inhibit host factors (M. Grodzki et al., *Genome Medicine*, 2022, 10). Also, there are more than 3,000 FDA/EMA-approved drugs with therapeutic effects on syncytia during SARS-Cov-2 infection. Thus, it might be interesting to extend the drug screen and probably validate the drugs in vivo using a mouse model.
- 6) The authors need to perform a thorough characterization of the molecular mechanisms of action of the candidate hits. RNA-seq reads can be aligned to the human genome and reads for each gene can be determined by STAR analysis.
- 7) The title "massively parallel paired-cell profiling" might need to be reconsidered. I do not think the authors have provided any novel tools to facilitate that other than using a standard droplet microfluidics system and FACS. Likewise, in the Conclusion section "The application of our high-throughput profiling systems together with CRISPR"?? I am not sure if referring to standard droplet microfluidics and FACS using "our" is accurate as well.

Mon 26 Jun 2023

Decision on Article NBME-22-2865A

Dear Dr Wong,

Thank you for your revised manuscript, "Revealing determinants for SARS-CoV-2 Spike-induced syncytium formation via parallel paired-cell profiling", which has been seen by the original reviewers. In their reports, which you will find at the end of this message, you will see that the reviewers acknowledge the improvements to the work and raise a few additional comments that we hope you will be able to address.

As before, when you are ready to resubmit your manuscript, please upload the revised files, a point-by-point rebuttal to the comments from all reviewers, the reporting summary, and a cover letter that explains the main improvements included in the revision and responds to any points highlighted in this decision.

As a reminder, please follow the following recommendations:

- * Clearly highlight any amendments to the text and figures to help the reviewers and editors find and understand the changes (yet keep in mind that excessive marking can hinder readability).
- * If you and your co-authors disagree with a criticism, provide the arguments to the reviewer (optionally, indicate the relevant points in the cover letter).
- * If a criticism or suggestion is not addressed, please indicate so in the rebuttal to the reviewer comments and explain the reason(s).
- * The rebuttal should include the reviewer comments in point-by-point format (please note that we provide all reviewers will the reports as they appear at the end of this message).
- * Provide the rebuttal to the reviewer comments and the cover letter as separate files.

We hope that you will be able to resubmit the manuscript as soon as possible. We look forward to receive a further revised version of the work. Please do not hesitate to contact me should you have any questions.

Best wishes,

Liqian

Dr Liqian Wang
Associate Editor, Nature Biomedical Engineering

Reviewer #1 (Report for the authors (Required)):

In the revised version of their manuscript, Chan et al. provide a lot of new additional experimental data, addressing most of my comments comprehensively. Well done and thanks a lot for all the extra efforts!

There are just two points that still need some correction:

1.) The following statement is simply wrong: "A pool-based method coupling cell-cell fusion with a screening readout of retroviral vector particle packaging and release that transfer genes encoding the fusion-competent membrane protein was reported 21. However, this method is not suitable for studying SARS-CoV-2 Spike protein due to its large size that greatly compromises the viral packaging efficiency and thus lowers the sensitivity of the screening. Also, syncytia with polyclonal sender cells would be formed during the screening process with this method, which could increase noise due to some relatively less fusion-competent cells being fused with neighbouring syncytia containing the more fusion-competent variants and enriched together as large syncytia and packed in the retroviruses." First of all, the cited method made use of packaging

constructs much bigger than the envelope protein (only corresponding to about 20% of the entire packaged sequence). Therefore, a 2-fold larger size of the SARS-CoV-2 Spike protein would probably pose absolutely no problem. Second, the former method also managed to prevent polyclonal sender cells, e.g. by adjusting cell density and cell type ratios. Taken together, the former method could have been applied to the problem addressed in the current work as well, and this should be discussed in a fair and balanced way. The authors openly admitted in their rebuttal letter that they initially overlooked this work, and this should in no ways be “compensated” by attributing somewhat random limitations now.

2.) The assumption that a symmetric channel network (on the 2D level of the photomask) ensures even flow rates across all 8 drop makers is wrong. In 3D, slight variations in the height of the photoresist (inevitably caused by e.g. thickening towards the outside of the wafer) will cause significant differences in the hydrodynamic resistance. The authors should either provide quantitative data on the polydispersity of the droplets, or simply admit that they did work with varying droplet sizes.

Apart from these points, the manuscript seems ready for publication from my side.

Reviewer #2 (Report for the authors (Required)):

The authors have addressed my concerns

Reviewer #3 (Report for the authors (Required)):

The authors have responded to the reviews with additional data and in depth explanations of several points. I am satisfied with the responses and believe the manuscript is suitable for publication.

Mon 03 Jul 2023

Decision on Article NBME-22-2865B

Dear Dr Wong,

Thank you for your revised manuscript, "Revealing determinants for SARS-CoV-2 Spike-induced syncytium formation via parallel paired-cell profiling". Having consulted with the original Reviewers #1 (whose comments you will find at the end of this message), I am pleased to write that we shall be happy to publish the manuscript in *Nature Biomedical Engineering*.

We will be performing detailed checks on your manuscript, and in due course will send you a checklist detailing our editorial and formatting requirements. You will need to follow these instructions before you upload the final manuscript files.

Best wishes,

Liqian

Dr Liqian Wang
Associate Editor, Nature Biomedical Engineering

Reviewer #1 (Report for the authors (Required)):

The authors have now addressed also my last remaining comments, and I think the manuscript is now ready for publication. Congrats for this nice piece of work!

Rebuttal 1

We sincerely thank the three Reviewers for appreciating this work and their helpful and insightful suggestions. Based on the outstanding concerns of the Reviewers, we have performed substantial additional experiments and analyses to enhance the quality of our manuscript. We believe that the additional work incorporated into the revised manuscript has addressed all remaining issues, and hope that the Reviewers agree with us that the improved manuscript is now acceptable for publication in *Nature Biomedical Engineering*.

Reviewer #1:

Chan et al present a droplet microfluidic platform for high throughput screening of a) mutations in the Spike protein of SARSCovid-2 and b) cellular factors in the receiver cell, enhancing syncytia formation as a proxy for pathological impact. In general, this work could be of interest for a wider community, but for truly assessing the impact, a couple of points have to be addressed comprehensively:

We appreciate that the Reviewer finds our work could be of interest for a wider community. Below please find our specific responses to the Reviewer's comments.

1. Another method for high-throughput screening of mutations enhancing syncytia formation has been published previously (Nucleic Acids Res. 34(5):e41. doi: 10.1093/nar/gkl053. 2006) and should be discussed in the context of the current work. It does overcome many disadvantages of conventional low-throughput imaging-based methods that are mentioned in the introduction. What is the specific advantage of using droplets in the current work?

We thank the Reviewer for raising this point, which has been missed in the original submission. The screening method described in Merten et al. (Nucleic Acids Res. 34(5):e41; doi: 10.1093/nar/gkl053. 2006) coupled cell-cell fusion with the release of retroviral vector particles that package and transfer genes encoding the fusion-competent membrane protein. While we agree with the Reviewer that it does overcome the disadvantages of conventional low-throughput imaging-based methods, applying this method for studying SARS-CoV-2 Spike protein however presents technical limitations/disadvantages and is not suitable in its current form. In Merten et al.'s work, the authors screened the GaLv Env protein with only 1,995 nucleotides in size while SARS-CoV-2 Spike has a length of 3,819 nucleotide. The large size of SARS-CoV-2 Spike greatly compromises the viral packaging efficiency and thus using it as a readout would lower the sensitivity of the screening. In addition, syncytia with polyclonal sender cells would be formed during the screening process with this method. This could increase noise due to some relatively less fusion-competent cells being fused with neighbouring syncytia containing the more fusion-competent variants and enriched together as large syncytia and packed in the retroviruses. On the other hand, applying the droplet-based microfluidic system compartmentalizes individual sender cell and receiver cells in droplets, and we have confirmed that this approach results in a greater depletion for the fusion-incompetent cells (i.e., less noise) and thus offers a wider assay range than non-droplet-based screening method (i.e., size-exclusion selection-based screening method in which all sender and receiver cells are mixed as in Merten et al.'s method) in defining the enriched fusion-competent variants (Extended Data Fig. 5). We have now included these new discussions in the revised manuscript (p.5-6).

2. The selection of large syncytia using a cell strainer will result in syncytia with polyclonal sender cells. Doesn't this significantly increase the noise and false discovery rate? The authors set up a system with a cell-cell fusion rate of about 58% for droplets with at least one sender and one receiver cell. How can this be successfully used for the identification of inhibitory variants? Based on such a low fusion rate, a lot of false positives would inevitably be selected.

We sincerely thank the Reviewer for raising these points and appreciate the opportunity here to clarify the performance of our system setup. We opted for a moderate cell-cell fusion rate of about 58% to allow bi-directional screening of variants with both enhanced and depleted syncytium-forming potentials. We chose 24 hours post-mixing for screening in droplets, as a longer incubation period (e.g., 48 hours) would increase the fusion rate to ~70-80% which leaves little room to screen for cells with increased syncytium-forming potential. A shorter incubation period would also leave little room to screen for cells with decreased syncytium-forming potential. We agree with the Reviewer that the system setup could identify some false positives, and the selection of large syncytia using a cell strainer will result in syncytia with polyclonal sender cells. Some relatively less fusion-competent cells could fuse with neighbouring syncytia containing the more fusion-competent variants and get enriched together as large syncytia with polyclonal sender cells. This could increase the noise. To verify this point, we have now analysed the assay range in defining fusion-(in)competent variants using the size-exclusion selection-based strategy with different experimental parameters: 1) GFP-positive cells that passed through the strainer after 24hr post-mixing, which are the smaller-sized syncytia; 2 and 3) cells that were collected on strainer after 24hr (2) and 48hr (3) post-mixing, which contain the medium- and large- sized syncytia, respectively (Extended Data Fig. 5). We found that the "Cells on strainer" approaches resulted in less depletion for the fusion-incompetent cells than the "Cell passing through strainer" approach, suggesting a greater noise and false discovery rate when studying the larger syncytia. Also, 24hr is also more preferred than 48hr to get more depletion for the fusion-incompetent cells. As recommended by the Reviewer, we have also more comprehensively evaluated how the read counts of variants in our droplet microfluidics-based and size-exclusion selection screens correlate with their syncytium-forming potency (detailed in our response to Point #4 below), and from there look at the false positive rate. Overall, we observed a high consistency ($R = 0.72$) between the droplet-microfluidics-based screen and individual validation results (Fig. 2f), highlighting that our screen provides a reasonably well quantitative measurements on the variants' syncytia-forming potentials. The droplet-microfluidics-based screen could identify syncytium-inhibiting variants, while we acknowledge that the current system setup also resulted in some false positives (i.e., 4 out of 22 inhibitory variants as compared with the individual validation assays) (Fig. 2f). In comparison, the selection of large syncytia using a cell strainer gave slightly more false positives (i.e., 6 out of 22 inhibitory variants). We have now included the above new analyses and discussions on clarifying the performance of the droplet microfluidics-based and the size-exclusion selection-based screening strategies in p.10-12 and 14 of our revised manuscript.

3. Extended Data Figure 2 nicely shows that the number of plasmids/cells of a particular variant correlates with the obtained number of corresponding reads only for a very narrow range. How could it be ensured that this range was maintained throughout the entire screening procedure (transduction, cell cultivation and proliferation, etc.)? In line with this, I was missing sequencing data showing an equal distribution of the different variants across the library immediately before

screening. How can it be ruled out that certain variants were already overrepresented in the unselected library?

We thank the Reviewer for raising these points, and we apologize for the insufficiency of the details provided. To ensure that the high-coverage library contained a sufficient representation for each variant (>500-fold coverage of each Spike variant), we used >1,500-fold more cells for lentiviral infection than the size of the library being tested. We used lentiviruses to deliver the pooled library of the Spike variants with GFP11-P2A-mCherry into HEK293T sender cells (Extended Data Fig. 2c). A multiplicity of infection (MOI) of ~0.3 was used to ensure that most cells acquired a single copy of the variant. To maintain the library distribution throughout the entire screening process, all the mCherry-positive infected sender cells were sorted out, expanded, and then cocultured with receiver A549 cells expressing ACE2, GFP1-10, and BFP for 24 hours before droplet breakage and cell fixation. We performed NovaSeq-based sequencing on the genomic DNA from the collected GFP-positive cells to quantify the abundance of each Spike variants as an index of syncytium-forming potential (Extended Data Fig. 2d). We also collected the genome DNA from the infected sender cell pool immediately before mixing with the receiver cells for comparison, and confirmed that there were no overrepresented variants in the library (Extended Data Fig. 2c). We have now clarified these points in the revised manuscript (p.9).

4. Functional validation of newly identified variants promoting or inhibiting syncytia formation has only been performed for a very few variants (Extended Figures 5 and 10). What is needed is a more comprehensive analysis on how the read counts of selected variants correlate with their syncytia-forming potency. Does any of the two setups (using droplets or a cell strainer) indeed allow to quantitatively select the very best variants for the given selection criteria or is it rather a selection of any variant above a certain threshold in terms of functionality?

We thank the reviewer's recommendations. In our revised manuscript, we have now extended our work to both functionally validate more newly identified syncytium formation-promoting and inhibiting variants, as well as perform a more comprehensive analysis on how the read counts of variants correlate with their syncytia-forming potency.

In our FPPR screen, we observed Spike variants with enhanced ability to form syncytia (Fig. 2b; Supplementary Table 2). Using the observed fold change distributions of wild-type (WT) Spike variants with synonymous codons and those with stop codons, we defined a threshold of fold change > 1.625 at which no variant with stop codons was identified and thus could minimize the identification of false positives (Extended Data Fig.5). With this, 11 syncytium-enhancing hits were identified (Supplementary Table 2). We performed individual validation assays and confirmed 9 hits exhibited greater syncytium-forming potential than wild-type (i.e., resulted in larger average size of syncytium and/or total area of syncytia (Fig. 2c-d; Extended Data Fig. 6). Among them, the K854H and A846W variants showed the greatest enhancement in forming syncytia. Our profiling result also showed that Spike variants with mutations at C840, D848, and C851 tended to have decreased syncytium formation potential (Fig. 2b; Supplementary Table 2). Individual validation assays confirmed that mutations at C840, D848, and C851 reduced syncytium formation, when compared to wild-type Spike (Fig. 2c-d).

To evaluate the data quality of our droplet-microfluidics-based screen more comprehensively, we have further analysed how the screen read counts of variants correlate with

their syncytium-forming potency. In addition to the above 14 validated variants, we randomly picked 27 variants in the FPPR library and validated their syncytium-forming potentials using individual assays (Extended Data Fig. 6). Overall, we observed a high consistency ($R = 0.72$) between the droplet-microfluidics-based screen and individual validation results (Fig. 2f), highlighting that our screen provides quantitative measurements on the variants' syncytia-forming potentials. Among the total of 41 variants, 19 of them were found to be syncytium-enhancing in the individual validation assays. 16 out of the 19 variants were discovered as syncytium-enhancing "hits" in our screen (Fig. 2f), indicating a high true discovery rate of our system in defining syncytium-enhancing mutations. The droplet-microfluidics-based screen could also identify syncytium-inhibiting variants, albeit to some "false positives" (i.e., 4 out of 22 inhibitory variants validated in the individual assays) could be detected (Fig. 2f). The precision, recall, and accuracy of our screen were 80.0%, 84.2%, and 82.9%, respectively.

We also evaluated the data quality collected from the screen via the size-exclusion selection strategy. Among the 19 (out of 41) FPPR library variants of which their syncytium-enhancing potentials were individually validated (Extended Data Fig. 6), 17 of them were discovered as syncytium-enhancing "hits" (with fold change > 1) in our screen using the GFP-positive small syncytia collected via FACS (Fig. 3d). This gives a similar true discovery rate to the droplet microfluidics-based screening approach. The hit number dropped to 10 and 9 when screening larger syncytia collected and remained on the cell strainer after 24- and 48- hour post-mixing, respectively (Fig. 3d), resulting in reduced true discovery rates. Indeed, we noted that the cells-remain-on-strainer-based strategy resulted in less depletion for the fusion-incompetent cells in the pooled assay, which gives greater noise and a narrower assay range in defining the enriched fusion-competent variants (Extended Data Fig. 5). The non-compartmented cell pool in the size-exclusion selection-based system and the longer duration allowed for syncytia formation potentially favor the trapping of fusion-incompetent cells by the neighbouring syncytia as bystanders and their retention on the cell strainer. Some relatively less fusion-competent cells may also be fused with neighbouring syncytia containing the more fusion-competent variants and enriched together as large syncytia with polyclonal sender cells. These could account for the relatively less enrichment and likelihood of the true syncytium-enhancing variants (particularly the weaker ones) to be discovered as hits and more non-enhancing variants being isolated as false positives, thus increasing the false discovery rate. Among the three experimental parameters for the size-exclusion-based selections, allowing a shorter (i.e., 24 hour) duration to form smaller-sized syncytia, in particular ones that are small enough to be collected by FACS is more recommended.

Overall, the two setups (using droplets or the size-exclusion selection that collect GFP-positive small syncytia via FACS) performed reasonably well in quantitatively evaluating the variants' syncytium-forming potency and offer a high true discovery rate in selecting syncytium formation-enhancing with a threshold defined using the observed fold change distributions of variants with stop codons. Selecting either the droplet microfluidics-based or the size-exclusion selection-based screening strategy could depend on the desired sensitivity and throughput of the genetic screen to be performed (Supplementary Table 1). We have now included the above new data and discussion in p.10-12 and 14. of our revised manuscript.

Furthermore, we have now performed more functional validations of our Furin cleavage site library screen hits (original Ext Data Fig. 5; now Ext Data Fig. 8 and discussed in p.12-13 of our revised manuscript) and CRISPR screen hits (original Ext Data Fig. 10; now Figure 4b-d plus Ext Data Fig. 17 and discussed in p.16 and 18-19 of our revised manuscript). For the Furin

cleavage site library screen, using the threshold defined with the observed fold change distributions of variants with stop codons, 5 syncytium-enhancing hits were identified (Supplementary Table 2) and 4 of them (i.e., R683H, A684H, V687I, and A688T) were successfully validated with individual assays showing larger average size of syncytium and/or total area of syncytia (Extended Data Fig. 8a-b). The P681Y mutation was more recently reported in an immunocompromised patient with persistent SARS-CoV-2 omicron BA1 subvariant replication (Gonzalez-Reiche et al., medRxiv, 2022: <https://doi.org/10.1101/2022.05.25.22275533>). We also identified this mutation as a syncytium-enhancing mutation in our Furin cleavage site library screen, and our individual validation assays also confirmed this effect (Extended Data Fig. 8a-b). For the CRISPR screen, we have now increased our hit validations to a total of 5 syncytium formation-inhibiting hits (10 sgRNAs targeting *FCHO2*, *AP2M1*, *CAB39*, *RNF2*, and *GBP6*) identified at arbitrary cut-offs of RRA score > 4 and fold change > 1.5 as potential host factors that play important roles in syncytium formation (Fig. 4b-d; Supplementary Table 6). We further added the validation of 3 top syncytium formation-promoting hits (6 sgRNAs targeting *ZEB1*, *UBIAD1*, and *NDUFB10*), indicating that our screen could also uncover gene knockouts that enhance syncytium formation (Extended Data Fig. 17), although it is not the focus of this current study. In sum, our additional work has further confirmed the validity of our screens in isolating the syncytium formation-promoting and inhibiting hits.

Minor things:

5. The scale bar for the heat maps in Fig. 2b is too small.

We thank the Reviewer's suggestion. We have now enlarged the scale bar in Fig. 2b.

6. The Chip design shown in Extended Data Figure 1b inevitable results in droplets of different size and occupancy (the contents of the single cell- and oil-inlets will be distributed unevenly across the eight drop makers). Why not using a single droplet maker, capable of generating thousands of droplets per second?

We thank the Reviewer for raising this point, and we apologize for the insufficiency of the above information. The main reason of using a 8-channel device in this study is to increase the throughput. Although the droplet generation frequency with a single-channel device can be as high as 1.5kHz, the generation time could be as long as 8 hours for generating 42 millions of droplets needed for a screening experiment of ~1,000 variants. Such a long duration of microfluidic processing is not desired as it reduces cell viability. In comparison, the 8-channel device can maintain an ultrahigh overall frequency of 12kHz to complete the droplet generation process in ~1 hour. To minimize the uneven distribution of the contents of the single cell- and oil-inlets across the eight drop makers, we designed a symmetrical layout for the 8 channels such that the flow rates of the aqueous and oil phases can be evenly distributed in each channel. We optimized the flow rates of two phases to generate monodispersed droplets for encapsulating human cells. In our experiments, we observed droplets with a narrow range of diameters ranging from 70-76 μm (Extended Data Fig. 1b). We have now included these clarifications in our revised manuscript (p.7-8).

Reviewer #2:

Charles W. F. Chan et al. developed two screening approaches for studying determinants of cell-cell fusion. First, they describe a microfluidic strategy to encapsulate cell pairs into droplets. The pairs which fuse are detected by GFP complementation using flow cytometry and sorted for further characterization. They performed a deep mutational scan of two SARS-CoV-2 spike regions identifying several new mutations impacting syncytia formation. Second, they describe a strategy based on size exclusion, as a higher-throughput alternative to the compartmentation method. They performed a genome wide screen and identify core regulators of clathrin-mediated endocytosis as potential host factors that play a role in S-induced syncytium formation, which they then validate experimentally. The approaches developed in this paper are innovative and fill a methodological gap as high-throughput screening methods for cell-cell fusion determinants are missing. The techniques may have many applications for the study of cell-cell fusion in physiological and pathological settings.

We are grateful for the Reviewer's support of our paper. We thank the Reviewer's comments that the approaches developed in this paper are innovative and fill a methodological gap as high-throughput screening methods for cell-cell fusion determinants are missing. We also appreciate the reviewer's remarks that our techniques may have many applications for the study of cell-cell fusion in physiological and pathological settings. Below please find our specific responses to the reviewer's suggestions.

Main comments

1. Fig. 2: Why did the authors only validate 2 mutations which increase syncytia and 3 which decrease? There are many other mutations in the screen such as: A852M, D843M, G838K, D843G/Y, I844D and others. If there was a cut-off, even if arbitrary, it should be indicated

We sincerely thank the Reviewer for raising this point, and we apologize for the insufficiency of the details provided. In our FPPR screen, we observed Spike variants with enhanced ability to form syncytia (Fig. 2b; Supplementary Table 2). Using the observed fold change distributions of wild-type (WT) Spike variants with synonymous codons and those with stop codons, we defined a threshold cut-off of fold change > 1.625 at which no variant with stop codons was identified and thus could minimize the identification of false positives (Extended Data Fig. 5). With this, 11 syncytium-enhancing hits were identified (Supplementary Table 2). We have performed additional individual validation assays and confirmed 9 hits exhibited greater syncytium-forming potential than wild-type (i.e., resulted in larger average size of syncytium and/or total area of syncytia (Fig. 2c-d; Extended Data Fig. 6). Among them, the K854H and A846W variants showed the greatest enhancement in forming syncytia. Our profiling result also showed that Spike variants with most if not all mutations at C840, D848, and C851 tended to have decreased syncytium formation potential (Fig. 2b; Supplementary Table 2). Individual validation assays confirmed that the three randomly selected mutants (i.e., C840S, D848N, and C851S) reduced syncytium formation, when compared to wild-type Spike (Fig. 2c-d).

Because of the long lists of potential syncytium formation- promoting and inhibiting variants and the arbitrariness of setting cut-offs, we have also performed a more comprehensive analysis to evaluate how the screen read counts of variants correlate with their syncytium-forming potency. In addition to the abovementioned 14 validated variants, we randomly picked

27 variants in the FPPR library and validated their syncytium-forming potentials using individual assays (Extended Data Fig. 6). Overall, we observed a high consistency ($R = 0.72$) between the screen and individual validation results (Fig. 2f), highlighting that our screen provides a reasonably well quantitative measurements on the variants' syncytia-forming potentials. More specifically, among the total of 41 variants, 19 of them were found to be syncytium-enhancing in the individual validation assays. 16 out of the 19 variants were discovered as syncytium-enhancing "hits" in our screen (Fig. 2f), indicating a high true discovery rate of our system in defining syncytium-enhancing mutations. The droplet-microfluidics-based screen could also identify syncytium-inhibiting variants, albeit to some false positives (i.e., 4 out of 22 inhibitory variants validated in the individual assays) could be detected (Fig. 2f). The precision, recall, and accuracy of our screen were 80.0%, 84.2%, and 82.9%, respectively. Our additional work here has further confirmed the validity and clarified the performance of our screen in isolating the syncytium formation- promoting and inhibiting hits. We have now included the above new analyses and discussion in p.10-12 and 14. of our revised manuscript.

2. Fig 2.e. and Extended Fig. 5. – Surface level quantification of S expression by flow cytometry on the sender cells should be checked to eliminate transfection bias.

We thank the Reviewer's suggestion. We have now included the surface level quantification of Spike expression by flow cytometry on the sender cells (using a S2 antibody that targets the S2 subunit of SARS-CoV-2 Spike) and confirmed the similar cell surface expression level of the Spike variants. The observed syncytium formation- promoting/inhibiting effects were unlikely due to transfection bias. These results are now included in Extended Data Fig. 3 (to supplement Fig 2e) and Extended Data Fig. 8c (to supplement the original Extended Data Fig. 5, now Extended Data Fig. 8a-b) in our revised manuscript.

3. Extended Fig. 2. – Axis legend only states "replicates 1-2". What do the values on the axis represent?

We apologize for the missing information. We have added back the information to the axis legends and figure legend of Extended Data Fig. 2. The values represent the fold change comparing each variant's relative abundance in GFP-positive cell pool versus the cell pool before mixing and normalized to wild-type.

4. Figure 2 b. – It is surprising that R685A does not have a greater impact as it should abrogate furin cleavage and be similar to a Δ PRRA mutant and therefore inhibit syncytia. Could you comment on the high mutability of the R685 site in this screen?

We thank the Reviewer for raising this point. Our DMS results revealed that most single mutations at PRRA and its neighbouring sequence including the basic residue R685 were not sufficient in abolishing syncytium formation (Fig. 2b). This suggests that despite the high mutability of these sites, many of the single mutants can be efficiently cleaved by the proteases. This could be attributed to the rather flexible motif sequence (i.e., XB₂BBX, where B is a basic amino acid residue and X is a hydrophobic residue) that the furin protease (or other proteases) could recognize. We have now included this discussion in p.12-13 of our revised manuscript.

5. Fig. 3. c – Bottom axis legend is missing.

We apologize for the missing information. We have added back the information to the bottom axis and the figure legend of Fig. 3c. The values represent the fold change (FC) observed.

6. Extended fig. 3 – It is unclear how the ACE2 binding ability was calculated for each of the single mutations. Was this performed on the pool of transduced cells, and if so, how were the mutations correlated to ACE2 binding? Was it done on individually transduced cells? This should be further detailed in the figure legend and M&M.

We apologize for the missing information. Cells transduced with the FPPR DMS library pool were used for the ACE2 binding assay. The FC value represents the ACE2 binding ability for each of the single mutations in the library pool. This value is calculated as the fold change comparing each variant's relative abundance in FACS-sorted (i.e., AF405-positive) ACE2-bound cell pool versus the unsorted cell pool of Spike's FPPR variants and is normalized to wild-type. We have added these information to the Methods section and the legend of Extended Fig. 3.

Minor comments:

1. Fold changes (FC) and how data are normalized should be better described in the text or figure legends.

We thank the Reviewer for this suggestion. In our droplet microfluidics-based DMS screens, spike variants that have increased syncytium formation potential were enriched (with fold change > 1 , comparing each variant's relative abundance in GFP-positive cell pool versus the cell pool before mixing and is normalized to wild-type (WT)), while those that have decreased syncytium formation potential and those that acquired a premature stop codon after mutation were depleted (with fold change < 1 , also comparing each variant's relative abundance in GFP-positive cell pool versus the cell pool before mixing and is normalized to WT). Similarly, in our size-exclusion selection-based DMS screens, the fold change is determined by comparing each variant's relative abundance in the syncytia collected using the cell strainer or small GFP⁺ syncytia are collected by FACS versus the cell pool before mixing and is normalized to wild-type (WT)). In our CRISPR screen, FC represents each variant's relative abundance in the unfused receiver cell pool versus the unmixed cell pool and is normalized to WT. We have now added these descriptions in p.9-10 of our revised manuscript and the legends of Figure 2, Figure 3, Figure 4, and Extended Data Fig. 2.

2. FPPR (fusion-peptide proximal region) is not defined

We apologize for the error made in the definition. We have now updated the definition in p.8 of our revised manuscript.

3. Some sentences are very long, like line 269: "Using size-exclusion selection-based strategy thus offers a simple and scalable way to perform large-scale DMS for syncytia screening, whilst we noted that the droplet microfluidics-based screening strategy resulted in a greater depletion for the fusion-incompetent cells in the pooled assay which gives wider assay range in defining the enriched fusion-competent variants (Extended Data Fig. 7), which could be due to some

fusion-incompetent cells being trapped by neighbouring syncytia as bystanders and thus retained on the cell strainer.”

We thank the Reviewer’s suggestion. We have now separated and modified the sentence as “Using size-exclusion selection-based strategy thus could offer a simple and scalable way to perform large-scale DMS for syncytia screening.” and “Indeed, we noted that the cells-remain-on-strainer-based strategy resulted in less depletion for the fusion-incompetent cells in the pooled assay, which gives greater noise and a narrower assay range in defining the enriched fusion-competent variants (Extended Data Fig. 5). The non-compartmented cell pool in the size-exclusion selection-based system and the longer duration allowed for syncytium formation potentially favor the trapping of fusion-incompetent cells by the neighbouring syncytia as bystanders and their retention on the cell strainer.” in p.14 of our revise manuscript.

Reviewer #3:

In this work, the authors used droplet microfluidics to scan SARS-Cov-2 mutations associated with syncytium formation. A GFPsplit complementation system was used to detect sender-receiver cells fusion (i.e., GFP-positive syncytia) using FACS. The authors used a pooled saturation mutagenesis library and identified syncytia-forming mutations, some of which were previously identified. The authors then carried out a genome-wide KO screen to identify the genetic host factors contributing to syncytia formation. FACS was used for screening after removing large syncytia with a cell strainer. The top hits were functionally validated with CRISPR KOs and two drugs were screened with known inhibitory effect of clathrin-mediated endocytosis (CME). Overall, SARS-Cov-2 remains a major concern and identifying factors contributing to syncytia formation in the lung is interesting. However, the engineering-related aspects are not the main focus but perhaps the biology is novel enough to stand on its own. Below are additional comments.

We are grateful that the reviewer finds our work to identify factors contributing to SARS-Cov-2 syncytia formation is interesting, and the biology is novel enough to stand on its own. Below please find our specific responses to the reviewer’s comments.

1) The droplet microfluidic system used in this study is very standard.

We thank the Reviewer for raising this point. In our droplet microfluidic system design, we integrated 8 channels into one device. Although the generation format for each channel is standard, important modifications have been made to the system. For example, the integration allows us to run 8 channels at the same time, thus achieving an ultrahigh throughput of around 12 kHz. This integration step is important to enhance the throughput and make our experiments practicable. Although the droplet generation frequency with a standard single-channel device can be as high as 1.5kHz, the generation time could be as long as 8 hours for generating 42 millions of droplets needed for a screening experiment of ~1,000 variants. Such a long duration of microfluidic processing is not desired as it reduces cell viability. In comparison, the 8-channel device can complete the droplet generation process in ~1 hour. To minimize the uneven distribution of the contents of the single cell- and oil-inlets across the eight drop makers, we designed a symmetrical layout for the 8 channels such that the flow rates of the aqueous and oil

phases can be evenly distributed in each channel. In our experiments, we observed droplets with a narrow range of diameters ranging from 70-76 μm . We have clarified the important point raised by the reviewer on the original manuscript, and have now included these clarifications in our revised manuscript (p.7-8).

2) No information was provided on the size of syncytia formed which is very important. Given that the authors have already identified a panel of mutations, it might be interesting to correlate each mutation with the average size of formed syncytium.

We thank the Reviewer's suggestions. We have now performed additional validation experiments, and with a total of 41 individually validated FPPR variants, we observed a good correlation ($R = 0.85$) between the quantified average size of the formed syncytium and total syncytium area (Extended Data Fig. 6c). We have also quantified the size of syncytia formed in all our validation experiments in our revised manuscript (Figs. 2d; 4d, g, h; 5b, d; Extended Data Figs. 6; 8b; 11a; 17c).

3) It is true that using a cell strainer is a size-exclusion approach, however it would have been more interesting if the authors could have used this type of high-throughput microfluidics approach for that purpose.

We agree with the Reviewer that it will be interesting to do size selection using droplet microfluidics, but it is not adopted in the current work for the following reason. The droplet screening is usually processed at the frequency of 1 kHz; processing of one million cells/ ten million droplets (encapsulation rate of 10%) needs around 2.8 hr. In contrast, the cell strainer-based method can easily process millions of cells within several seconds, offering an ultrahigh throughput in its current form. Therefore, we selected the cell strainer-based method in our study. As an interesting way forward, we have now discussed the future opportunity (in p.15 of our revised manuscript) to develop a new droplet microfluidics system to achieve cell size measurement and integrate with fluorescence as dual readouts to further enhance the screening data quality. Potential challenges to be solved for such system may include the following. After encapsulation, each cell can move to and be spotted at a different focal plane in the droplet (Fig. a and b below). The same cell would be quantified to have a different size at a different focal plane. Also, when a cell moves towards the interface (Fig. c below), its cell measurement would be affected. Thus, more analytical efforts are needed to accurately measure the cell size through droplet screening.

The same cell inside the droplet (a) at the focal plane, (b) out of the focal plane, and (c) near the interface.

4) Several CRISPR screens were previously performed to identify the host factors that promote SARS-Cov-2 infection. The authors might need to check to see whether similar hits were recovered.

We thank the Reviewer for suggestion. The CRISPR screen performed in this study was designed to specifically look for determining factors for Spike-induced syncytium formation given its impact on the disease severity, which differs from the prior genome-wide CRISPR screens that gave an overview of host factors involved in the virus infection process and life cycle and thus more and different hits may be identified in those screens. Specifically, we found that a few other CME-related genes *APIGI*, *AP1B1*, *AAGAB* were scored as hits in some of the previous SARS-CoV-2 virus infection-based CRISPR screens (Rebendenne et al., *Nat. Genet.*, 2022: doi: 10.21203/rs.3.rs-555275/v1; Biering et al., *Nat. Genet.*, 2022: doi: 10.1038/s41588-022-01131-x; Israeli et al., *Nat. Comm.* 2022: doi: 10.1038/s41467-022-29896-z), the potent syncytium formation-modifying hits (including *FCHO2* and *AP2MI*) identified in our screen were however not previously uncovered, emphasizing the different aspects of viral biology are revealed by these screening platforms. We have now included this information in p.20 of our revised manuscript.

5) The drug screen should be extended because the drugs tested were previously evaluated for their ability to inhibit host factors (M. Grodzki et al., *Genome Medicine*, 2022, 10). Also, there are more than 3,000 FDA/EMA-approved drugs with therapeutic effects on syncytia during SARS-Cov-2 infection. Thus, it might be interesting to extend the drug screen and probably validate the drugs *in vivo* using a mouse model.

We thank the Reviewer's suggestion. The Grodzki et al paper (*Genome Medicine*, 2022: doi.org/10.1186/s13073-022-01013-1) showed that the CME inhibitor Promethazine reduces SARS-CoV-2-induced cytotoxicity in Vero E6 cells. In another study in which a screen of more than 3,000 approved drugs was performed (Braga et al., *Nature*, 2021: doi.org/10.1038/s41586-021-03491-6), Promethazine, Fluvoxamine, and Itraconazole (ITZ) were scored to have some inhibitory effects on SARS-CoV-2-induced syncytium formation in Vero E6 cells, albeit not further characterized in the study. Here we carried out a more comprehensive evaluation of the effect of five approved endocytosis inhibitors (i.e., Chlorpromazine (CPZ), Fluvoxamine, and Promethazine, Imipramine, ITZ). In our experiments, we found that treatment with the three CME inhibitors (i.e., CPZ, Fluvoxamine, and Promethazine, but not Imipramine and ITZ which both primarily affect micropinocytosis) greatly impeded syncytium formation in both A549-ACE2 and Vero E6 cells (Fig. 5a-d). The drug doses used did not affect cell viability (Extended Data Fig. 13). We further moved to validate the findings using authentic SARS-CoV-2 in both *in vitro* and *in vivo* experiments. We confirmed that treatment with all three CME inhibitors (CPZ, Fluvoxamine, and Promethazine) inhibited syncytium formation in the SARS-CoV-2 D614G-infected cells (Fig. 6a-b). In addition, the three inhibitors also greatly reduced the expression levels of SARS-CoV-2 RNA-dependent RNA polymerase (RdRp) in both cell lysate and supernatant samples after virus infection (Fig. 6c; Extended Data Fig. 15a), indicating the reduced production of new viruses. In line with our *in vitro* results, treatment of CPZ and Fluvoxamine reduced the virus RdRp gene expression level (Fig. 6e), the amount of SARS-CoV-2 infectious particles (Fig. 6f), the area of positivity of the SARS-CoV-2 nucleocapsid protein (Fig. 6g), and bronchiolar epithelium damage, alveolar congestion, infiltration and haemorrhage (Extended Data Fig. 16) in the lung tissues of the virus-infected hamsters. Also, less syncytium-like multinucleated cells were detected within the SARS-CoV-2 nucleocapsid-positively stained lung tissues in the CPZ- and Fluvoxamine- treated hamsters (Fig. 6g). These results support that the CME inhibitors reduce the viral load, spread, and syncytium formation, in the SARS-CoV-2-

infected lung tissues, and demonstrate the *in vivo* relevance of our findings. We have now included these substantial additional experiments and analyses in p.17-18 of our revised manuscript.

6) The authors need to perform a thorough characterization of the molecular mechanisms of action of the candidate hits. RNAseq reads can be aligned to the human genome and reads for each gene can be determined by STAR analysis.

We sincerely thank the Reviewer's suggestion. From our CRISPR screen, knockout of *FCHO2* and *AP2M1* reduced syncytium formation induced by wild-type SARS-CoV-2 Spike at a greater extent than the other gene hits (Fig. 4c-d) and they also inhibited omicron Spike-induced syncytium formation (Extended Data Fig. 11a), *FCHO2* and *AP2M1* were thus selected for further characterization. *AP2M1* and *FCHO2* are core regulators of clathrin-mediated endocytosis (CME). To perform a thorough characterization of the molecular mechanisms of action of these hits, we have performed RNA-seq and gene ontology enrichment analysis on *FCHO2* and *AP2M1* knockout A549-ACE2 cells. Our results revealed the positive regulation of cell-substrate/matrix adhesion (Extended Data Fig. 12), among other processes, in both *FCHO2* and *AP2M1* knockout cells. Cell-substrate/matrix adhesion was reported to increase the force required for deforming a membrane during clathrin-coated vesicle formation and inhibit CME (Batchelder et al., Mol Biol Cell, 2010; doi: 10.1091/mbc.E09-12-1044). Furthermore, we showed that genetic knockdown of *clathrin heavy chain (CHC)* (Fig. 4e-g) and treatment with a clathrin inhibitor Pistop 2 (Fig. 4h) both suppressed the cell-cell fusion process. All the above results support the involvement of CME in Spike-induced syncytium formation. We have included these new results and discussion in p.17 of our revised manuscript.

In addition, we sought to gain additional insights into the molecular mechanisms by which the top mutations identified by our DMS screens lead to enhanced syncytium formation. We performed additional experiments to evaluate whether some of our validated syncytium formation-promoting variants may increase the cleavage of Spike to facilitate its membrane fusion and syncytium formation. We found that the syncytium formation-enhancing P681Y mutant (Extended Data Fig. 8a-b) showed increased S1 subunit cleavage at the cell surface expression. Interestingly, such increase was detected even in cells with minimal furin and TMPRSS2 expressions (Extended Data Fig. 8c-d), suggesting that the mutation may allow S1 subunit cleavage to be aided by other proteases. This P681Y mutation was more recently reported in an immunocompromised patient with persistent SARS-CoV-2 omicron BA1 subvariant replication (Gonzalez-Reiche et al., medRxiv, 2022: doi.org/10.1101/2022.05.25.22275533). Our result provides a plausible mechanism for this P681Y variant to enhance Spike's cleavage and syncytium formation, further supporting that the potential emergence of P681Y mutation should be monitored. We have included these new results and discussion in p.12 of our revised manuscript.

7) The title "massively parallel paired-cell profiling" might need to be reconsidered. I do not think the authors have provided any novel tools to facilitate that other than using a standard droplet microfluidics system and FACS. Likewise, in the Conclusion section "The application of our high-throughput profiling systems together with CRISPR"?? I am not sure if referring to standard droplet microfluidics and FACS using "our" is accurate as well.

We thank the Reviewer's recommendations. We have now removed "our" in the sentence in the Conclusion section to read as: "The application of high-throughput profiling systems together with CRISPR...". For the title of the paper, we have updated our title as "Revealing determinants for SARS-CoV-2 Spike-induced syncytium formation via parallel paired-cell profiling". As detailed in our response to Point #1, our droplet microfluidic system integrates 8 channels with a symmetrical layout into one device to enhance the throughput and make our experiments practicable. Therefore, we believe "parallel pair-cell profiling" is accurate in describing the method that we have used in this work.

Rebuttal 2

We sincerely thank the three Reviewers for appreciating this work and their helpful suggestions. We have now updated our manuscript based on the remaining comments. We hope that our manuscript is now acceptable for publication in *Nature Biomedical Engineering*.

Reviewer #1:

In the revised version of their manuscript, Chan et al. provide a lot of new additional experimental data, addressing most of my comments comprehensively. Well done and thanks a lot for all the extra efforts!

We appreciate that the Reviewer finds our new additional work and efforts have addressed most of the comments comprehensively. Below please find our specific responses to the Reviewer's remaining comments.

There are just two points that still need some correction:

1.) The following statement is simply wrong: "A pool-based method coupling cell-cell fusion with a screening readout of retroviral vector particle packaging and release that transfer genes encoding the fusion-competent membrane protein was reported ²¹. However, this method is not suitable for studying SARS-CoV-2 Spike protein due to its large size that greatly compromises the viral packaging efficiency and thus lowers the sensitivity of the screening. Also, syncytia with polyclonal sender cells would be formed during the screening process with this method, which could increase noise due to some relatively less fusion-competent cells being fused with neighbouring syncytia containing the more fusion-competent variants and enriched together as large syncytia and packed in the retroviruses." First of all, the cited method made use of packaging constructs much bigger than the envelope protein (only corresponding to about 20% of the entire packaged sequence). Therefore, a 2-fold larger size of the SARS-CoV-2 Spike protein would probably pose absolutely no problem. Second, the former method also managed to prevent polyclonal sender cells, e.g. by adjusting cell density and cell type ratios. Taken together, the former method could have been applied to the problem addressed in the current work as well, and this should be discussed in a fair and balanced way. The authors openly admitted in their rebuttal letter that they initially overlooked this work, and this should in no ways be "compensated" by attributing somewhat random limitations now.

We thank the Reviewer for raising this point. We have updated the discussion in the revised manuscript (p.5). It now reads as "A pool-based method coupling cell-cell fusion with a screening readout of retroviral vector particle packaging and release that transfer genes encoding the fusion-competent membrane protein was reported ²¹. This method could be adopted for studying the SARS-CoV-2 Spike protein. By adjusting cell density and cell type ratios in such pooled experiment would minimize syncytia with polyclonal sender cells, which increase noise due to some relatively less fusion-competent cells being fused with neighbouring syncytia containing the more fusion-competent variants and enriched together as large syncytia."

2.) The assumption that a symmetric channel network (on the 2D level of the photomask) ensures even flow rates across all 8 drop makers is wrong. In 3D, slight variations in the height of the photoresist (inevitably caused by e.g. thickening towards the outside of the wafer) will cause significant differences in the hydrodynamic resistance. The authors should either provide

quantitative data on the polydispersity of the droplets, or simply admit that they did work with varying droplet sizes. Apart from these points, the manuscript seems ready for publication from my side.

We thank the Reviewer for raising this point. We have now updated the description as “We optimized the flow rates of two phases to generate droplets for encapsulating human cells. In our experiments, we observed droplets with diameters ranging from 70-76 μm . The coefficient of variation (CV) was 1.5%, which is par with other microfluidics settings that generate droplets with a CV of less than 3% (Zhu and Wang, Lab on a Chip, 2016: doi: 10.1039/c6lc01018k)” in the revised manuscript (p.8).

Reviewer #2:

the authors have addressed my concerns

We are grateful for the Reviewer’s support of our paper.

Reviewer #3:

The authors have responded to the reviews with additional data and in depth explanations of several points. I am satisfied with the responses and believe the manuscript is suitable for publication.

We are grateful for the Reviewer’s support of our paper.